# Zfp260 choreographs the early stage osteo-lineage commitment of skeletal stem cells

Yuteng Weng [1,2], Yanhuizhi Feng[1,2], Zeyuan Li [1,2], Shuyu Xu[1,2], Di Wu[1,2], Jie Huang [1,2], Haicheng Wang [1,2] & Zuolin Wang [1,2] ✉

The initial fine-tuning processes are crucial for successful bone regeneration, as they guide skeletal stem cells through progenitor differentiation toward osteo- or chondrogenic fate. While fate determination processes are well-documented, the mechanisms preceding progenitor commitment remain poorly understood. Here, we identified a transcription factor, Zfp260, as pivotal for stem cell maturation into progenitors and directing osteogenic differentiation. Zfp260 is markedly up-regulated as cells transition from stem to progenitor stages; its dysfunction causes lineage arrest at the progenitor stage, impairing bone repair. Zfp260 is required for maintaining chromatin accessibility and regulates Runx2 expression by forming super-enhancer complexes. Furthermore, the PKCα kinase phosphorylates Zfp260 at residues Y173, S182, and S197, which are essential for its functional activity. Mutations at these residues significantly impair its functionality. These findings position Zfp260 as a vital factor bridging stem cell activation with progenitor cell fate determination, unveiling a element fundamental to successful bone regeneration.

The process of bone regeneration following injury was intricately controlled and primarily driven by skeletal stem cells (SSCs) lineage cells[1]. These SSCs, located within bone tissue, possessed the ability to self-renew and differentiate along the osteogenic lineage, playing a crucial role in bone repair and maintenance[2]. In vivo studies showed that SSCs labeled with distinct markers exhibited specific distribution patterns. A subset of SSCs identified by the marker Ctsk, found in the periosteum of long bones, cranial bones, and jawbones, was commonly referred to as periosteal stem cells (PSCs)[3]. During the process of fracture healing, PSCs contributed to repair through both intramembranous and endochondral ossification. In the maxillofacial region, our previous study indicated that an epithelium-derived progenitor cell, identified by the co-expression of Krt14 and Ctsk, originating from the Schneiderian membrane that covered the maxilla, governed maxillofacial bone regeneration through intramembranous ossification[4].

The differentiation of SSC lineage cells proceeded sequentially, transitioning from SSC to BCSP and subsequently to chondro-lineage (PCP), osteo-lineage (THY, BLSP), and stromal-lineage (6C3, HEC) cells irreversibly[5]. This differentiation process was regulated by multiple signaling pathways, including Hedgehog[6], BMP[7], and Wnt[8]. The prompt and precise differentiation of stem cells was essential for bone regeneration following injury, and any impediment to this process could result in suboptimal bone healing outcomes[9], including bone defects, delayed healing[10], and non-union[11]. Thus, a comprehensive understanding of the regulatory mechanisms dictating the osteogenic differentiation fate of SSCs held substantial clinical and research significance.

The temporal activation of transcription factors played a pivotal role in SSC differentiation. Prior research established Runx2 and Sox9 as crucial transcription factors governing the determination of osteogenic and chondrogenic fates in progenitors[12,13]. Additional transcription factors implicated in osteogenic and chondrogenic processes, including Dlx5, YAP/TAZ, and SMAD, have been shown to simultaneously modulate the expression levels of Runx2 and Sox9,

[1]Shanghai Engineering Research Center of Tooth Restoration and Regeneration & Tongji Research Institute of Stomatology, Shanghai 200072, China. [2]Department of Oral and Maxillofacial Surgery, Department of Oral Implantology, Shanghai Tongji Stomatological Hospital and Dental School, Tongji University, Shanghai 200072, China. ✉e-mail: zuolin@tongji.edu.cn

thereby influencing bone regeneration[14]. Nevertheless, recent investigations proposed the presence of a transitional state, termed pre-BCSP, within the differentiation pathway from SSC to BCSP[1]. The precise importance and underlying regulatory pathways of the unidirectional differentiation process from SSCs to BCSPs have yet to be fully elucidated.

In this study, we utilize models of long bone fractures and maxillary sinus floor lifting (MSFL) to screen for transcription factors (TFs) involved in the initial phases of bone repair and regeneration. We identify Zfp260 as a critical regulatory factor driving the differentiation of SSC into BCSP. Deletion of Zfp260 compromises fracture healing and MSFL bone regeneration, leading to an accumulation of precursor BCSP. Our study, using ChIP-seq, CUT&Tag, and ATAC-seq techniques, reveals that Zfp260 formed a super-enhancer complex controlling the expression of key transcription factors such as Runx2 and Zfhx4, thereby influencing osteogenic fate determination and altering their ATAC signal accessibility during the BCSP stage. Furthermore, our findings indicate that phosphorylation of Zfp260 by the critical kinase PKCα at residues Y173, S182, or S197 regulated its nuclear localization and transcriptional activity. Subsequent in vivo and in vitro rescue experiments further validate the functional significance of Zfp260 phosphorylation modifications. These findings suggest that Zfp260 is crucial in regulating the transition from SSCs to BCSPs, linking the onset of stem cell differentiation with progenitor cell fate determination in bone injury repair.

## Results

### Zfp260 exhibited an early-stage upregulation expression pattern in the SSC lineage

In the context of bone injury repair, the early differentiation of the SSC-to-BCSP transition was associated with the initial deposition of the extracellular matrix. Nevertheless, the specific regulatory mechanisms governing this pivotal process have yet to be fully elucidated (Fig. 1a). To identify potential universal regulatory factors, an analysis was conducted on single-cell and bulk RNA sequencing data obtained from the femur fracture model encompassing both intramembranous and endochondral ossification, as well as from the model of MSFL (intramembranous-like ossification only) (Fig. 1b).

To identify candidate transcription factors (TFs) that satisfy the dual criteria of high expression in SSC lineage cells and relevance to early SSC fate determination, we employed the following strategy: Utilizing scRNA-seq datasets from fracture and MSFL, we filtered for TFs that are specifically highly expressed in SSC lineage cells (encompassing stem and progenitor cells, osteolineage cells, and osteochondro lineage cells). The filtering parameters applied were avg_Log$_2$FC > 0.25, pct.1 > 0.25, pct.1 - pct.2 > 0.25, and p.adj <0.05. Furthermore, a time course analysis of the bulk RNA-seq data from fracture and MSFL was conducted, with a focus on TFs significantly enriched in clusters associated with extracellular matrix organization. By intersecting the four identified TF clusters, we identified five potential candidate TFs as Zfp260, Plagl1, Snai2, Creb3l1, and Prrx1 (Fig. 1c). All clusters and their corresponding features from the time course analysis were presented in sFig. 1a, b. The TF list filtered in each module has been supplemented in the Supplementary Data 2.

To investigate the expression patterns of candidate TFs along the SSC lineage hierarchy in both fracture and MSFL models, we conducted analyzed scRNA-seq datasets in depth. Taking the fracture dataset as an example, we initially performed cell clustering using cell-specific markers[15] as illustrated in sFig. 1c, d, resulting in the identification of 13 clusters visualized by Uniform Manifold Approximation and Projection (UMAP) in Fig. 1d. Subsequently, we focused on stem/progenitor cells, osteolineage cells (OLC), and osteochondro-lineage cells (OCLC) for further subclustering as depicted in Fig. 1e. Clusters were annotated based on the expression levels of markers related to stemness (Klf4, Prrx1, Zeb1, Ctsk), osteogenesis (Alpl, Bglap, Dmp1,

Runx2, Sp7), and chondrogenesis (Acan, Col2a1, Sox9) (see sFig. 2a, b). To explain the connectivity between cell populations and reveal the pathways of cell differentiation, PAGA analysis (Fig. 1f) and trajectory analysis (sFig. 2c) were utilized. Additionally, the expression patterns of transcription factors along the SSC hierarchy were plotted by using trajectory values as the x-axis and FPKM values of TFs as the y-axis. The findings indicated a significant upregulation of Zfp260 during the transition from SSC to BCSP, with sustained high expression levels thereafter as demonstrated in Fig. 1g, a pattern consistent with trajectory analysis (sFig. 2d). Parallel analyses were conducted on the MSFL scRNA-seq dataset[4,16] (Fig. 1h-j, sFig. 3a–c), revealing a similar upregulation of Zfp260 during the SSC-to-BCSP transition in the MSFL model, particularly in the context of intramembranous ossification, distinguishing it from other candidate TFs (Fig. 1k, sFig. 3d).

Utilizing flow cytometry panels as previously described[5] (sFig. 4a, b), cells representing various stages of the SSC hierarchy were isolated from callus samples collected 14 days post-fracture, and the expression levels of transcription factors (TFs) were quantified using RT-qPCR. The findings revealed that Zfp260 was the TF that was significantly upregulated from SSC to BCSP among the five candidates. Subsequent multiplex immunohistochemistry (mIHC) was performed to assess the co-expression patterns of Zfp260, Runx2, and Sox9 with cells at different stages of the SSC hierarchy in the two models. Toluidine Blue (Fig. 1m) and Masson's trichrome staining (Fig. 1p) were employed to display the histological morphology of callus and newly formed tissue induced by MSFL. Co-staining of Itgav, CD90, CD105, and CD200 with Zfp260, Runx2, and Sox9 was conducted to characterize the clusters using specific marker combinations (Fig. 1n, q), to correlate staining signals with distinct cluster distributions (Fig. 1o, r). Statistical analysis indicated a significantly higher co-staining rate of Zfp260 with SSCs and BCSPs compared to Runx2 and Sox9, showing a notable increase from BCSP to THY + BLSP and a decrease from BCSP to PCP in the fracture model (Fig. 1s). In the context of the MSFL model, Zfp260 displayed an increasing co-staining rate across the SSC hierarchy, as illustrated in Fig. 1t. Overall, our findings indicated that Zfp260 demonstrated an early-stage upregulation of expression within the SSC lineage.

### Zfp260 gated the transition from SSC to BCSP in the injury repair

Previous research has demonstrated that periosteal skeletal stem cells (PSCs), a periosteal SSC population defined by the expression of Ctsk together as a primary biomarker along with additional cell surface markers located on the outer membrane of bone, were crucial for the healing of long bone fractures, facilitating both intramembranous and endochondral ossification[3]. In the MSFL model, Krt14$^+$Ctsk$^+$ osteoprogenitors were instrumental in transitioning into mature Krt14$^-$Ctsk$^+$ osteogenic cells and facilitating osteogenesis through an intramembranous-like ossification mechanism[17]. To identify its in vivo biological functions, Zfp260 was selectively ablated in Ctsk-expressing cells for further examination, administered by tamoxifen labeling in 7-week-old *Zfp260*$^{fl/fl}$ (control group) and *Ctsk*$^{CreERT2}$;*Zfp260*$^{fl/fl}$ (experimental group) mice (Fig. 2a). Analysis of the fracture model using micro-CT revealed periosteal reactions in the control group at 10 days post-fracture (dpf), while absent in the ablation group. In the control group, notable intramembranous ossification was observed at the bone callus margin at 14 dpf, with the area of endochondral ossification displaying increasing density from 14 to 28 dpf. In the experimental group, a significantly reduced level of mineralization was observed in the region of endochondral ossification, accompanied by an absence of significant indicators of intramembranous ossification between 14 and 28 dpf, as illustrated in Fig. 2b. Statistical analysis indicated a notable decrease in both bone mineral density (BMD) and absolute bone volume (BV) of calluses following Zfp260 ablation compared to the control group, as depicted in Fig. 2c. Regarding the overall prognosis of fracture healing, the knockout of Zfp260 resulted

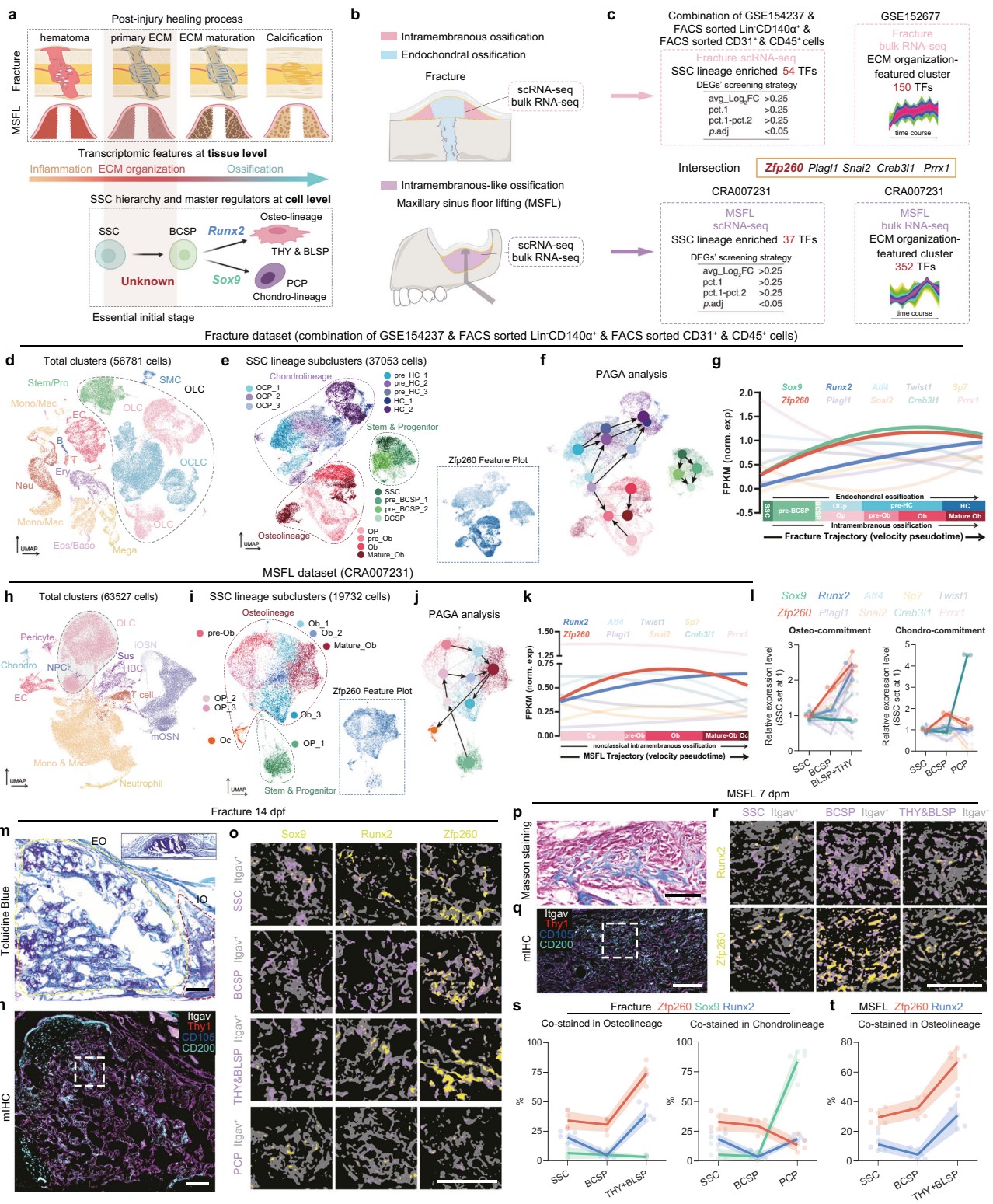

in a significant increase in the incidence of non-unionization (Fig. 2f). Toluidine blue staining revealed the formation of bone matrix at the edges of the bone callus through intramembranous ossification in the control group. On the contrary, the *Zfp260*-deficient group exhibited a prevalence of fibrous-like structures within the bone callus, devoid of typical osteoid at its peripheries at 10, 14 and 28 dpf. The process of endochondral ossification was characterized by the formation of cartilage matrix, its mineralization, and subsequent replacement with mature bone structures during the healing period. Nevertheless, the

absence of Zfp260 resulted in a reduction of cartilage matrix formation and mineralization, as depicted in Fig. 2g and h. The expression profiles of type I (Col-I) and type II (Col-II) collagen further illustrated that the absence of Zfp260 led to a notable decrease in Col-II expression within the region of endochondral ossification, which was associated with a pronounced delay in the formation and mineralization of the cartilage matrix. In the region of intramembranous ossification, there was also a significant reduction in Col-I expression, and the typical bone matrix-like structures were notably absent (Fig. 2i). Flow

**Fig. 1 | The expression pattern of candidate TF Zfp260 in both bone regenerating models. a** A schematic diagram depicting the healing processes of both models, including the levels of transcriptome, tissue, SSC hierarchy and respective master regulator, along with the regulatory factors guiding SSC-to-BCSP transition, remained elusive. **b** A schematic diagram of osteogenic mechanisms and workflows. **c** The screening strategy for candidate TFs, with Wilcoxon rank-sum test to identify DEGs. **d** Distribution of 56,781 cells from the fracture dataset, visualized by UMAP, with abbreviations listed in the "Methods" part. **e** Distribution of 37,053 re-clustered cells from the combination of Stem/Pro, OCLC and OLC from **d**. Fourteen subclusters were visualized by UMAP projection, with the feature plot of Zfp260. **f** The PAGA analysis of (**e**). **g** FPKM of classical chondrogenic- and osteogenic-related TFs and candidate TFs along the SSC hierarchy trajectory. **h** Distribution of 63,527 cells from the MSFL dataset, visualized by UMAP, with abbreviations listed in the Methods part. **i** Distribution of 19,732 re-clustered cells from the OLC of (**h**),

visualized by UMAP projection, with the feature plot of Zfp260. **j** The PAGA analysis of (**i**). **k** FPKM of TFs along the SSC hierarchy trajectory. **l** RT-qPCR of the FACS-sorted callus cells (14 dpf). 5 mice/biological repeat, *n* = 3 biological repeats. **m** Representative low-magnification images of Toluidine Blue-stained wild-type 14 dpf fracture callus, with the yellow dotted line indicating the EO region and the red one marking the IO region. **n** Representative mIHC image, with the white dotted box indicated the high-magnification region for (**o**). **o** Co-staining images of Sox9, Runx2 and Zfp260 with SSC hierarchy subclusters. **p** Representative image of Masson-stained wild-type 7 dpm MSFL. **q** Representative mIHC image, with the white dotted box indicated the high-magnification region for (**r**). **r** Co-staining images of Runx2 and Zfp260 with SSC hierarchy subclusters. For (**m–r**), *n* = 6 from 3 biological replicates. **s, t** Quantifications of the co-staining rates of (**o** and **r**). Scalebars: 50 μm. All data in this figure are represented as mean ± SD. Source data and exact *p* values are provided in the Source Data file.

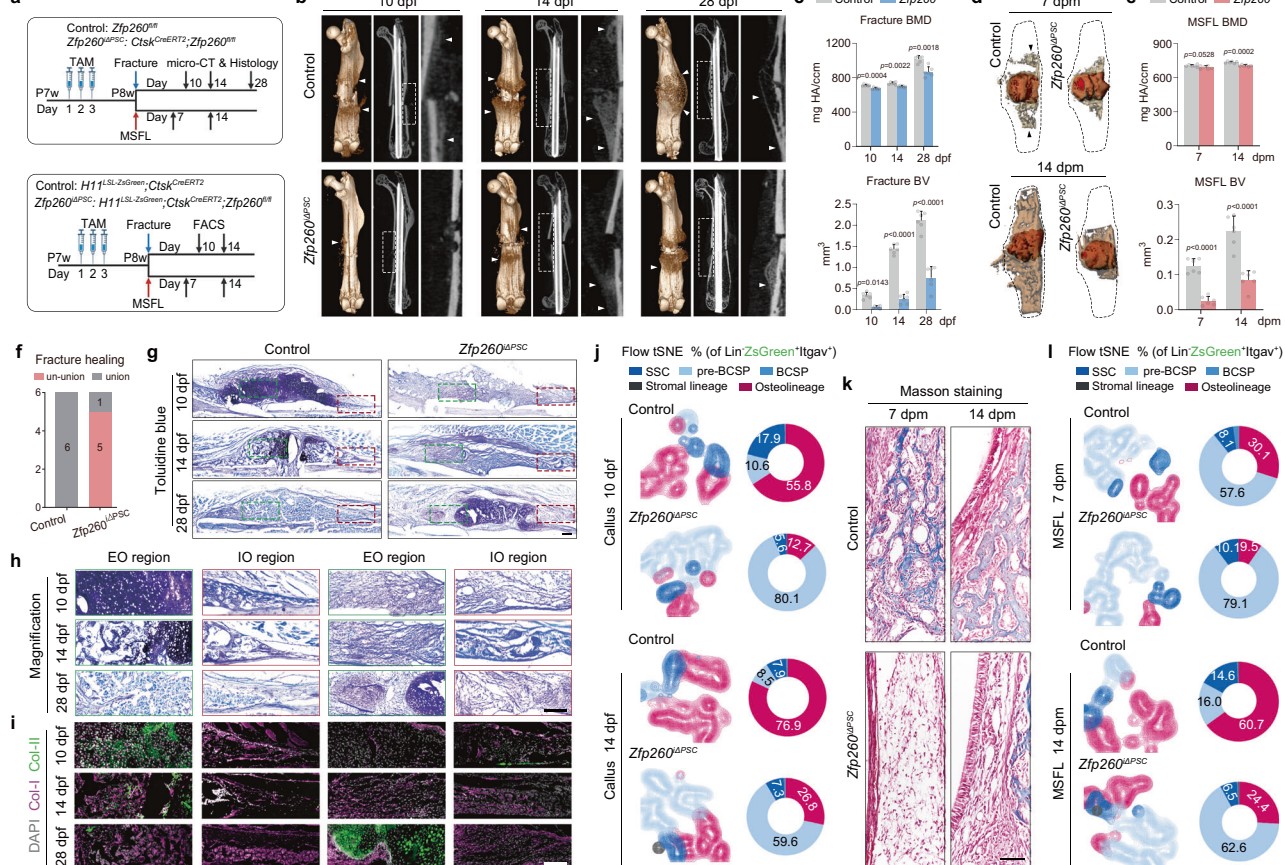

**Fig. 2 | Ablation of Zfp260 ablation in PSC impaired fracture- and MSFL-mediated bone regeneration. a** Workflow of TAM treatments, surgeries and subsequent analyses. **b** Representative Micro-CT reconstruction and X-ray images of fracture-induced healing at 10, 14, and 28 dpf. *n* = 6 from 6 biological replicates. **c** Statistical analysis of bone mineral density (BMD) and absolute bone volume (BV) of fracture calluses. *n* = 6 from 6 biological replicates. **d** Representative Micro-CT reconstruction images of MSFL-induced bone regeneration at 7 and 14 dpf. *n* = 6 from 6 biological replicates. **e** Statistical analysis of BMD and BV of MSFL. *n* = 6 from 6 biological replicates. **f** The fracture healing status (un-union or union). **g** Representative low-magnification images of Toluidine Blue-stained fracture calluses from control and *Zfp260^{iΔPSC}* mice at 10, 14 and 28 dpf. The red dotted line indicated the IO region, and the green one marked the EO region, with magnified

images shown in (**h**). **i** Representative images of co-staining with Col-I and Col-II in the magnified regions of EO and RO. For (**g–i**), *n* = 6 from 6 biological replicates. **j** FACS of SSC hierarchy from fracture calluses at 10 and 14 dpf, displayed by tSNE projection (left) and average percentage numbers (right). 5 mice per biological repeat, *n* = 2 biological replicates per time point. **k** Representative images of Masson-stained MSFL from control and *Zfp260^{iΔPSC}* mice at 7 and 14 dpm. *n* = 6 from 6 biological replicates. **l** FACS of SSC hierarchy from the regenerative tissues induced by MSFL at 7 and 14 dpm, displayed by tSNE projection (left) and percentage numbers (right). 15 mice per biological repeat, *n* = 1 biological replicate per time point. Two-way ANOVA. Scalebars: 50 μm. All data in this figure are represented as mean ± SD. Source data and exact *p* values are provided in the Source Data file.

cytometry analysis of Lin⁻ZsGreen⁺Itgav⁺ cells (Fig. 2a) demonstrated a significant increase in pre-BCSP clusters following Zfp260 knockout (Ctrl 10.6% vs Zfp260-cKO 80.1% at 10 dpf and 59.6% at 14 dpf), accompanied by a significant decrease in the downstream osteolineage cells at both 10 and 14 dpf (Fig. 2j).

Similarly, the MSFL model exhibited a significant inhibition of bone formation due to Zfp260 deficiency. Micro-CT reconstructed images displayed a notable decrease in newly formed bone tissues at 7 and 14 days post MSFL (dpm), consistent with the statistical analysis (Fig. 2d, e). Masson staining results showed that, despite minimal bone

matrix presence at 14 dpm in the Zfp260-deficient group, there was a notable hindrance in the overall osteogenic process, particularly in intramembranous ossification (Fig. 2k). Additionally, FACS analysis demonstrated an increase in pre-BCSPs and a decrease in the downstream osteolineage cells following Zfp260 knockout at both 7 and 14 dpm, similar to the findings in the fracture model. In conclusion, Zfp260 impeded both endochondral and intramembranous bone formation processes by obstructing the differentiation pathway from SSC to BCSP.

## Zfp260 initiated Runx2 expression through a super-enhancer-dependent mechanism

To investigate the regulatory mechanism underlying the inhibition of SSC-to-BCSP transition due to Zfp260 deletion, SSCs isolated from Ctsk labeled cells in fracture callus and MSFL models of both control and *Zfp260-cKO* mice were subjected to RNA sequencing as well as CUT&Tag for H3K4me1 and H3K27ac. Furthermore, ChIP-seq analysis of Zfp260-V5 in PSCs was conducted (Fig. 3a). Principal component analysis (PCA) of the RNA-seq data revealed distinct clustering patterns of cells in the Zfp260 ablation models compared to their respective controls (Fig. 3b). The results of GO and KEGG enrichment analyses of down-regulated DEGs following Zfp260 knockout demonstrated associations with extracellular matrix regulation, osteogenesis, and chondrogenesis (Fig. 3c), consistent with previous phenotypic observations. Decreases in H3K4me1 and H3K27ac levels were notably observed after Zfp260 knockout (Fig. 3d), with significant differences in super-enhancers between control and cKO groups associated with osteogenesis and extracellular matrix formation (Fig. 3e). ChIP-seq analysis of Zfp260-V5 revealed peaks predominantly located in distal intergenic regions, with associated genes involved in extracellular matrix formation, osteogenesis, and signaling pathways such as Rap1, Wnt, and Hippo (Fig. 3f–i). The intersection of differentially expressed genes (DEGs) identified from the aforementioned sequencing highlighted Runx2 as the sole gene of interest, confirming its crucial role in osteo-lineage commitment (Fig. 3j). TPM of RNA-seq identified significant down-regulated Runx2 expression level in both fracture and MSFL derived Lin-ZsGreen+ cells (Fig. 3k). Further examination of the Runx2 locus uncovered a 5.5 kb super-enhancer region upstream of the P1 promoter, which exhibited significant enrichment for Brd4 (GSE206155), H3K27ac, and Zfp260-V5 (Fig. 3l). Immunoprecipitation assays revealed the formation of a complex consisting of Zfp260-V5, Brd4, P300, and Med1 upon osteogenic induction (Fig. 3m). Nuclear co-localization of Zfp260, Brd4, Med1, and P300 was observed during osteogenic induction, with no significant overlap in signals during the resting state or after Zfp260 ablation as assessed by mIHC (Fig. 3n–p). ChIP-qPCR results showed a notable decrease in H3K27ac and Brd4 enrichment within the super-enhancer region following Zfp260 loss (Fig. 3q, r). Furthermore, our observations indicated that Zfp260-V5 bound to core downstream transcription factors of various osteogenic-related signaling pathways, such as β-Catenin of the Wnt signaling pathway, Smad5 of the TGF-β pathway, and Gli1 of the Hedgehog pathway, along with the super-enhancer regions of osteogenic-related transcription factors Zfhx4 and Tbx15, as well as the promoter regions of Atf4 and Mef2a (sFig. 5). In summary, Zfp260 played a crucial role in the assembly of super-enhancer complexes and in regulating several essential transcription factors related to osteogenesis, such as the pivotal regulator Runx2.

## Zfp260 was critical for the chromatin accessibility of BCSP

To investigate the potential regulatory relationship between Zfp260 dysfunction and Runx2 expression, as well as the disorder of SSC-to-BCSP transition (pre-BCSP arrest), we employed ATAC-seq to analyze chromatin accessibility features at various stages of SSC hierarchy. Subclusters from MSFL and fractures along the SSC hierarchy were identified in vivo and isolated for analysis (Fig. 4a). PCA of the ATAC-

seq revealed unique cell clustering patterns in the experimental models with Zfp260 ablation compared to their respective controls (Fig. 4b). The findings from GO and KEGG enrichment analyses revealed that genes associated with cell migration and proteoglycan synthesis were up-regulated as cells transitioned from SSC to pre-BCSP. Subsequently, during the progression from pre-BCSP to BCSP, the promoter regions of genes involved in extracellular matrix organization, osteoblast differentiation, TGF-β signaling, Wnt signaling, and Hippo signaling exhibited increased accessibility, indicating a heightened response to osteogenic lineage commitment under physiological conditions (Fig. 4c). These results underscore the significance of the transition from pre-BCSP to BCSP. Moreover, a notable decrease in ATAC signals was detected in BCSP following Zfp260 knockout, as depicted in Fig. 4d, e. Subsequent analysis indicated that Zfp260 knockout predominantly impacted the ATAC-seq signals associated with genes involved in multicellular organism development, extracellular matrix organization, and osteogenic differentiation in BCSP. This phenomenon was consistent across cells originating from both MSFL and fracture sources, as illustrated in Fig. 4f, g. Additional GO terms involved in the Zfp260 ablation were listed in Supplementary Data 3.

By integrating the findings from ChIP-seq analysis with the role of Zfp260 in facilitating the formation of super-enhancer complexes, we conducted a detailed examination to ascertain the direct regulatory impact of Zfp260 on the accessibility of critical enhancer regions associated with the transcription factors Runx2 and Zfhx4. Our observations revealed a notable reduction in ATAC signals within the Runx2 and Zfhx4 super-enhancer regions following Zfp260 knockout at the BCSP stage, a pivotal process in the commitment of progenitors into osteogenic lineages (Fig. 4h, i). The results of osteogenic induction experiments indicated that the absence of Zfp260 impeded osteogenesis during the early stages of the skeletal stem cell (SSC) hierarchy, specifically in SSC and pre-BCSP. However, this deficiency only diminished the osteogenic potential in the later stage of the SSC hierarchy, specifically in osteo-lineage cells (Fig. 4j). Moreover, utilizing alkaline phosphatase staining assay (ALP), colony-forming unit fibroblasts assay (CFU-Fs), and RT-qPCR, our study demonstrated that the knockout of Zfp260 weakened the early stage osteolineage commitment of SSCs (Fig. 4k), while their clonogenic potential remained largely unaffected (Fig. 4l). Furthermore, we performed subcapsular renal transplantation of wild-type and Zfp260-knockout SSCs and BCSPs, obtaining results similar to those observed with in vitro Alizarin Red staining. Zfp260 knockout at the SSC stage significantly impaired mineral nodule formation and led to cell arrest at the pre-BCSP stage (Fig. 4m). In contrast, Zfp260 knockout at the BCSP stage also reduced mineralization, but to a lesser extent than in the SSC stage (Fig. 4n).

In summary, Zfp260 maintained the opening of several critical osteogenic genes at the BCSP stage, such as Runx2. This regulating mechanism might contribute to the retention of Zfp260 at the pre-BCSP stage following ablation.

## Deficiency of Zfp260 in PSCs resulted in decreased Runx2 expression within the SSC hierarchy

Expanding upon the aforementioned in vitro investigation, our results suggested that Zfp260 played a crucial role in the Runx2 opening and transcription initiation. Subsequently, we conducted further research on the role of Zfp260 within the SSC hierarchy in vivo using the mIHC technique. Following the conversion of staining signals into SSC hierarchy locations (Fig. 5a, c, f, h), we quantified the co-staining rates of each subcluster with Runx2 (Fig. 5b, d, g, i). The findings, consistent with prior FACS research, demonstrated a significant abundance of pre-BCSPs, a limited quantity of osteolineage cells in the fracture callus, and areas of new bone formation in the MSFL model within the *Zfp260iΔPSC* group. Analysis of co-positivity rates, supported by

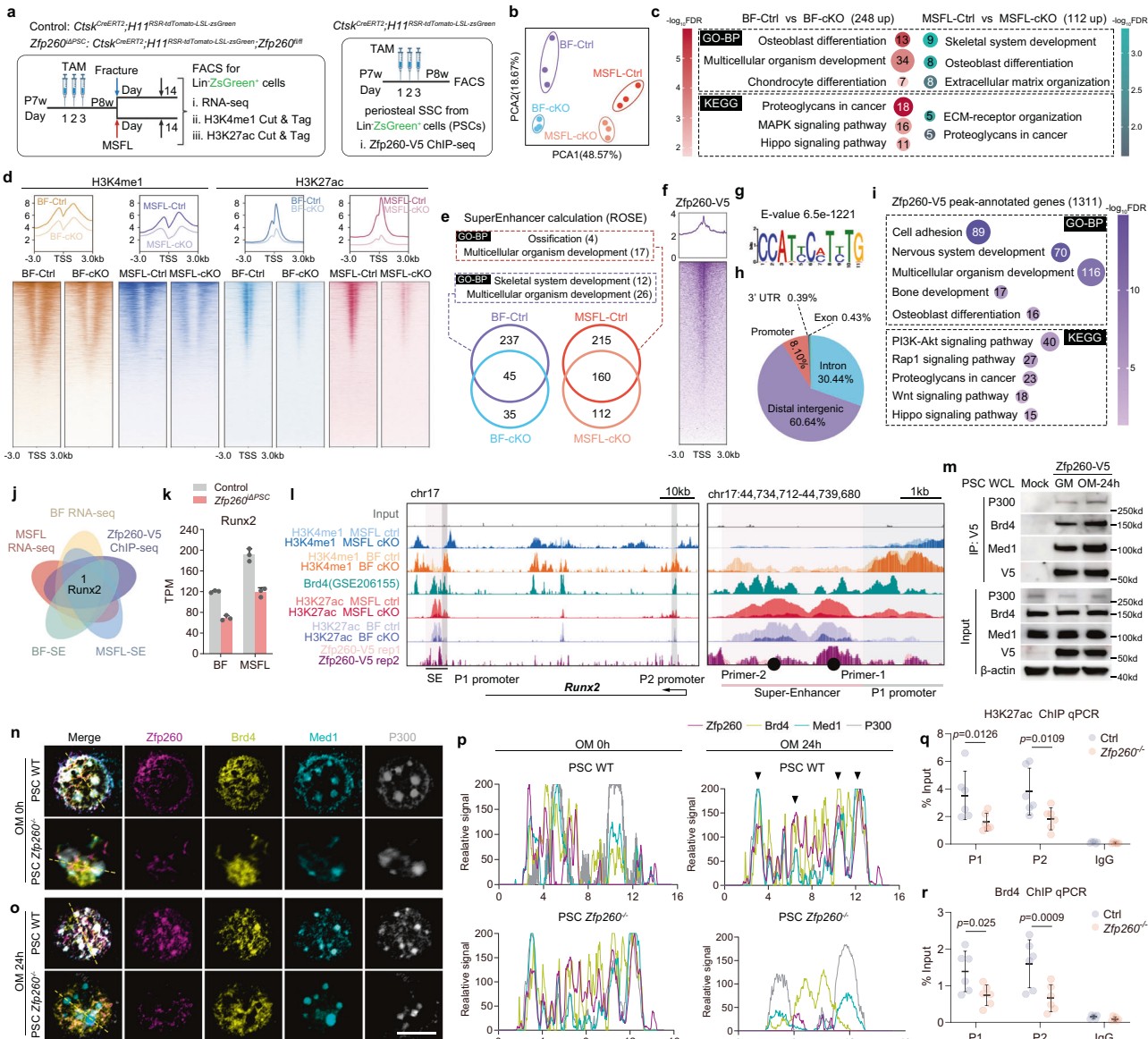

**Fig. 3 | Zfp260 regulated Runx2 expression in a super-enhancer-dependent manner. a** Workflow of in vivo labeling strategy using TAM and the experimental design. **b** PCA indicating the variations of transcriptomes among Lin⁻ZsGreen⁺ cells isolated from BF-Ctrl, BF-cKO, MSFL-Ctrl and MSFL-cKO groups. **c** GO and KEGG enrichment analysis of FACS-RNA-seq data. **d** Heatmap of replicate data for H3K4me1 and H3K27ac enrichment as detected by CUT&Tag. *n* = 3 from 3 biological replicates. **e** GO-biological process enrichment analysis of differentially enriched super-enhancers. **f** Heatmap of replicate data for Zfp260-V5 enrichment detected by ChIP-seq. *n* = 2 from 2 biological replicates. **g** Top enriched de novo motifs of Zfp260-V5 enriched genes. **h** Distribution of peaks in the genome. **i** GO and KEGG enrichment analysis of Zfp260-V5 enriched genes. **j** Screening strategy for the potential master downstream regulator. **k** Transcripts Per Kilobase (TPM) of Runx2 expression level from fracture and MSFL derived Lin⁻ZsGreen⁺ cells. *n* = 3 from 3 biological replicates of RNA-seq data. **l** Genome browser view of peaks

enriched for H3K4me1, Brd4, H3K27ac, and Zfp260-V5 over the Runx2 gene locus on chromosome 17 (left) with the magnified super-enhancer region displayed on the right. Primers 1 and 2 indicated the primer sets for the subsequent ChIP-qPCR detection. **m** Co-IP was performed to examine the condensates for the super-enhancer via immortalized PSCs. *n* = 3 from 3 biological replicates. **n, o** mIHC co-staining for Zfp260 (purple) with Brd4 (gold), Med1 (cyan), and P300 (gray) in the homeostatic and osteogenic states of PSCs. The yellow dotted line indicated the route for the subsequent fluorescence intensity measurements. *n* = 3 from 3 biological replicates. **p** Fluorescence intensity measurements along the route, with black triangles indicating the merged signals of the four channels. **q, r** ChIP-qPCR assays for H3K27ac and Brd4 binding via immortalized PSCs. *n* = 6 from 2 biological replicates. Two-way ANOVA. Scalebars: 5 μm. All data in this figure are represented as mean ± SD. Source data and exact *p* values are provided in the Source Data file.

statistical data, indicated that *Zfp260* knockout resulted in variable down-regulation of Runx2 signals in stem/progenitor and osteolineage cells in both experimental models. In stem/progenitor cells, the knockout of *Zfp260* resulted in a significant alteration of Runx2 levels in SSC, pre-BCSP, and BCSP populations within the MSFL model, with minimal impact observed in the fracture model. In osteolineage cells, the knockout of Zfp260 resulted in a significant reduction in the proportion of Runx2-positive cells in both models (Fig. 5e, j).

## Phosphorylation of Zfp260 was critical for its downstream function

Modifications of transcription factors frequently served as a regulatory mechanism for their translocation and function. Through GST pull-down and mass spectrometry analyses (Fig. 6a), we identified the enzymes interacting with Zfp260. Among these, Prkca was found to be the most enriched, as detailed in Fig. 6b. Co-immunoprecipitation assays in the 293 T cell line demonstrated that Zfp260 interacts with

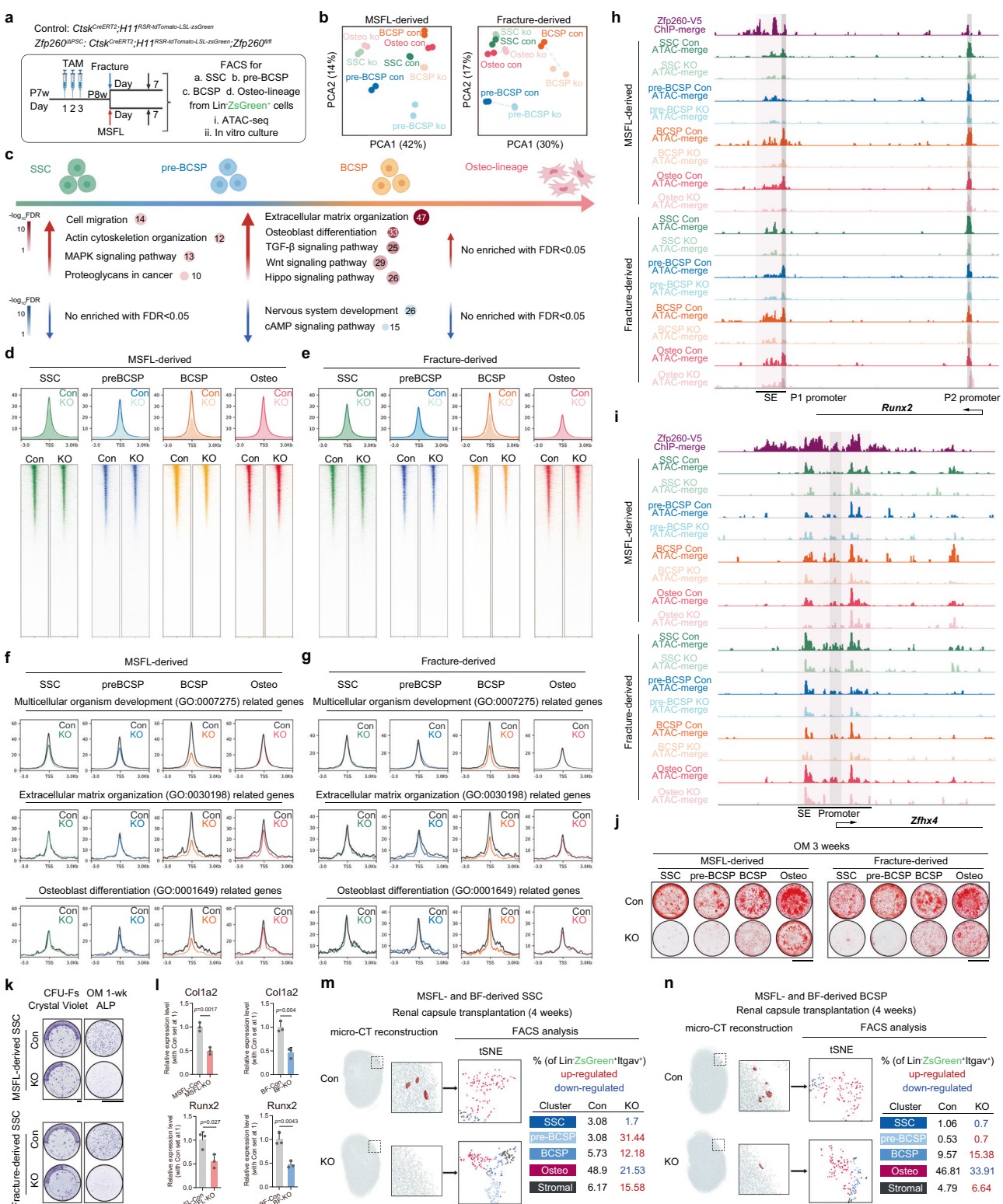

Prkca (Fig. 6c), a finding that was also confirmed in PSC (Fig. 6d). Immunofluorescence analysis further revealed co-localization signals of Zfp260 and Prkca in the cytoplasm (Fig. 6e). Following the validation of the interaction between Zfp260 and Prkca, the question of whether Prkca could regulate the phosphorylation level of Zfp260 remained unanswered. The use of a Prkca inhibitor resulted in a decrease in the nuclear translocation capacity of Zfp260 (Fig. 6f, g). Both In vitro Phos-PAGE and In-cell pull-down assays demonstrated the presence of phosphorylated Zfp260, which was regulated by the activity of Prkca.

M/S analysis identified the phosphorylation sites Y173, S182, and S197 (Fig. 6h, i). The AF2-mediated structural prediction analysis revealed that residues Y173, S182, and S197 of Zfp260 were in close proximity to the catalytic domain of Prkca, as depicted in Fig. 6j. Utilizing this information, simulations were conducted to predict the effects of seven loss-of-function mutations: three single-point mutations (Y173F, S182G, and S197G), three double-point mutations (Y173F-S182G, Y173F-S197G, and S182G-S197G), and one triple-point mutation (Y173F-S182G-S197G). The docking analysis of seven mutants of Zfp260 with Prkca revealed

**Fig. 4 | Zfp260 regulated chromatin accessibility of BSCP. a** Workflow of in vivo labeling and FACS strategy for ATAC-seq. **b** PCA indicating variations in chromatin accessibility among isolated subclusters (SSC, pre-BCSP, BCSP and Osteo-lineage cells) from Lin⁻ZsGreen⁺ cells. **c** GO and KEGG enrichment analysis of the differentially accessible chromatin regions during maturation and lineage commitment of SSC lineage cells. **d, e** Profiles and heatmaps showing enrichment of ATAC-seq signals in MSFL and fracture-derived SSC, pre-BCSP, BCSP, and Osteo-lineage cells from both control and knockout groups. $n = 2$ from 2 biological replicates. **f, g** Profiles showing enrichment of ATAC-seq signals in genes related to multicellular organism development (GO:0007275), extracellular matrix organization (GO:0030198), and osteoblast differentiation (GO:0001649). **h, i** Genome browser view showing Zfp260-V5 binding and ATAC-seq signals of Runx2 and Zfhx4 in MSFL and fracture-derived SSC, pre-BCSP, BCSP, and Osteo-lineage cells from both

control and knockout groups. $n = 2$ from 2 biological replicates. **j** Representative ARS staining images displaying the osteogenic capabilities of MSFL or fracture-derived SSC, pre-BCSP, BCSP and Osteo-lineage cells from both control and knockout groups. $n = 3$ from 3 biological replicates. **k** Representative CFU-Fs assay and ALP staining images of MSFL or fracture-derived SSCs from both control and knockout groups. $n = 3$ from 3 biological replicates. **l** RT-qPCR of MSFL or fracture-derived SSCs, $n = 3$ biological replicates. Two-tailed unpaired $t$ test. Representative images of subcapsular renal transplantation of MSFL and fracture derived SSC (**m**) and BCSP (**n**) from both control and knockout groups by micro-CT reconstruction, followed by FACS analysis, $n = 3$ biological replicates. Scalebars: 5 mm. All data in this figure are represented as mean ± SD. Source data and exact $p$ values are provided in the Source Data file.

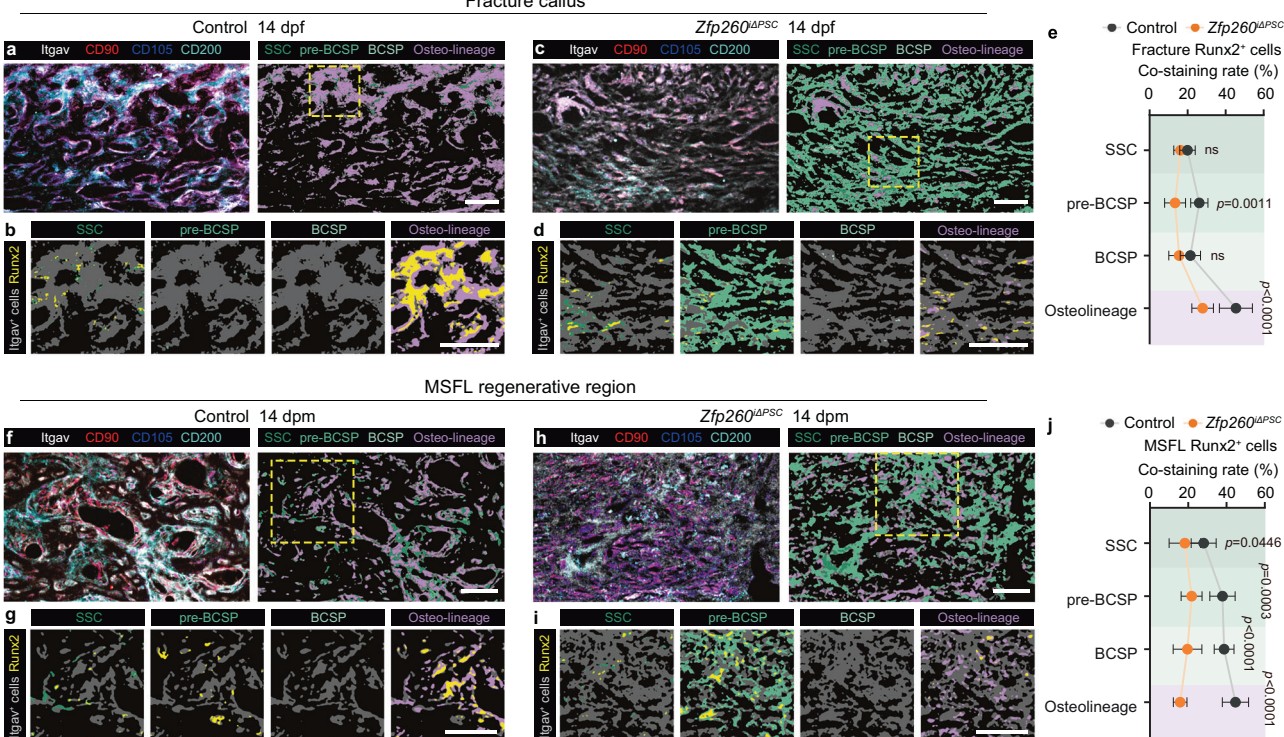

**Fig. 5 | Zfp260 ablation in vivo impaired Runx2 expression levels in SSC hierarchy subclusters. a** Representative mIHC images of fracture callus at 14 dpf from the conrtol group, labeling Itgav, Thy1, CD105 and CD200 (left). The SSC hierarchy subclusters derived from the combinations of immuno-signals (right). The yellow dotted box indicated the high-magnification region for (**b**). **b** Co-staining images of Runx2 with SSC, pre-BCSP, BCSP, and Osteo-lineage (combination of PCP, THY, and BLSP). **c, d** Representative images of fracture callus at 14 dpf from the *Zfp260^{iΔPSC}* group, with display strategy similar to (**a, b**). **e** Quantification of the co-staining

rates indicated in (**a–d**). **f, g** Representative images of MSFL-induced regenerative tissues at 14 dpm from the control group, with display strategy similar to (**a, b**). **h, i** Representative images of MSFL-induced regenerative tissues at 14 dpm from *Zfp260^{iΔPSC}* group, with display strategy similar to (**a, b**). **j** Quantification of the co-staining rates indicated in (**f–i**). For this figure, $n = 6$ from 3 biological replicates. Two-way ANOVA. Scalebars: 50 μm. Source data and exact $p$ values are provided in the Source Data file.

spatial separation of Zfp260 residues from Prkca's catalytic domain (sFig. 6), as supported by statistical analysis (Fig. 6k). Subsequent overexpression of these mutants, along with wild type Zfp260-V5, in *Zfp260^{-/-}* PSCs followed by immunoprecipitation assays demonstrated a significant decrease in binding capacity with Prkca for each mutated form of Zfp260, highlighting the substantial impact on binding stability (Fig. 6l). Subsequent rescue experiments were conducted by overexpressing the empty vector (Mock), wild-type (Zfp260-V5), and mutant forms of Zfp260 in *Zfp260^{-/-}* PSCs in order to investigate the potential of phosphorylation-deficient forms of Zfp260 to reverse the phenotypic changes induced by Zfp260 ablation. The findings revealed that all mutant forms were incapable of restoring the diminished nuclear translocation capabilities (Fig. 6m, n), decreased formation of mineralized nodules (Fig. 6o), and reduced enrichment of Zfp260-V5, Brd4,

and H3K27ac in the Runx2 super-enhancer region (Fig. 6p-) resulting from Zfp260 dysfunction.

## Zfp260 phosphorylation sites were required for both intra-membranous and endochondral ossification processes in vivo

In order to investigate the in vivo function of each Zfp260 mutant, an expression vector containing a Ctsk promoter was employed for the packaging of rAAV9. Analysis of Ctsk expression levels at the single-cell level revealed significant expression in mesenchymal cells (sFig. 7b). Subsequent infection with rAAV9 containing the Ctsk promoter demonstrated notable infectivity in craniofacial, mandibular, and long bones (sFig. 7a). Utilizing this system, AAV-Zfp260-WT and its respective mutants were generated. In line with the experimental designation, intravenous administration of AAVs was conducted to

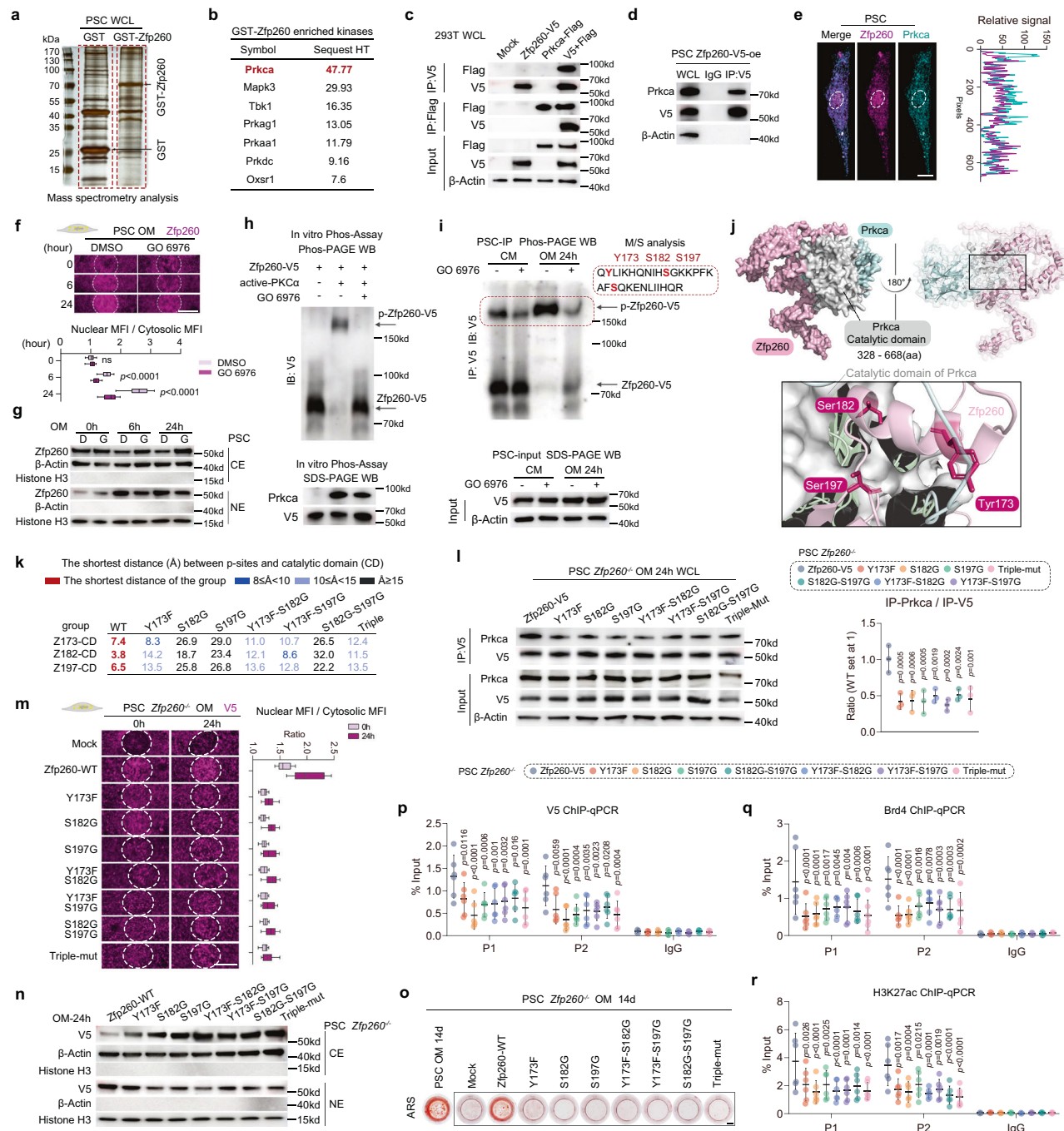

**Fig. 6 | Phosphorylation of Zfp260 was crucial for its biological functions.**
**a** GST-pull down assay of PSC's whole cell lysate (WCL). The red dotted box indicated the regions for M/S analysis. **b** GST-Zfp260 specially enriched kinases with high HT sequest scores. **c** Co-IP of 293 T cell line. **d** Co-IP of immortalized PSCs. **e** Co-staining image for Zfp260 and Prkca in PSCs. The white dotted circle indicated the nucleus (Left). The white dotted line indicated the route for the fluorescence intensity measurements (right). **f** Representative images of immunofluorescence of Zfp260 with osteogenic induction, with the MFI/cytosolic MFI calculated (right). **g** Separation of nuclear (NE) and cytosolic extracts (CE) followed by Western blot. **h** In vitro Phos-Assay was performed by Phospho-PAGE. **i** Co-IP, Phos-PAGE, and SDS-PAGE were jointly performed in immortalized PSCs. The red dotted rectangle indicating the Y173, S182, and S197 residues. **j** Suggested binding mode of Zfp260 and Prkca by AlphaFold2. **k** The shortest distance between Zfp260-aa173 and the catalytic domain (CD) of Prkca (Z173-CD), aa182 and CD (Z182-CD), and aa197 and

CD (Z197-CD) were calculated. **l** Co-IP was performed in immortalized PSCs, with statistical analysis (right). **m** Representative images of immunofluorescence of Zfp260-V5 before and after osteogenic induction (left), with the nuclear MFI/cytosolic MFI evaluated (right). **n** Separation of NE and CE, followed by Western blot. **o** Representative ARS staining images of PSCs. Scalebar: 2 mm. **p**–**r** ChIP-qPCR assays for Zfp260-V5, Brd4, and H3K27ac binding. $n = 6$ from 2 biological replicates. For (**a**, **d**, **e**, **h**, **i**), experiments were conducted independently 3 times, consistently producing similar results. For (**c**, **g**, **h**, **i**, **l**, **n**, **o**), $n = 3$ from 3 biological replicates. For (**f**, **m**), $n = 30$ from 3 biological replicates, with 10 randomly selected cells calculated per replicate. For (**e**, **f**, **m**), scale bars: 5 μm. Two-way ANOVA. Box plots display the minimum and maximum values, with the center line representing the median, and the bounds of the box representing the 25th to 75th percentiles. Other data in this figure are represented as mean ± SD. Source data and exact $p$ values are provided in the Source Data file.

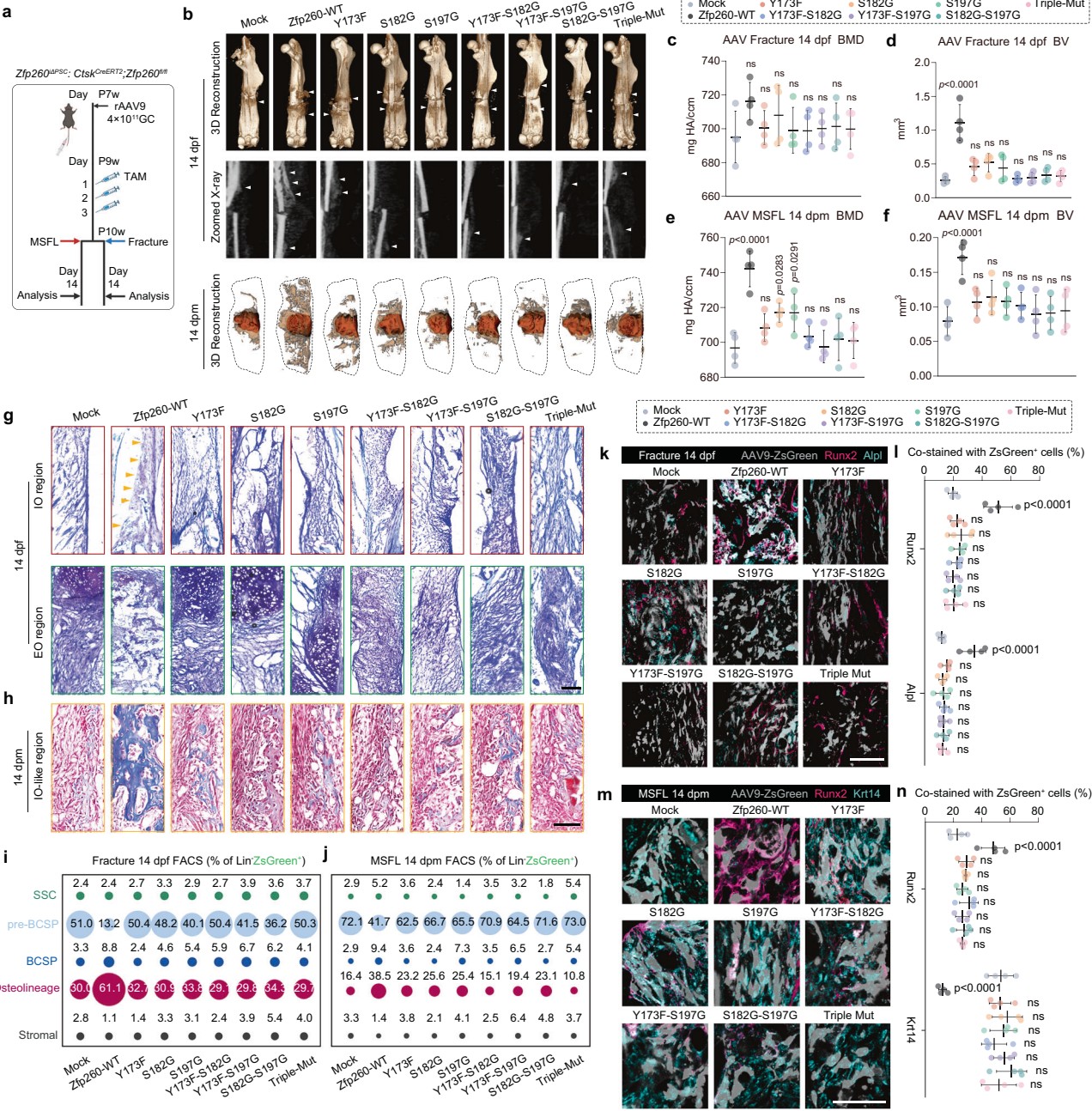

**Fig. 7 | AAV9-Zfp260 mutants failed to rescue the impaired osteogenesis in vivo with Zfp260 ablation. a** Workflow of AAV9 administrations, TAM treatments, surgeries, and subsequent analyses. **b** Representative Micro-CT reconstruction and X-ray images of fracture-induced healing and MSFL-induced bone regeneration at 14 dpf. $n = 4$ from 4 biological replicates. **c–f** Statistical analyses of bone mineral density (BMD) and absolute bone volume (BV) of fracture callus or MSFL with the use of rAAV9. $n = 4$ from 4 biological replicates. **g** Representative magnified images of Toluidine Blue-stained fracture calluses in the IO region (outlined in red) and EO region (outlined in green) at 14 dpf. $n = 4$ from 4 biological replicates. Yellow tri-angles indicated the calcified bone matrix. **h** Representative magnified images of Masson-stained MSFL-induced regenerative tissues at 14 dpf. $n = 4$ from 4 biological replicates. **i** FACS of SSC hierarchy from fracture calluses of 14 dpf. 5 mice per biological repeat, $n = 2$ biological replicates. **j** FACS of SSC hierarchy from the

regenerative tissues induced by MSFL at 14 dpm. 15 mice per biological repeat, $n = 1$ biological replicate. **k** Representative mIHC images of fracture callus at 14 dpf, with the administration of rAAV9 carrying wild-type or mutant forms of Zfp260. Merged channel images of Runx2 (magenta) and Alpl (cyan) were shown. $n = 4$ from 4 biological replicates. **l** Quantification of co-staining rate with ZsGreen+ cells in (**k**). **m** Representative mIHC images of MSFL-induced regenerative tissues at 14 dpm, with the administration of rAAV9 carrying wild-type or mutant forms of Zfp260. Merged channel images of Runx2 (magenta) and Krt14 (cyan) were shown. $n = 4$ from 4 biological replicates. **n** Quantification of co-staining rate with ZsGreen+ cells in (**m**). One-way ANOVA. Scalebars: 50 μm (**g**, **h**), 25 μm (**k**, **m**). All data in this figure are represented as mean ± SD. Source data and exact $p$ values are provided in the Source Data file.

infect 7-week-old $Ctsk^{CreER}$; $Zfp260^{fl/fl}$ mice. Tamoxifen was administered two weeks later, followed by modeling at ten weeks (Fig. 7a). Micro-CT analysis demonstrated that in the fracture model, the AAV-Zfp260-WT group exhibited significantly higher density in the IO region, with both bone mineral density (BMD) and absolute ossification volume (BV)

significantly greater than those in the Mock group. No significant improvements in BMD and BV were observed in the groups treated with any AAV-Zfp260 mutant.

A comparable trend was noted in the MSFL, as depicted in Fig. 7b–f. Histologically, enlarged images of the IO and EO regions were

provided, with their corresponding original images in sFig. 7c, d. Within the fracture model, a mineralized bone matrix was evident in the IO region of the AAV-Zfp260-WT group, alongside the cartilage matrix and replacement bone matrix in areas of endochondral ossification. Conversely, minimal bone matrix formation was observed in the cohorts treated with each AAV9-Zfp260 mutant.

Significantly, the presence of cartilage matrix was evident in the regions of endochondral ossification. However, a lack of calcified matrix was also noted (Fig. 7g). Within the MSFL model, notable bone matrix formation was observed in the AAV-Zfp260-WT group. Conversely, the AAV9-Zfp260 mutant groups displayed hindered bone matrix formation, failing to demonstrate substantial enhancement compared to the Mock group (Fig. 7h). FACS demonstrated that the introduction of Zfp260-WT in both fracture and MSFL models resulted in a decrease in pre-BCSP accumulation and a significant increase in the osteolineage cells, consistent with histological observations (Fig. 7i, j). Furthermore, co-staining with markers Krt14, Runx2, and Alpl allowed for evaluating osteogenic differentiation and maturation stages of AAV9-infected (ZsGreen+) cells in vivo by measuring their co-positivity rates with these markers. In the fracture model, ZsGreen+ cells in the Zfp260-WT group demonstrated significantly higher co-positivity rates with Runx2 and Alpl compared to those in the Mock group. In contrast, the mutant groups did not show a significant increase (Fig. 7k, l). Analysis of the MSFL model, which incorporated Krt14 to assess osteoprogenitor cell differentiation maturity, in accordance with previous studies, indicated that, except for the Zfp260-WT group, the remaining mutant groups displayed elevated co-positivity rates with Krt14 and decreased levels of Runx2, mirroring in vitro observations (Fig. 7m, n). In conclusion, the introduction of Zfp260 mutants did not ameliorate the impaired bone healing phenotype associated with Zfp260 deficiency in PSCs.

## Discussion

Transcription factors (TFs) have been shown to play a significant role in regulating bone repair and regeneration[18]. Previous research has identified key TFs that control the differentiation of BCSP into osteogenic and chondrogenic lineages[19,20]. However, the specific mechanisms governing the differentiation of SSC into BCSP have not been fully elucidated. Through screening and identifying TFs involved in the early stages of long bone fracture repair and craniofacial bone regeneration in mouse MSFL models, we have identified Zfp260 as a critical regulator of SSC differentiation into BCSP during bone repair and regeneration. Deletion of Zfp260 in Ctsk-labeled SSCs resulted in compromised fracture healing and MSFL bone regeneration, characterized by an accumulation of pre-BCSP (defined as phenotypic SSC). The absence of Zfp260 during the transition from SSC to BCSP hindered the formation of mineralized nodules. Comprehensive multi-omics analyses revealed that Zfp260 modulated the activation of key transcription factors (TFs) such as Runx2, Zfhx4, and Tbx15 by forming a super-enhancer complex in conjunction with Med1 and Brd4. The knockout of Zfp260 resulted in a significant decrease in the modifications of H3K4me1 and H3K27ac on the cis-regulatory elements of osteogenesis-related genes in the SSC lineage, as well as a notable reduction in the ATAC accessibility signal of Runx2 at the BCSP stage. Furthermore, our study identified PKCα as the primary kinase responsible for mediating Zfp260 phosphorylation, targeting residues at Y173, S182, and S197. Subsequent in vivo and in vitro rescue experiments confirmed the functional significance of these phosphorylation sites on Zfp260. These results indicated that Zfp260 played a crucial role in regulating the transition of SSC to BCSP, thereby connecting this conversion process with the determination of the osteogenic differentiation fate of BCSPs during the stem cell response to bone injury repair (Fig. 8).

The meticulous regulation of the initial phases of bone repair and regeneration was essential for influencing the long-term outcomes of bone healing[9]. This process encompassed the early fate decisions of SSCs and the establishment of the initial extracellular matrix. Chan et al.[1] performed a comprehensive analysis of surface marker combinations and, utilizing a series of FACS-RNA sequencing and renal subcapsular transplantation experiments, delineated the differentiation stages of the skeletal stem cell (SSC) lineage, endeavoring to map their differentiation sequence. Chan et al. illustrated a progression from skeletal stem cells (SSCs) to bone, cartilage, and stromal progenitors (BCSPs), ultimately leading to the differentiation into osteolineage/osteochondro-lineage or stromal lineage, as evidenced by extensive post-sorting in vitro data. Although the unidirectional cascade differentiation sequence had been established and demonstrated through in vitro methodologies, there remained insufficient robust evidence to confirm its stable existence in vivo. Specifically, the extent to which this differentiation sequence downstream of SSCs adhered to the same strict unidirectional cascade in vivo had not been thoroughly investigated. Besides, the early transition from SSC to BCSP was particularly challenging to study due to the close similarities in functional differentiation and transcriptional profiles, the complexity of marker combinations, and the absence of suitable in vitro cell lines. Our findings suggested that this transition involved a notable increase in ATAC signals for crucial genes related to osteogenesis and extracellular matrix formation genes, highlighting distinct differences between stem and progenitor cells.

This study examined the temporal expression patterns of TFs at both tissue and cellular levels during bone repair, illustrating the functional sequence of TFs under physiological conditions. Our screening revealed that Zfp260 was the TF significantly upregulated during the transition from SSC to BCSP among all candidates, whereas several established osteogenic TFs, including Runx2, ATF4, Sp7, and Zfhx4, were exclusively expressed at the BCSP osteogenic fate determination stage. The deficiency of these transcription factors resulted in varying effects on matrix mineralization, without significantly influencing matrix formation, indicating that the transition from SSC to BCSP might not be entirely rely on these factors[21–23]. The ordered expression patterns highlighted a potential regulatory association between Zfp260 and these established osteogenic transcription factors, with the utilization of multi-omics methodologies offering support for this correlation. ChIP-seq analysis and the the measurements of modifications in H3K4me1 and H3K27ac following Zfp260 knockout suggested a pivotal role for Zfp260 in regulating super-enhancer complex formation[24]. Additionally, the notable decrease in ATAC signal accessibility for critical genes like Runx2 and Zfhx4 during the BCSP stage following Zfp260 ablation provided further evidence of Zfp260's involvement in facilitating the activation of crucial target gene signals, akin to super-enhancer complexes[25]. Through the integration of the temporal features of Zfp260 upregulation during the SSC-BCSP stage, the activation of ATAC signals for master osteogenic regulators during the SSC-BCSP stage, and their subsequent expression during the BCSP-osteolineage stage, a causal relationship was elucidated between Zfp260 and the transition of SSCs to BCSPs, as well as the determination of fate towards osteogenic differentiation in BCSPs.

Our investigation into the impact of Zfp260 on the osteogenic regulation of SSC lineage cells revealed that Zfp260 was essential for mineralization induction in SSCs and BCSPs in vitro. However, Zfp260 loss-of-function in osteolineage cells only led to a decrease in mineralized nodules. This reduction might be attributed to the decreased expression of Zfp260 in mature osteoblasts and osteocytes, as well as the compensatory mechanisms of the osteogenic TF network during this stage[26–28]. In vivo, Zfp260 was knocked out using Ctsk promoter-driven inducible Cre recombinase knock-in mice. This led to notable hindrances in the process of intramembranous ossification and the disruption of endochondral mineralization. At the cellular level, there was an accumulation of pre-BCSPs and a depletion of both upstream

**Fig. 8 | A schematic diagram depicting the discovery and role of Zfp260 in osteogenesis.** The upper left part of the diagram illustrated the physiological differentiation process of the SSC hierarchy in both endochondral and intramembranous ossifications, highlighting the accessible enhancer level and expression levels of Zfp260 and Runx2 during this process. The upper right part showed the accumulation of pre-BCSP due to Zfp260 dysfunction, resulting from the downregulated accessible enhancer level and expression level of Runx2. The lower part depicted the intracellular functions of Zfp260 and the consequences of its ablation.

and downstream cells. Multiplex IHC analysis revealed a significant reduction in Runx2 co-positivity among downstream SSC lineage cells, consistent with in vitro Zfp260 intervention results. Due to constraints in the in vivo labeling technique, Ctsk-labeled cells within the periosteum of long bones encompassed SSCs, osteoprogenitors, and stromal cells[3]. Similarly, the Schneiderian membrane contained osteoprogenitors co-expressing Krt14 and Ctsk or expressing Ctsk alone[4]. Consequently, a minor presence of bone tissue was still evident in the regions undergoing intramembranous ossification in both models, potentially attributable to the limited bone regenerative capacity of the local osteolineage cells.

Notably, the deletion of Zfp260 in Ctsk-expressing cells also exerted a significant reduction on the formation and mineralization of the cartilage matrix within the callus. According to the findings of Debnath et al., Ctsk lineage cells played a regenerative role via both intramembranous and endochondral ossification. Therefore, it was hypothesized that the phenotypic differences observed in the endochondral ossification region within the fracture model may be attributed to the loss of Zfp260 function in the subset of Ctsk lineage cells responsible for endochondral ossification. To focus on the the the influence of Zfp260 on the endochondral ossification process, applications of transgenic mice with Col2a or Sox9 promoter-driven Cre recombinase for the knockout of Zfp260 should be carried out[29,30].

Phosphorylation serves as a vital post-translational modification for the activity of numerous transcription factors (TFs)[31,32]. Through mass spectrometry and in vitro phosphorylation intervention, we identified the principal functioning kinase PKCα and critical phosphorylation sites (Y173, S182, S197) of Zfp260. Subsequent rescue experiments via Zfp260 mutants conducted both in vivo and in vitro have provided substantiating evidence of their functional importance. Notably, in this in vivo experiment, Ctsk-driven AAV9 was utilized to target Ctsk-expressing cells that were knocked out by Zfp260 and achieved high bone targeting efficiency. This technique enhanced the credibility of the results of in vivo rescue experiment.

In conclusion, Zfp260 was identified as a crucial TF regulator during the transition from skeletal stem cells (SSC) to bone/cartilage stem progenitor (BCSP) cells, underscoring the importance of this transition and connecting the osteolineage commitment of the following step. In addition, the identification of phosphorylation sites of Zfp260 in this study offered valuable references for potential drug development. Current personalized treatment approaches for impaired bone healing highlighted the importance of augmenting endogenous regenerative mechanisms, implementing early targeted detection methods, and mitigating healing complications. The targeting of Zfp260 as an initial functional target for potential intervention strategies presented significant promise for the prompt management of bone-related disorders.

# Methods

## Animal models

*Igs2*[em1(CAG-LSL-ZsGreen-wpre-pA-CAG-RSR-tdTomato-wpre-pA)Smoc] (*H11*[LSL-ZsGreen-RSR-tdTomato]) (NM-KI-200319), *Ctsk*[em1(2A-CreERT2-WPRE-polyA)Smoc] (*Ctsk*[creER]) (NM-KI-200067) with C57BL/6 J background were purchased from the Shanghai Model Organisms Center, China. *C57BL/6J-Zfp260*[em1Cflox/Cya] (*Zfp260*[flox/flox]) (S-CKO-09613) was obtained from Cyagen, USA, with its construction strategy displayed in sFig. 8. All mice were used for analysis regardless of sex. All mice were housed at the Laboratory Animal Center of Tongji University with constant temperature (22 ± 2°C) and humidity (55 ± 10%) in a 12-h light cycle. All animal studies were approved by the Institutional Animal Care and Use Committee of Tongji University.

## In vivo modeling and tamoxifen induction

All mice were used for analysis regardless of sex. For descriptive and functional studies, 7wk-old mice were randomly assigned into different experimental groups. For the conditional genetical deletion of Zfp260 or labeling of Ctsk+ periosteal stromal cells, *Zfp260*[fl/fl], *Ctsk*[creER]*Zfp260*[fl/fl], *Ctsk*[creER]*H11*[LSL-ZsGreen-RSR-tdTomato], *Ctsk*[creER]*H11*[LSL-ZsGreen-RSR-tdTomato]*Zfp260*[fl/fl] mice were inraperitoneally administered with Tamoxifen (50 mg/kg) (T5648, Merck) for 3 consecutive days at 7wk-old in accordance with the experimental design. Mice were modeling at 8wk-old under the anesthesia of 2% isoflurane, followed by sterilization. For MSFL, the surgery was performed based on the following processes[4]. After making the median section, the 1/2 tungsten steel round bur was applied to drill holes in the center of the nasal bone without breaking the nasal membrane. A mini implant (WEGO) with diameter of 0.6 mm and length of 0.8 mm was implanted, covered by a 5 mm × 5 mm*0.1 mm Teflon film. For fracture, a full fracture was created at the mid-shaft and fixed by the insertion of a sterilized 22-gauge needle into the medullary cavity[33].

## Bulk RNA sequencing and analysis

For the RNA sequencing of FACS-sorted cells (Lin-ZsGreen+ cells of Ctrl or Zfp260-cKO groups derived from the fracture callus or the newly formed tissues in MSFL), total RNA was extracted with their integrity numbers >7.0, measured by Agilent 2100 Bioanalyzer. Similar to our previous study, the library was constructed in accordance with the instructions of the low input Trio RNA-Seq™ library preparation Kit (0357-32, Nugen), with the insert length varied around 350 bp. Novaseq 6000 was used for high throughput sequencing with the strategy of PE150. To acquire the transcriptomic features of the full fracture healing process, the dataset GSE152677 (ctrl, 1 d, 3 d, 5 d, 7 d, 14 d) was applied and further analyzed in this study[34]. As for the MSFL healing process, the laser microdissection (LCM) based bulk-seq dataset CRA007231 applied in our previous research was re-analyzed in this study[4]. For data processing, the raw fastq files were trimmed and aligned using Hisat (https://www.psc.edu/resources/software/hisat-2) with the mm10 genome and nomarlized to TPM[35]. Mfuzz (http://mfuzz.sysbiolab.eu) was used for time course analysis[36]. Cytoscape 3.7.1 (https://cytoscape.org/index.html) was used to depict the protein–protein interaction network. Principle component analysis (PCA) was applied to displayed the correlations among different cell types.

## Sample selection, library preparation and single cell RNA sequencing (scRNA-seq)

To identify key transcription factors specifically expressed in the osteogenic lineage cells at the single-cell level, we selected scRNA-seq data (Ctrl and Fracture-PFD14) published by Kishor et al. (GSE154247), which excluded CD45+ and CD117+ cells from the homeostatic and post-fracture samples were chosen[37]. As data excluded hematopoietic and lymphoid cells from the callus, we further sorted CD31+ and CD45+ cells (lymphoid, and endothelial cells) and Lin-CD140α+ cells (stromal cells) from the mid-shaft of the femur in 8-week-old mice using flow cytometry, followed by scRNA-sequencing (detailed procedures in the 'Flow Cytometry Sorting' section). For the MSFL samples, based on previous studies[4], we performed single-cell sequencing on the digested tissues of the nasal respiratory mucosa and the newly formed region at 3, 6, 9, and 14 days after MSFL surgery (CRA007231). For the aforementioned CD31+ and CD45+ cells and Lin- CD140α+ cells of the femur, we utilized the Chromium Next GEM Single Cell 3' Kit version 3.1 (1000075, 10x Genomics) for scRNA-seq, following the manufacturer's instructions. In brief, single live cells were resuspended in PBS containing 0.04% BSA (A1933, Sigma-Aldrich) to a final concentration of 500–1200 cells/ml, as determined by a TC20 cell counter (BioRad). Ten thousand cells were encapsulated in droplets to generate nanoliter-scale gel beads in EMulsion (GEMs). GEMs were then reverse transcribed in a C1000 Touch Thermal Cycler (Bio-Rad) programmed at 53 °C for 45 min, 85 °C for 5 min, and held at 4 °C. After reverse transcription and cell barcoding, emulsions were broken and cDNA was isolated and purified with Cleanup Mix containing DynaBeads (37002D, Thermo Fisher Scientific) and SPRIselect reagent (B23318, Beckman), followed by PCR amplification. The amplified cDNA was then used for 3' gene expression library construction, and single-cell RNA libraries were sequenced using an Illumina NovaSeq 6000 sequencer with 150 bp paired-end reads.

## scRNA-seq data analysis

Consistent with our previous study, the Cell Ranger toolkit v3.1 (10x Genomics) was applied to aggregate raw data, filter low-quality reads, align reads to the mouse reference genome (mm10), assign cell barcodes, and generate a unique molecular identifier (UMI) matrix. The raw UMI matrix was processed to exclude genes detected in fewer than 10 cells and cells with fewer than 200 genes. We further quantified the number of genes and UMI counts for each cell, maintaining high-quality cells with thresholds of 500-120,000 UMIs and 400-8000 genes. To exclude cells with a high mitochondrial proportion, we used the default parameter of 20%, ensuring that most of the heterogeneous cell types were included for downstream analysis. To minimize the effect of mitochondrial genes on subsequent analysis, a regression was performed prior to clustering. Scrublet[38] (https://github.com/AllonKleinLab/scrublet) was applied to remove potential doublets, with an expected doublet rate of 6%, and cells with doublet scores above 90% were excluded. Batch normalization was performed using CCA2, which models the UMI counts with a regularized negative binomial model to remove sequencing depth variation, while adjusting variance by pooling information across genes with similar abundances. Dimension reduction and unsupervised clustering were conducted following the workflow of the Python toolkit Scanpy[39]. Briefly, dispersion-based methods were employed to detect the top 20 highly variable genes (HVGs). The normalized dispersion was obtained by scaling with the mean and standard deviation of the dispersions for genes. Percentages of mitochondrial gene counts and cell cycle-related gene counts were regressed out from the normalized expression matrices. For clustering 56,781 cells (Fracture) and 63,527 cells (MSFL) in total, PCA was performed on the variable gene matrix to reduce noise. Fifty components were calculated, and the top 40 PCs were used for downstream analyses[40]. Then, the Leiden algorithm[41] was applied to identify cell clusters, with a resolution parameter of 1, referring to the construction of the TISCH database[42]. Notably, the same principal components were also used for non-linear dimension reduction to generate the Uniform Manifold Approximation and Projection (UMAP) for visualization.

Following the first round of unsupervised clustering, we annotated each cluster based on their marker genes. In the fracture dataset, the major cell types included stem cells and progenitors (Stem/Pro), osteo-lineage cells (OLC), osteochondro-lineage cells (OCLC), smooth muscle cells (SMC), monocytes or macrophages (Mono/Mac),

endothelial cells (EC), B cells, T cells, erythroblasts (Ery), neutrophils (Neu), eosinophils or basophils (Eos/Baso) and megakaryocyte (Mega). For MSFL, the major cell types remained consistent with previous research, including chondrocytes, osteo-lineage cells (OLCs), pericytes, neural progenitor cells (NPCs), neutrophils, macrophages, epithelial cells (ECs), T cells, sustantacular cells (Sus), horizontal basal cells (HBC), immature olfactory sensory neurons (iOSN) and mature olfactory sensory neurons (mOSN). To depict the curve of gene variations with cell hierarchy from a higher resolution perspective, the OLC cluster in MSFL, along with the Stem&Pro, OLC and OCLC clusters in the fracture dataset, underwent a second round of clustering, following the same process as in the first round. The hierarchical sub-clusters were annotated based on key markers indicating osteogenesis (Alpl, Col1a2, Runx2, Sp7, Dmp1), chondrogenesis (Sox9, Col2a1, Acan) and stemness (Klf4, Prrx1, Ctsk).

To explain the connectivity between cell populations and reveal the pathways of cell differentiation, trajectory from osteoprogenitors to osteocytes in MSFL and from skeletal stem cells to mature osteoblasts in the fracture dataset were inferred using PAGA with RNA velocity-directed edges, utilizing the scVelo toolkit[43]. To calculate velocities, the 'Dynamic modeling' was applied with default settings[44]. Cells were organized based on the predicted trajectory using the 'velocity pseudotime' function, with the starting point located in the farthest cells within the osteoprogenitor cluster in MSFL or skeletal stem cell cluster in the fracture dataset compared to the other clusters.

## Histology
The tissue treatment process was detailed in our previous study[45]. Briefly, tissues were fixed in 4% PFA for 24 h and decalcified in 10% EDTA solution for 2 weeks at 4 °C. For paraffin embedded nasal specimens, the blocks were sectioned into 4 μm slices, followed by deparaffinized in xylene and gradient rehydration before staining. For cryosectioning of fracture samples, the specimens were dehydrated overnight in 30% sucrose and sectioned into 10 μm slices. Masson's trichrome (G1340, Solarbio) and Toluidine blue staining (E670105, Sangon) were performed using commercial staining kits following the manufacturer's instructions. Images of regular staining were captured using a Nikon ECLIPSE Ci-E microscope with a 10x objective.

## Multiplex immunohistochemistry and confocal imaging
Multiplex immunohistochemistry staining was performed using the TSA-based system (Thermo Scientific) as reported previously[4]. Briefly, after antigen retrieval, peroxidase, and non-specific signal blockages, multicolor IHC commenced with sequential incubation of different primary antibodies, HRP modified secondary antibodies, and TSA reagents with different fluoresceins, followed by the stripping of the previous round of antibody and incubation of the next one. The primary antibodies used in mIHC (dilution 1:400 for all antibodies) included goat anti-mouse/human/rat Itgav (AF1219, Novus Biologicals), mouse anti-mouse/rat CD90 (NB100-65543, Novus Biologicals), mouse anti-mouse/human CD105 (NBP2-22122, Novus Biologicals), rabbit anti-human/mouse/rat CD200 (AF2724, Novus Biologicals), rabbit anti-mouse/human/rat Runx2 (ab236639, Abcam), rabbit anti-mouse/human/rat Sox9 (ab185966, Abcam), rabbit anti-mouse/human Alpl (MA5-24845, Invitrogen), rabbit anti-mouse/human/rat Zfp260 (ABE295, Merck), mouse anti-human/mouse/rat p300 (NB100-616, Novus Biologicals), rabbit anti-human/mouse MED1 (NB100-2574, Novus Biologicals), rabbit anti-human/mouse BRD4 (NBP2-76393, Novus Biologicals), mouse anti-human/mouse/rat Prkca (NB600-201, Novus Biologicals), rabbit anti-V5 tag (13202, CST), mouse anti-Collagen type I (67288-1-Ig, proteintech), rabbit anti-Collagen type II (28459-1-AP, proteintech). The tyramide reagents used in mIHC included AF350 (B40952, Invitrogen), AF488 (B40953, Invitrogen), AF546 (B40954, Invitrogen), AF594 (B40957, Invitrogen) and AF647 (B40958, Invitrogen). Images were acquired using a Leica SP8 confocal microscopy with the adjustable wavelength receiving module to separate the spectrum of the channels and avoid overlap.

## Image analysis
Imagej 1.54 h was applied for the analysis of multicolor images. To visually demonstrate the co-staining regions of transcription factors Zfp260, Runx2, and Sox9 with SSC lineage cells in morphology, the mIHC was applied. Due to the difficulties in distinguishing cell clusters with multiple merging channels, the clusters were segmented in the bone calluses of fracture and newly formed tissues of MSFL based on the marker panels reported in the previous researches, including SSC (Itgav$^+$THY1$^-$CD200$^+$CD105$^-$), BCSP (including pre-BCSP (Itgav$^+$THY1$^-$CD200$^-$CD105$^-$) and BCSP (Itgav$^+$THY1$^-$CD105$^+$)), THY (Itgav$^+$THY1$^+$CD200$^-$CD105$^+$), BLSP (Itgav$^+$THY1$^+$CD105$^-$), PCP (Itgav$^+$THY1$^+$CD200$^+$CD105$^+$) and their co-staining regions with Zfp260, Runx2, and Sox9 signals. At the final presentation, Itgav$^+$ cells were colored in gray and clusters of SSC/BCSP/THY&BLSP/PCP were labeled in purple, with Zfp260/Runx2/Sox9 marked in yellow. The operating pipeline was as follows. Each channel image was adjusted to 8-bit. The threshold of each channel was adjusted to obtain the positive staining area, stored as the positive ROI. The 'Invert' command was used to select the negative staining area, followed by the recording of negative ROI. By stacking ROI regions of certain channels, the ROI of a specific cluster was obtained and displayed by using the command 'creating mask'. Signal areas smaller than 5×5 dpi were excluded as non-cellular noise, with the remaining signal region colored by gray, purple, or yellow as previously mentioned. To calculate the co-concurrence percentage of each transcription factor in each cluster, the areas of co-staining TF and certain clusters were measured separately using the 'measure' command, followed by the calculation with the formula Area$_{TF\&Cluster}$/Area$_{Cluster}$. The percentage of three different regions was evaluated in each mIHC staining image. Similarly, the co-staining percentage of ZsGreen$^+$ cells labeled by rAAV9 with Alpl and Runx2 or Krt14 and Runx2 was calculated.

To calculate the nuclear translocation efficiency of Zfp260 or transfected Zfp260-V5, the nuclear or cytosolic region were selected, with respective ROI established. After measuring the MOI values of the above regions, the values were substituted into the formula MOI$_{nucleus}$/MOI$_{cytosol}$ to calculate the translocation efficacy. Randomly selected 30 cells of each group were calculated.

For the co-localization analysis of Zfp260, Brd4, p300, and Med1 within the cell nucleus, the 'straight line' tool was applied to select ROI. Within each signal channel, the command 'plot profile' was applied to obtain a matrix table of fluorescence intensity changes along the selected ROI for each channel. The matrix table of each channel was plotted as a curve with distance (μm) on the x-axis and fluorescence intensity on the y-axis.

## Flow cytometry cell sorting or analysis
For the analysis of SSC lineage cells, the callus of fracture mice (5 mice per biological repeat, $n = 2$ per time point), along with the newly formed tissues of MSFL models (15 mice per biological repeat, $n = 1$ per time point) were dissected under stereomicroscopy. For scRNA-seq, the mid-shaft of mice femurs (5 8-week-old mice) were dissected. The tissues were digested in DPBS containing 0.3% collagenase, 0.3% dispase II and 0.01% DNase I at 37 °C for 30 min in a shaking bath, followed by filtering with 70 μm cell strainers (352350, BD Falcon). After centrifugation at 4 °C at 300 × g for 10 min, the supernatant was discarded, and cells were suspended in red blood cell lysis buffer (00433357, eBioscience) at 4 °C for 5 min. After neutralization and centrifugation, the cell pellets were resuspended by staining buffer containing the CD16/32 blocking antibody (156603, Biolegend) and incubated for 30 min, followed by staining with antibodies. All antibodies were diluted at 1:200 except for the anti-CD45 antibody, which was diluted at 1:800. Antibodies used included: CD140a (Pdgfra)-BV605 (APA5,

Biolegend), CD31-PE/Dazzle™ 594 (MEC13.3, Biolegend), CD45-PE/Dazzle™ 594 (30-F11, Biolegend), TER119-PE/Dazzle™ 594 (TER-119, Biolegend), CD105-APC/Cy7 (MJ7/18, Biolegend), CD200-APC (OX-90, Biolegend), CD51 (Itgav)-biotin (RMV-7, Biolegend), BV421-Streptavidin (405226, Biolegend), Ly-51-PE/Cy7 (6C3, Biolegend), CD90.1 (Thy1.1)-PerCP/Cy5.5 (OX-7, Biolegend), CD90.2 (Thy1.2)-PerCP/Cy5.5 (30-H12, Biolegend). Zombie Aqua™ was applied to distinguish live and dead cells. BD FACSAria™ III was applied for cell sorting, while the BD-LSRFortessa was used for FACS analysis. The gating strategy for mouse skeletal stem cell lineages referred to previous research, with signal compensation performed on FMO controls. For sorting Lin⁻ZsGreen⁺ cells from $Ctsk^{creER};H11^{RSE\text{-}tdTomato\text{-}LSL\text{-}zsGreen}$ and $Ctsk^{creER};H11^{RSE\text{-}tdTomato\text{-}LSL\text{-}zsGreen};Zfp260^{fl/fl}$ modeling mice, the cells were gated from single live cells, Lin⁻ cells and ZsGreen⁺ cells. For scRNA-sequencing, single live Lin⁻CD140α⁺ cells and single live CD31⁺ and CD45⁺ cells were sorted and applied for the subsequent sequencing separately.

## Micro-CT analysis

Mice modeled by fracture and MSFL from different treatment groups were sacrificed and perfused with 4% PFA. Following an additional 24 h of fixation, their femurs and nasal bones were dissected for the subsequent radiological analysis without decalcification. Two-dimensional (2D) images of fracture scanning sections, reconstructed 3D images of whole femurs or newly formed tissues of MSFL, and calculated structural indices were obtained using Micro-CT 50 (Scanco Medical) with a resolution of 10 μm. Quantitative analysis of the callus by fracture and newly formed tissues by MSFL were performed using the associated analyzing software. Abbreviations used for morphometric parameters were as follows: BMD, bone mineral density; BV, bone volume.

## Isolation, culture and treatment of periosteal stem cells (PSCs)

PSCs were isolated from intact femur, tibia and maxillofacial bones from 8wk-old $Ctsk^{creER};H11^{RSE\text{-}tdTomato\text{-}LSL\text{-}zsGreen}$ and $Ctsk^{creER};H11^{RSE\text{-}tdTomato\text{-}LSL\text{-}zsGreen};Zfp260^{fl/fl}$ mice. The mice were labeled with Tamoxifen (50 mg/kg) (T5648, Merck) for 3 consecutive days at 7 weeks of age. Adherent soft tissues were removed and the periosteum was scratched and digested in DPBS containing 0.3% collagenase, 0.3% dispase II, and 0.01% DNase I at 37 °C for 30 min in a shaking bath. Following digestion, FACS cell sorting was performed to isolate single live Lin⁻ZsGreen⁺Itgav⁺THY1⁻6C3⁻CD105⁻CD200⁺ cells. After a two-week culture period, all clones were selected and passaged to P1. The P1 PSCs were applied for all experiments, except for co-immunoprecipitation or ChIP in this study.

Due to the necessary quantity of cells, PSCs for co-immunoprecipitation or ChIP would undergo an immortalization-based amplification approach. The P1 PSCs were infected by lentivirus expressing human telomerase reverse transcriptase (hTERT). The cells were then further expanded to satisfy the quantity need for the subsequent pull-down and ChIP assays.

GO 6976, a potent PKCalpha inhibitor, was used for the inhibition experiments. PSCs were pretreated with 1 μM GO 6976 (HY-10183, MCE) 6 h before osteogenic induction in the experimental group, with an equal amount of DMSO supplemented in the control group.

For osteogenic induction, PSCs were cultured in osteogenic induction medium (α-MEM containing 10% FBS, 10 nM dexamethasone, 50 μg/mL ascorbic acid, 1% penicillin/streptomycin and 10 mM β-glycerophosphate). Alizarin Red S staining was performed after 14 or 21 days of induction. For the observation of nuclear translocation and immunoprecipitation, induction durations of 6 or 24 h were administered according to the experimental design.

## Total RNA extraction and quantitative real-time PCR (qRT-PCR)

Total RNA was extracted from FACS-sorted cell clusters (SSC, BCSP, THY + BCSP and PCP) using centrifuge tubes, following the manufacturer's instructions (12183016, Invitrogen). For in vitro cultured PSCs, RNA extraction was performed using TRIzol (15596018, Invitrogen) as previously described. First strand cDNA synthesis was applied using the RevertAid First Strand cDNA Synthesis Kit (K1622, Thermo Scientific). Quantitative real-time PCR (qRT-PCR) was performed using PowerTrack™ SYBR Green Master Mix (A46012, Applied Biosystems). Detection was conducted on a QuantStudio 7 Flex (Thermo Scientific), with β-Actin as the internal control. The primer sequences were listed in the Supplementary Data 1.

## Chromatin immunoprecipitation (ChIP)

PSCs with different treatments were subjected to ChIP according to the manufacturer's protocol of Pierce Magnetic ChIP kit (26157, Thermo Scientific). Briefly, cells were crosslinked with 1% formaldehyde for 10 min, followed by quenching with 125 mM glycine for 5 min. Cells were then washed three times with ice-cold PBS, and the pellets were collected. Cell lysis was performed for 10 min on ice. Nuclei were collected by centrifugation and digested with MNase to ensure DNA fragments ranged from 150 bp to ~1000 bp. Soluble chromatin was immunoprecipitated overnight with magnetic beads with antibodies for V5 (ab15828, Abcam), H3K27ac(8173, CST), or Brd4 (13440, CST), along with appropriate isotype controls. For each reaction, 4 μg of antibody was used. Immunoprecipitated beads were collected, followed by protein digestion and DNA cleanup.

For wild-type PSCs transfected with Zfp260-V5 and immunoprecipitated with anti-V5, DNA-library construction was performed using the VAHTS Universal DNA Library Prep Kit for Illumina V4 (ND610, Vazyme) according to the manufacturer's protocol. Novaseq 6000 was applied for high-throughput sequencing with a strategy of PE150. For wild-type and $Zfp260^{-/-}$ PSCs, immunoprecipitation was carried out using anti-H3K27ac or anti-Brd4 antibodies. For $Zfp260^{-/-}$ PSCs transfected with Zfp260-V5 and its mutants (Y173F, S182G, S197G, Y173F-S182G, Y173F-S197G, S182G-S197G and triple mutants), immunoprecipitation was performed with anti-H3K27ac, anti-Brd4, or anti-V5 antibodies. Quantitative PCR (qPCR) was then performed on purified ChIP and input DNAs at target loci, with enrichment compared to respective isotype control IgG. The primer sequences for target loci were listed in the Supplementary Data 1.

## CUT & Tag

FACS sorted Lin⁻zsGreen⁺ cells from fracture or MSFL models were collected for the Cut & Tag workflow using the Commercial CUT&Tag-IT Assay Kit (53160, Active Motif), following the manufacturer's protocol. Brief, the FACS sorted cells were centrifuged, resuspended, and captured by ConA beads. Cells were permeabilized with 0.05% digitonin and then incubated overnight with primary antibodies against H3K27ac (8173, CST) or H3K4me1 (5326, CST). A guinea pig anti-rabbit secondary antibody (provided in the kit) was diluted in 1:100 and used for incubation. After a 1-h incubation for assembled pA-Tn5 transposomes binding, the tagmentation buffer was administered to fragment the DNA for 1 h. The fragmented DNA was then enriched and purified for PCR amplification using i5 and i7 index primers for Illumina sequencing. Novaseq 6000 was applied for high-throughput sequencing with a PE150 strategy.

## ATAC-seq

FACS sorted cells were used for ATAC-seq with library construction following the manufacturer's constructions (TD711, Vazyme). Briefly, the cells were collected, washed and lysed using 50 μL of lysis buffer (10 mM Tris-HCl, pH 7.4, 10 mM NaCl, 3 mM MgCl2, and 0.02% digitonin, 0.1% NP-40) on ice for 10 min. For the fragmentation process, 50 μL of fragmentation buffer containing 10 μL 5X TTBL and 4 μL TTE mix V50 was added to the lysed cells. The mixture was incubated at 37 °C for 30 min. The reaction was stopped by adding 5 μL stop buffer at room temperature for 5 min. After fragmentation, DNA was

extracted and the library was amplified, followed by size selection with AMPure XP beads (20498100, Beckman) ranging from 200 to 800 bp. All libraries were sequenced on the Illumina NovaSeq 6000 platform.

## Analysis of ChIP-seq, CUT&Tag and ATAC-seq

The raw data from ChIP-seq, CUT&Tag, ATAC-seq, and the downloaded dataset (GSM6245143) were analyzed using the HiChIP pipeline[46]. Briefly, FastQC (https://www.bioinformatics.babraham.ac.uk/projects/fastqc/) and Cutadapt (https://cutadapt.readthedocs.io/en/stable/) were used for quality control and adaptor trimming, respectively. Paired-end reads were mapped to the mm10 genome reference using Bowtie2 (https://bowtie-bio.sourceforge.net/bowtie2/index.shtml), followed by duplicate removal with Sambamba[47]. Narrow peaks for H3K4me1, H3K27ac, Brd4, and Zfp260-V5 were identified using the model-based analysis of ChIP-seq (MACS2) (https://pypi.org/project/MACS2/) package with a cutoff of 1.00e-05. For data visualization, BEDTools[48] and custom scripts were used to generate per-million read density profiles with a 200-bp window size and a 20-bp step size. Signal tracks were visualized using Integrative Genomics Viewer (IGV) software.

The number of reads in the TSS ± 3 kb region of all protein-coding genes was estimated and normalized to 10 million (RP10M), with log2 transformation and quantile normalization applied to each of the ChIP-seq, CUT&Tag, and ATAC-seq libraries. The read density (RPM, reads per milion) over the TSS ± 3 kb region was calculated using the ngs.plot tool (v2.02)[49]. The binding motif of Zfp260-V5 was identified using MEME, and peak-annotated genes were used for the enrichment analysis of Gene Ontology-Biological Process (GO-BP) (http://geneontology.org/) and the Kyoto Encyclopedia of Genes and Genomes (KEGG) (https://www.genome.jp/kegg/).

Super enhancer calling was performed using H3K27ac data from both control and Zfp260-cKO Lin−zsGreen+ cells sorted from fracture and MSFL models. Using the ROSE tool (v1.0.0, http://younglab.wi.mit.edu/super_enhancer_code.html)[50], super enhancers were identified based on default parameters, including the merging of remaining peaks within a distance of 12.5 kb or less. Compared to the Zfp260-cKO group, the differential super enhancers identified in the control groups were used for GO-BP enrichment analysis.

## GST pull-down, mass spectrometry and analysis

The coding sequence of *mus musculus Zfp260* was cloned and fused into the pGEX6p-1-GST plasmid using the In-Fusion strategy (639650, Clontech). The pGEX-GST and pGEX-GST-Zfp260 plasmids were transformed into BL21 *E. coli*, and target protein synthesis was induced by 1 mM IPTG at 16 °C for 24 h. The *E.coli* pellets were collected, washed, and subjected to thorough ultrasonication and lysis. Centrifugation was performed to separate bacterial debris from the bait protein-containing supernatant. The glutathione magnetic agarose beads (78602, Thermo Scientific) were equilibrated and used to capture GST and GST-Zfp260 in the supernatants for 3 h. The beads were then washed and incubated with whole cell lysates of PSCs at 4 °C overnight. The bound proteins from each group were eluted and visualized by silver staining (24600, Thermo Scientific). The peptides were enzymatically hydrolyzed, extracted, and purified for separation by EASY-nLC 1000 (Thermo Scientific) using an analytical column (C18, 1.9 μm, 75 μm × 20 cm) at a flow rate of 200 nL/min. For mass spectrometry analysis, the Orbitrap Fusion Lumos (Thermo Scientific) was used in Data Dependent Acquisition (DDA) mode.

Identification of peptides and phospho-modifications by database searching were performed by using Proteome Discoverer 2.4 software for subsequent analysis.

## Plasmids construction and lentivirus packaging

In vitro manipulation of Zfp260 expression was achieved through lentivirus infection. To overexpress Zfp260, the pCDH-*Zfp260*-V5

lentivirus expression vector was constructed by cloning and fusing the CDS of *mus musculus Zfp260* into the MCS site (EcoRI and BamHI) of the pCDH-V5-Puro plasmid via the In-Fusion strategy (639650, Clontech). For lentivirus packaging, a three-plasmid system (pol/gag, VSV-G and pCDH-V5/pCDH-*Zfp260*-V5) were transfected into the 293 T cell line (H4-1401, Cyagen Biosciences). The lentivirus was concentrated and purified by ultracentrifugation. The aforementioned viruses were used to infect wild-type PSCs with an MOI of 100. The infected PSCs from each group were applied for subsequent tests according to the experimental design.

To establish point mutations based on pCDH-*Zfp260*-V5, the QuickMutation Plus point mutation kit (D0208, Beyotime) was applied with primers designed following the manufacturer's protocol. Taking pCDH-*Zfp260*-V5 as the template, pCDH-Y173F-V5 (518A-T), pCDH-S182G-V5 (544A-G), pCDH-S197G-V5 (589A-G), pCDH-Y173F-S182G-V5 (518A-T; 544A-G), pCDH-Y173F-S197G-V5 (518A-T; 589A-G), pCDH-S182G-S197G-V5 (544A-G; 589A-G), and pCDH-triple-mut-V5 (518A-T; 544A-G; 589A-G) were constructed for subsequent lentivirus packaging as mentioned above. The aforementioned viruses were used to infect *Zfp260*−/− PSCs with an MOI of 100, and subsequent analysis was carried out according to the experimental design.

## Plasmids construction and packaging of Adeno-associated virus-9 (rAAV9)

In the in vivo rescue experiment, the pAAV-CMV-MCS-IRES-ZsGreen vector was modified to achieve PSC-specific expression[51]. The CMV promoter and enhancer sequences of the original vector were removed by cutting at the NheI and PstI restriction sites. Using high-fidelity enzyme (P510, Vazyme), the promoter sequence of the mouse Ctsk gene (TSS + 2 kb) was cloned into the site via In-Fusion strategy (639650, Clontech), resulting in the pAAV-Ctsk-MCS-IRES-ZsGreen vector. The promoter sequence was listed in the Supplementary Data 1. Using the aforementioned 8 pCDH vectors (pCDH-*Zfp260*-V5, pCDH-Y173F-V5, pCDH-S182G-V5, pCDH-S197G-V5, pCDH-Y173F-S182G-V5, pCDH-Y173F-S197G-V5, pCDH-S182G-S197G-V5, and pCDH-triple-mut-V5) as templates, the wild-type and all mutant types of *Zfp260* coding sequences were cloned into the MCS site, with the Kozak sequence inserted ahead of the CDS region to enhance downstream gene transcription. AAV packaging was performed using the AAV-293T cell line (632273, Takara). The pAAV-RC9 and pAAV-helper packaging vectors were mixed with the pAAV expression vectors (a total of 8 groups) to form a triple-plasmid system. PEI transfection reagents (HY-K2014, MCE) were applied for transfection when cell confluency reached 80%-90%. Cells and supernatant were collected 72 h after transfection. Virus extraction and purification were performed using the AAVpro® Purification kit (6666, Takara) according to the manufacturer's protocol, followed by titer determination. In the in vivo experiment, each mouse was intravenously administered the rAAV9 at a dose of $4 \times 10^{11}$GC $(2 \times 10^{13}$GC/kg). Two weeks after administration, the infection efficiency in the heart, liver, spleen, lung, kidney, skull, jaw and legs was evaluated using IVIS® Spectrum In Vivo Imaging System (PerkinElmer) in the EGFP channel.

## Phos-assay SDS-PAGE

Phos-Assay SDS-PAGE[52] was performed to detect phosphorylated modifications of Zfp260. An 8% SDS-PAGE gel was prepared by adding 5 mM Phos-assay and 10 mM ZnCl$_2$ to final concentrations of 25 μM and 50 μM, respectively. Electrophoresis was conducted under a constant current of 30 mA. For MS analysis, Coomassie Brilliant Blue (CBB) staining was performed. For Western immunoblotting, gels were pre-treated with 10 mM EDTA and washed before membrane transfer.

## Immunoprecipitation (IP) and western immunoblotting (WB)

The whole cell lysates (WCL) of different cell types subjected to various treatments were prepared according to the study design. To extract

WCL, cells were washed twice with ice-cold PBS and lysed with ice-cold IP lysis buffer supplemented with 1 mM PMSF and a protease inhibitor cocktail at the working concentration. The cell lysates were incubated on ice for 30 min before centrifugation at $13,000 \times g$ for 10 min at 4 °C.

For immunoprecipitation, a pre-clear step was performed using Pierce Protein A/G Magnetic Beads (88802, Thermo Scientific) for 2 h at 4 °C before the formal IP process. The magnetic beads were then discarded, and the pre-cleared supernatants from each group, containing equal amounts of protein were subjected to the formal IP process with specific antibodies through overnight incubation, followed by magenetic bead capture.

The detailed procedure for Western immunoblotting has been described in previous research[53]. Antibodies used in this study at a 1:1000 dilution included: mouse anti-p300 (NB100-616, Novus Biologicals), rabbit anti-MED1 (NB100-2574, Novus Biologicals), rabbit anti-BRD4 (NBP2-76393, Novus Biologicals), rabbit anti-V5 tag (13202, CST), rabbit anti-Flag tag (14793, CST), mouse anti-Prkca (NB600-201, Novus Biologicals), rabbit anti-β Actin (4970, CST), and rabbit anti-Histone H3 (4499, CST). To avoid interference from light and heavy chains, the VeriBlot for IP Detection reagent (conjugated with HRP, ab131366, Abcam) was used at a 1:200 dilution for the detection of immunoprecipitated samples. For input samples, conventional HRP-conjugated secondary antibodies specific to the corresponding species were diluted 1:2000 for WB signal detection.

### The protein structure modeling, docking and molecular dynamic simulations
The protein structure files involved in this study were obtained from the PDB database or predicted using AlphaFold2[54]. The proteins were dehydrated and hydrogenated using Pymol (http://www.pymol.org/pymol), including wild-type Zfp260 (AF2, AF-Q62513-F1, average pLDDT Score=80.05). Additionally, the structural models of seven Zfp260 mutants were predicted online using AlphaFold2, with their highest-ranked prediction model selected for subsequent analysis. The average pLDDT values for these mutants were all greater than 80.

For the docking of Zfp260 (wild-type or mutant) with Prkca, the GRAMM-X server was applied[55]. The conformation with the lowest binding energy between Zfp260 and Prkca was analyzed and displayed, and the shortest distance between the amino acids 173, 182 or 197 of Zfp260 and the Prkca catalytic domain (328-668 amino acids) was calculated.

### Renal capsule transplantation
As previously described[3], eight-week-old male C57BL6/J mice were anesthetized and shaved on the left flank and abdomen prior to the sterilization of the surgical site. A 1-cm incision was made to externalize the kidney, and a 2-mm pocket was created in the renal capsule. A 5-μl Matrigel (Corning, 356231) containing 8,000 cells was implanted beneath the capsule, and the opening was sealed using a cauterizer before repositioning the kidney into the body cavity. The animals were euthanized after four weeks. For FACS, the kidneys were dissected and subjected to digestion in accordance with the part **"Flow cytometry cell sorting or analysis"**. To assess bone formation, the kidneys were fixed in 4% PFA for six hours and subsequently analyzed using micro-CT.

### Statistical analysis
No methods were used to predetermine sample size, and no blinding or randomization was employed for data analysis. Data were presented as means ± SD, with the number of independent experiments indicated in the legends. Independent two-tailed Student's $t$ test, one-way ANOVA, or two-way ANOVA was applied to analyze the data after determining whether the data were normally distributed. SPSS 26.0 (IBM) was used for all statistical calculations. Significant differences were considered at $p$ values below 0.05 and were indicated in the figures.

### Reporting summary
Further information on research design is available in the Nature Portfolio Reporting Summary linked to this article.

## Data availability
All of the scRNA-seq, bulk RNA-seq, CUT&Tag, ChIP-seq and ATAC-seq data generated in this study have been deposited in the Genome Sequence Archive (GSA) database of the National Genomics Data Center (NGDC) with the accession number CRA019163 under the BioProject PRJCA026768. The mass spectrometry data for GST and GST-Zfp260 pull-down assays reported in this study have been deposited in the NGDC OMIX database (OMIX ID: OMIX007435) and to the ProteomeXchange Consortium via the integrated proteome resources (iProX) database with the dataset identifier PXD057880 [https://www.iprox.cn/page/project.html?id=IPX0010239000]. Other published datasets applied in this study are as follows: GSE154247; GSE152677; GSE206155; CRA007231. All other data generated in this study are provided in the Supplementary Information/Source Data file. Source data are provided with this paper.

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

## Acknowledgements

This study was sponsored by the Original Exploration Program of the National Natural Science Foundation of China (82350001 to Z.W.), the Young Elite Scientist Sponsorship Program by CAST (2023QNRC001 to Y.W.), the Young Scientist Fund of the National Natural Science Foundation of China (82201033 to Y.W.), the Shanghai Sailing Program of Science and Technology Commission of Shanghai Municipality (22YF1451600 to Y.W.) and the Chenguang Program of Shanghai Education Development Foundation and Shanghai Municipal Education Commission (23CGA23 to Y.W.). Figures 1a, 4c, 7b and Fig. 8 are created in BioRender. li, z. (2024) https://BioRender.com/d18i396 and https://BioRender.com/k73i285.

## Author contributions

Z.W. designed and supervised the research and wrote the manuscript. Y.W. conducted the majority of the experiments and participated in the analysis of scRNA-seq and bulk RNA-seq data. Y.F. was responsible for the phenotypic analysis in animal experiments. Z.L. assisted with the molecular biology experiments. S.X. participated in in vitro cell culture. D.W. contributed to the FACS experiments. J.H. helped create the schematic diagrams. D.W., J.H., and H.W. participated in breeding and crossbreeding of the mouse models.

## Competing interests

The authors declare no competing interests.
