## [Transparent Peer Review file · Nature Communications]

Zfp260 choreographs the early stage osteo-lineage commitment of skeletal stem cells

Corresponding Author: Professor Zuolin Wang

Version 0:

Reviewer comments:

Reviewer #1

(Remarks to the Author)

In this manuscript by Weng et al., the authors perform single cell transcriptional profiling on CTSK-lineage SSCs in the setting of fracture healing and MSFL, identifying Zfp260 as a transcription factor associated with the early differentiation stages in this lineage. Subsequently, the phenotypic contribution of Zfp260 to this lineage, associated transcriptional/epigenetic regulation, and phosphoregulation are studied. The topic is of great interest overall, and the studies are largely seen as robust, though a few important points where additional data is needed are noted below. This manuscript has several areas of strength. One, the work here to try and identify the precise differentiation stage at which Zfp260 acts is notable and distinguishes the present work from many similar studies, though some additional experiments are suggested here to more strongly support this conclusion. Additionally, the in vivo rescue of CTSK-creER Zfp260 fl/fl mice by the AAV9-delivered Zfp260 mutants is considered impressive and beyond the typical scope/rigor of many similar studies in the literature. Overall, while there are some areas where further studies are needed to fully support the conclusions, there is enthusiasm for this study. Major points number 1, 2 and 7 below are considered the most critical areas where new data is needed.

Major points:

1. Is there a lineage reporter in CTSK-creERT2 mice? Both gating the flow analysis on the CTSK-lineage in the Zfp260 fl/fl CTSK-creERT2 mice and analyzing the histologic distribution of the CTSK lineage in these same mice vs controls is important for clarifying the nature of the cellular defect and to ensure that the directly impacted lineage is analyzed separately from the broader pool of skeletal cells. Use of this lineage reporter would also enable demonstration of Zfp260 deletion efficiency.
2. The claim that there is a SSC to BCSP transition defect in the absence of Zfp260 is currently only supported by indirect evidence. Alternatively, the data observed could be due to cell death in CTSK-lineage BCSPs or other mechanisms. To provide direct evidence of a SSC to BCSP transition defect, WT and Zfp260-deficient CTSK-lineage SSCs should be isolated, placed into an transplantation based organoid system in the kidney capsule/muscle/inguinal fat pad, and re-analyzed by flow to demonstrate the progress of cellular differentiation. Direct transplantation of WT and Zfp260-deficient CTSK-lineage BCSPs in the same system is also recommended to distinguish the SSC to BSCP versus post-BSCP defects present. If possible, uCT analysis of bone formed in this organoid system would be ideal and would help to establish the lineage intrinsic ossification defect associated with Zfp260 deficiency, though in this setting the uCT data is less important than the flow cytometry-based cellular differentiation endpoints.
3. It is suggested that the data in Fig 5 be confirmed by sorting of the relevant populations and Runx2 qPCR or qPCR for other Runx2 target genes in at least one of the models to confirm the co-staining results. It is very confusing to provide a "stem and pro" group alongside separate SSC, pre-BCSP, ect... recommend removing this.
4. Annotating a scRNA-seq cluster as representing an SSC is not straightforward. It is suggested to try and further build out the case for the proposed stem cell annotation. This can include analysis of a wider range of associated transcripts, like CD200, or cross-referencing broader transcriptional data in Bok et al. or Debnath et al. on CTSK-lineage stem cells. Similarly, much more detail is needed to justify the annotation of pre-BCSPs and BCSPs.
5. It is not expected in the PAGA analysis that SSCs are separated from what are presumably their downstream derivatives

including osteolineage cells. Additionally, is the sample expected to only contain CTSK-lineage intramembranous specialized cells? There is some question whether both endochondral and intramembranous competent SSCs are present in the specimen and whether this can be represented in the scRNA-seq analysis.

6. What is the basis to consider "PCP" as separate from BCSPs and "BLSP" cells as separate from THY+ cells? Having a strong justification for the cell types considered here is considered important and the evidence that each of these populations are biologically distinct may not be fully established. It is likely beyond the scope of this study to establish such evidence here, but additional consideration of these elements of the classification schema used would strengthen the study.

7. The baseline phenotype of Zfp260 deletion in CTSK-lineage cells should be established outside of the fracture and MSFL models.

Minor points 1. Line 47-48, typos are present. 2. PAGA and trajectory analysis don't really confirm cluster annotation accuracy as claimed on line 127. 3. Line 131 and 135, how can it be claimed that ZFP260 is upregulated in the SSC to BSCP transition when BCSPs are not annotated in the scRNA dataset? 4. Line 142, recommend not claiming that Zfp260 is the sole upregulated transcription factor or at least slightly qualifying this, as the sparse transcriptional sampling in scRNA-seq means that many relevant transcription factors could be missed.

3. As more of a comment to consider in the writing than an issue to be addressed experimentally, the initial Chan reports focus much more on establishing SSC identity than the downstream differentiation sequence. While the report here is an accurate representation of the Chan differentiation sequence, it is worth keeping in mind that this topic of whether pre-BCSPs truly turn into BCSPs than in turn turn into THY1+ and 6C3+ cells is an area that has had only minimal dedicated study and may therefore be subject to future revisions. Some slight additional qualification along these lines is appropriate.

5. It is worth clarifying (such as line 162) that CTSK alone does not specify PSCs, but rather CTSK in concert with additional cell surface markers and ideally anatomic location.

6. Line 176, suggest more clearly spelling out how intramembranous and endochondral regions of the callus are discriminated as this is often not trivial. Keep in mind that CTSK-lineage PSCs gain endochondral bone formation capacity in response to fracture, so endochondral defects might be expected with severe functional defects in this lineage.

7. Fig 2d--what is "dpm"? Fracture BMD can be tricky to interpret. Was the uCT used calibrated with phantoms for this purpose? The fracture/MSFL BMD data may be more appropriate in the extended data

8. Why was phospho-regulation of Zfp260 expected? The logical flow of the manuscript at this point is a bit weaker, and "transcription factors are often phosphorylated" is not the strongest rationale to justify this line of investigation. It should be better explained in the text that Prkca was identified by an IP mass spec approach in Fig 6a, b.

9. It should be clarified how sufficient numbers of PSCs were obtained for the pulldown study in Fig 6, typically obtainable cell numbers are far too small for biochemical studies. Was in vitro expansion used and what were the conditions and duration of such expansion? Recommend putting this in the Fig legend.

10. Line 352--it is questionable whether this phenotype is best described as an osteogenesis imperfecta phenotype. Perhaps just "impaired fracture healing" is more appropriate. Recommend rephrasing the section title in more positive terms: "Zfp260 phosphorylation sites are required for..."

11. Line 430 should be Chan et al., not Charles et al. This discussion is where some slight qualification that the differentiation sequence downstream of SSCs is not well studied overall is suggested.

12. Is Zfp260 specific for CTSK-lineage stem cells or is it more widely expressed during the differentiation of non-CTSK lineage stem cells?

Reviewer #2

(Remarks to the Author)

In the manuscript, "Zfp260 choreographs the early stage osteo-lineage commitment of skeletal stem cells", Weng et al identified Zinc Finger Protein 260 (Zfp260) as a key driving factor of skeletal stem cells (SSCs) to bone, cartilage and stromal progenitors (BCSPs) maturation and osteogenic fate determination during skeletal injury. The deletion of Zfp260 in periosteal stem cells using CTSKCRE line arrests the transition of pre-BCSP to PCSP stage, resulting in impaired bone fracture healing and regeneration. The authors demonstrated that during the early stages of skeletal injury repair, Zfp260 expression is upregulated and that it regulates osteogenic gene expression via formation of the super-enhancer complexes. They also identified PKCa as an upstream kinase of Zfp260 and demonstrated that PKCa-mediated phosphorylation is important for biological function of Zfp260. Overall, novel roles of Zfp260 in osteogenic differentiation of SSCs, the soundness of the experimental approach, and potential in bone fracture therapy are appealing for publication in the Nature Communications. However, experiments are limited to understand Zfp260's roles in periosteal stem cell development during the early stages of bone fracture repair and regeneration. Additionally, the roles of Zfp260 in osteoclast development due to CTSKCRE-mediated deletion in osteoclasts and in general skeletal stem cell populations under physiological conditions are largely missing. Finally, there is no evidence showing Zfp260 as a transcription factor.

Major Concerns:

1. The authors need to clarify Zfp260 as a chromatin remodeling factor, not a transcription factor. There is no evidence supporting the biological functions of Zfp260 as a transcription factor.

2. The data clearly demonstrated Zfp260's roles in the transition between pre-BCSP and BCSP. This should be clarified

throughout the manuscript.

3. Since CTSKCRE line is also used to delete osteoclast genes and osteoclasts are important for bone fracture healing/regeneration processes, Zfp260's roles in osteoclast differentiation and bone resorption should be examined.
4. What is the kinetics of Zfp260 expression during skeletal stem cell development under physiological conditions (early skeletal development)? Its expression and functions are specific to periosteal stem cell development?
5. It is important to recognize that callus size alone does not necessarily correlate with improved healing outcomes. The authors need to examine unionization rates of fracture sites and fibrosis formation in non-unionization sites.
6. The rationale using CTSKCRE-ERT2 line, instead of CTSKCRE line, is not clear since tamoxifen treatment also affects osteoclast development during fracture healing. What are basal skeletal phenotypes of Zfp260^{fl/fl};CTSKCRE mice?
7. Figure 4j and 6m: ARS staining represents mineralization activity only. To examine osteogenic development of SSCs, the authors need to perform CFU, alkaline phosphatase activity, and osteogenic gene expression. Kidney capsule transplantation of SSCs would be beneficial to assess bone forming activity of SSCs.
8. Figure 6: Phospho-mass spectrometry data identifying phosphorylation sites Y173, S182, and S197 should be included in the figure. At least the phosphorylated peptide sequences should be provided.
9. Figure 6f and 6l are not convincing to show nuclear and cytosolic localization of Zfp260. Nuclear staining is needed. Additionally, it is difficult to see the increase of Zfp260 in cytosol when it decreases in nucleus. Nuclear/cytosolic fractionation assay would be beneficial.
10. Figure 7: Validation experiments showing the expression levels of Zfp260 WT and mutants in AAV-treated Zfp260 cKO mice are missing. AAV-mediated expression of a transgene under a short version of the CTSK promoter might be minimal.

Minor Concerns:

1. Key words: zfp260 and Ctsk need full names.
2. Figure 3m: Which tissues were used for mIHC?
3. No description of Figure 8 is included in text.
4. sFig. 8s should be changed to sFig7s.
5. sFig.5a-g: Remove a-g.
6. The subtitle, Deficiency of Zfp260 in PSCs resulted in decreased co-staining rates of Runx2 within the SSC hierarchy, should be rephrased. What is biological meaning of "co-staining rates"?
7. sFig.6a-g: Remove a-g.
8. The diagram showing the generation of Zfp260-floxed mice should be added to Supplementary Figure since they are a novel mouse line.
9. Sequences of qPCR primers, Chip-seq primers, the CTSK promoter of AAV (TSS + 1kb) should be moved to Supplementary Table.

Reviewer #3

(Remarks to the Author)

In this study, Yuteng Weng and coauthors investigated the molecular mechanisms underlying bone healing processes. They performed comprehensive transcriptome analysis on specific cell types, including SSCs and BCSPs, and identified Zfp260 as a key transcription factor in the transition from SSCs to BCSPs. Through genetic studies using a conditional gene knockout approach, they found that Zfp260 KO resulted in a significant delay in bone healing and impaired osteogenesis. Further epigenetic analysis showed that Zfp260 is necessary for active enhancer markers and open chromatin associated with osteogenesis. Additionally, they identified crucial phosphorylation sites in the ZFP260 protein. Overall, the study provided significant findings in bone biology with convincing data, although several points need to be addressed.

Comment 1:

In Figure 1c, the process used to identify candidate transcription factors was not clear. The authors need to describe the details of this process and provide a comprehensive list of all transcription factor candidates they identified.

Comment 2:

Although the bulk RNA-seq data clearly showed upregulation of Zfp260 during bone healing, the expression of Zfp260 was not clearly presented in the scRNA-seq analysis. The authors need to show the expression as a feature map in Figure 1i and Figure 2h, and as a violin plot in Supplementary Figures S2b and S3b.

Comment 3:

In Figure 2f, toluidine blue stains cartilage matrix. The results clearly indicate that chondrogenesis was severely delayed in Zfp260 knockout. Immunostaining for chondrogenic markers, such as Type II collagen and SOX9, and osteogenic markers, such as Type I collagen and Osterix, are required.

Comment 4:

Related to comment 3, if chondrogenic differentiation is affected by Zfp260, how do the authors interpret this data? They may need to discuss whether Zfp260 is crucial for both osteogenesis and chondrogenesis.

Comment 5:

In Figure 3j, what are the actual expression differences in Runx2 genes between wild-type and Zfp260 gene knockout? The authors need to show Runx2 expression between WT and KO in both BF and MSFL studies.

Comment 6:

In Figures 4f and 4g, the authors selectively showed enrichment of ATAC signals associated with specific terms. However, it

is important to present a more comprehensive view. What gene ontology terms were most enriched in the analysis of differentially accessible regions? Were there any upregulated ATAC signals? What do these regions represent?

Comment 7:

In the abstract, the authors wrote “Zfp260 drove chromatin opening”; however, there is no evidence that Zfp260 plays a role in opening the Runx2 regulatory regions. The authors only showed that Zfp260 was required for the maintenance of open chromatin states in these regions. The authors need to revise this statement.

Version 1:

Reviewer comments:

Reviewer #1

(Remarks to the Author)

The authors have thoroughly responded to the comments on the original submission, and the revised manuscript is felt to be technically solid and of interest for the field.

As a very minor point, it is suggested that this text change highlighted by the authors in the rebuttal letter be slightly modified as indicated:

“Previous research has demonstrated that periosteal skeletal stem cells (PSCs), a periosteal SSC population defined by the expression of Ctsk together as a primary biomarker together with additional cell surface markers located on the outer membrane of bone, were crucial for the healing of long bone fractures, facilitating both intramembranous and endochondral ossification processes.” It is felt to be useful to highlight that lineage reporters alone (e.g., CTSK-cre) cannot define these cells, but really the combination of the lineage reporter, surface markers and anatomic location. This was felt to be helpful to clarify as it is a frequent point of confusion in the field.

Reviewer #2

(Remarks to the Author)

All of the reviewer's comments were well responded along with additional new data. No more comments by the reviewer.

Reviewer #3

(Remarks to the Author)

The authors addressed the reviewer's comments and revised the manuscript properly. The reviewer accepts the revised manuscript.

Response to the referee comments on NCOMMS-24-37853-T-R1

We would like to thank all three referees for their kind reviews of our manuscript. We have carefully studied all comments from the reviewers, and performed new experiments/analyses. The following are our point-to-point responses:

Reviewer #1

In this manuscript by Weng et al., the authors perform single cell transcriptional profiling on CTSK-lineage SSCs in the setting of fracture healing and MSFL, identifying Zfp260 as a transcription factor associated with the early differentiation stages in this lineage. Subsequently, the phenotypic contribution of Zfp260 to this lineage, associated transcriptional/epigenetic regulation, and phosphoregulation are studied. The topic is of great interest overall, and the studies are largely seen as robust, though a few important points where additional data is needed are noted below. This manuscript has several areas of strength. One, the work here to try and identify the precise differentiation stage at which Zfp260 acts is notable and distinguishes the present work from many similar studies, though some additional experiments are suggested here to more strongly support this conclusion. Additionally, the in vivo rescue of CTSK-creER Zfp260 fl/fl mice by the AAV9-delivered Zfp260 mutants is considered impressive and beyond the typical scope/rigor of many similar studies in the literature. Overall, while there are some areas where further studies are needed to fully support the conclusions, there is enthusiasm for this study. Major points number 1, 2 and 7 below are considered the most critical areas where new data is needed.

Response:

We extend our sincere gratitude for your comprehensive review of our manuscript. We highly value your positive feedback and the professional, detailed guidance you have offered for our revisions. Your insights have significantly bolstered our confidence and motivation to pursue our research on Zfp260 with increased rigor. We have meticulously addressed your major and minor comments and have augmented our experiments accordingly. Please find our detailed point-by-point responses below for further clarification.

Major points:

1. Is there a lineage reporter in *CTSK-creERT2* mice? Both gating the flow analysis on the *CTSK*-lineage in the *Zfp260 fl/fl CTSK-creERT2* mice and analyzing the histologic distribution of the *CTSK* lineage in these same mice vs controls is important for clarifying the nature of the cellular defect and to ensure that the directly impacted lineage is analyzed separately from the broader pool of skeletal cells. Use of this lineage reporter would also enable demonstration of *Zfp260* deletion efficiency.

Response to Major point 1:

We express our gratitude for your thorough evaluation. During our phenotypic analysis, we encountered batch-related discrepancies, resulting in some mice possessing the lineage reporter gene while others did not. Consequently, the lineage reporter was omitted from the experimental flowchart depicted in Fig. 2a.

We have enhanced the imaging of fracture callus healing by incorporating reporter gene-labeled specimens stained with *Zfp260*. The results demonstrate that the *Ctsk* lineage is present in both intramembranous and endochondral ossification regions, corroborating the findings reported by Debnath et al. The staining analysis reveals that, within the trauma repair model, *Zfp260* frequently co-localizes with *Ctsk*-lineage cells. However, in *Zfp260*-deficient mice, there is a marked reduction in the co-expression of *Zfp260* and *Ctsk*-lineage cells, as illustrated below (Fig. R1-Ma1-1).

Fig. R1-Ma1-1 Ctsk-lineage cells in fracture healing and MSFL bone regeneration

In addition, we conducted an analysis of CTSK-lineage cells expressing the reporter gene and observed that most of the Lin⁻Ilgav⁺ cells were ZsGreen⁺, as shown below (Fig. R1-Ma1-2).

Fig. R1-Ma1-2 The percentage of Ilgav⁺ZsGreen⁺ and Ilgav⁺ZsGreen⁻ cells gated from single live Lin⁻ cells.

This finding suggests that the stem cells, progenitor cells, osteolineage cells, and stromal cells within the Lin⁻Ilgav⁺ population are predominantly derived from the Ctsk⁺ lineage. Subsequent analyses of these lineage cells produced results consistent with those presented in Figures 2h and 2j of the original manuscript, as illustrated below (Fig. R1-Ma1-3). Consequently, we have revised this section of the manuscript, as reflected in the R1-manuscript.

Fig. R1-Ma1-3 The SSC lineage clusters gated from Lin⁻ZsGreen⁺Ilgav⁺ cells of both models

2. The claim that there is a SSC to BCSP transition defect in the absence of *Zfp260* is currently only supported by indirect evidence. Alternatively, the data observed could be due to cell death in CTSK-lineage BCSPs or other mechanisms. To provide direct evidence of a SSC to BCSP transition defect, WT and *Zfp260*-deficient CTSK-lineage SSCs should be isolated, placed into a transplantation based organoid system in the kidney capsule/muscle/inguinal fat pad, and re-analyzed by flow to demonstrate the progress of cellular differentiation. Direct transplantation of WT and *Zfp260*-deficient CTSK-lineage BCSPs in the same system is also recommended to distinguish the SSC to BCSP versus post-BCSP defects present. If possible, uCT analysis of bone formed in this organoid system would be ideal and would help to establish the lineage intrinsic ossification defect associated with *Zfp260* deficiency, though in this setting the uCT data is less important than the flow cytometry-based cellular differentiation endpoints.

Response to Major point 2:

We express our gratitude for your insightful suggestions. As you have accurately identified, our existing evidence concerning the transition defect from SSC to BCSP is predominantly based on in vivo flow cytometry. The data obtained from in vitro staged osteogenic induction may not adequately substantiate this transition defect. In accordance with your recommendation, we have extended our study by conducting subcapsular renal implantation of wild-type and *Zfp260*-deficient CTSK-lineage SSC and BCSP cells, subsequently performing micro-CT and flow cytometry analyses on the resultant tissues.

The findings demonstrated that *Zfp260* knockout at the SSC stage resulted in impaired mineral nodule formation. Flow cytometry analysis further revealed an accumulation of pre-BCSP and a significant reduction in osteolineage cells. In contrast, *Zfp260* knockout at the BCSP stage led to a decrease in mineral nodule formation, accompanied by BCSP accumulation and a reduction in osteolineage cells. These results have been incorporated into Fig. 4m,n of the R1 manuscript, as shown below (Fig. R1-Ma2).

Fig. R1-Ma2 The subcapsular renal implantation of SSC and BCSP

3. It is suggested that the data in Fig 5 be confirmed by sorting of the relevant populations and Runx2 qPCR or qPCR for other Runx2 target genes in at least one of the models to confirm the co-staining results. It is very confusing to provide a "stem and pro" group alongside separate SSC, pre-BCSP, ect... recommend removing this.

Response to Major point 3:

Thank you very much for your valuable suggestions. Based on this and subsequent comments, we have removed the stem&pro, PCP, BLSP, and THY panels from Figure 5 and now only display SSC, pre-BCSP, BCSP, and osteolineage cells. Given the large number of mice required for the MSFL model, we sorted the SSC, pre-BCSP, BCSP, and osteolineage cells from the fracture group and tested Runx2 and its downstream target gene Sp7. We found that after Zfp260 knockout, Runx2 expression levels in both BCSP and osteolineage cells decreased significantly, with statistical significance. However, Sp7, a downstream target of Runx2, only showed downregulation at the osteolineage cell stage, indicating a later expression pattern, as shown below (Fig. R1-Ma3).

Fig. R1-Ma3 The expressions of Runx2 and Sp7 in FACS-sorted SSC lineage cells

4. Annotating a scRNA-seq cluster as representing an SSC is not straightforward. It is suggested to try and further build out the case for the proposed stem cell annotation. This can include analysis of a wider range of associated transcripts, like CD200, or cross-referencing broader transcriptional data in Bok et al. or Debnath et al. on CTSK-lineage stem cells. Similarly, much more detail is needed to justify the annotation of pre-BCSPs and BCSPs.

Response to Major point 4:

We sincerely appreciate your professional recommendation. In our investigation of key transcription factors involved in the early differentiation process from spermatogonial stem cells (SSC) to basal cell stem progenitors (BCSP), the precise definition of SSC, pre-BCSP, and BCSP is crucial, as it significantly influences the interpretation of our results. For the annotation of SSC, pre-BCSP, and BCSP in our scRNA-seq dataset, we

adhered to the criteria established by Chan and Debnath for flow cytometry marker combinations specific to SSC / progenitor stem cells (PSC).

1. Our focus was on the markers for SSC, pre-BCSP, and BCSP, with distinctions primarily observed in the combination of CD105 and CD200 (Fig. R1-Ma4-a).

2. Chan et al. conducted transcriptional profiling of various cell populations, identifying the secreted protein BMP7 as highly expressed during the SSC, pre-BCSP, and BCSP stages (Fig. R1-Ma4-b).

3. Subsequently, we annotated downstream functional cell stages by examining the sequential expression changes of stemness markers such as Zeb1 and Twist1 (Fig. R1-Ma4-c), osteogenic markers including Col1a2, Runx2, Sp7, and Alpl, as well as chondrogenic markers like Sox9, Col2a1, and Acan (Fig. R1-Ma4-d).

4. To ensure the accuracy of our cluster annotations, we validated them against the monocle2 trajectory results, confirming their alignment with the anticipated cell differentiation trajectory (Fig. R1-Ma4-e).

Take the four criteria for consideration, we made the final annotation of the dataset.

Fig. R1-Ma4 The basis for annotating SSC subclusters

5. It is not expected in the PAGA analysis that SSCs are separated from what are presumably their downstream derivatives including osteolineage cells. Additionally, is the sample expected to only contain CTSK-lineage intramembranous specialized cells? There is some question whether both endochondral and intramembranous competent SSCs are present in the specimen and whether this can be represented in the scRNA-seq analysis.

Response to Major point 5:

We express our gratitude for your perceptive inquiry. We share your interest in this topic and have similarly focused our attention on the issue you highlighted.

Partition-based Graph Abstraction (PAGA) is a trajectory analysis method that enhances a graph-based clustering approach by evaluating the connectivity between clusters. The PAGA algorithm generates a connectivity graph in which nodes represent cell clusters, and edges denote the connectivity between these clusters, determined by shared nearest neighbors. The strength of these connections is modulated by the transcriptional similarity between clusters as well as their spatial positioning within the UMAP embedding [PMID: 30890159].

In this study, the fracture scRNA-seq data underwent batch correction using CCA2, followed by integration analysis, resulting in a substantial number of clusters. This high level of complexity may have influenced the spatial distribution of stem/progenitor cells and mature functional cells within the UMAP, causing these groups to appear more distant from one another. Consequently, the observed spatial separation may have led to PAGA's inability to detect connectivity between these populations, despite their underlying biological relationship. This issue arises when the clusters exceed PAGA's threshold for connectivity identification, as PAGA assesses the extent of shared nearest neighbors between clusters, and substantial spatial separation can attenuate this signal.

Nonetheless, it is crucial to acknowledge that dependence on a singular computational analysis method does not yield a comprehensive understanding. In addition to the PAGA results, we also conducted a trajectory analysis using Monocle, as detailed in the manuscript, to elucidate the differentiation characteristics of SSC, pre-BCSP, and BCSP as the initial subpopulations. The trajectory analysis performed with Monocle enabled us to map continuous differentiation pathways, thereby confirming that these groups are integral components of the same differentiation trajectory.

Furthermore, as demonstrated by the findings of Sivaraj et al. (begin at dpMSCs1 and end at OBs) [PMID: 35091558] and our prior publication in *Cell Research* (begin at Op-1 and end at Oc) [PMID: 35821090], cell populations that manifest as distinct clusters on UMAP plots may still be interconnected as upstream and downstream derivatives within the same differentiation hierarchy (as illustrated below, Fig. R1-Ma5). Consequently, the absence of direct connectivity in PAGA analysis does not necessarily refute the biological connectivity identified through other methodologies.

Figure 2b, d, e Sivaraj et. al Nat Commun. 2022 PMID: 35091558

Figure 2g, h Weng et. al Cell Res. 2022 PMID: 35821090

Fig. R1-Ma5-1 The UMAP plots illustrating separation between the initiating and subsequent clusters

Regarding your second question:

I believe what you are referring to are progenitors capable of endochondral and intramembranous ossification. In our scRNA-seq analysis, we performed clustering of Lin⁻ cells utilizing osteogenic (e.g, Alp) and chondrogenic markers (e.g, Acan, Col2a1, Sox9). By categorizing subpopulations based on the expression gradients of these markers, ranging from low to high, we were able to delineate distinct stages of cellular differentiation. These stages include osteolineage cells, and osteochondro-lineage cells. Notably, all identified subpopulations express Ctsk, suggesting their affiliation with the Ctsk lineage (as shown below, Fig. R1-Ma5-2). Consequently, we believe that our integrated dataset encompasses progenitor cells with the potential to facilitate both endochondral and intramembranous ossification.

Fig. R1-Ma5-2 ScRNA-seq dataset of fracture healing encompasses progenitor cells with the potential to facilitate both endochondral and intramembranous ossification

6. What is the basis to consider "PCP" as separate from BCSPs and "BLSP" cells as separate from THY+ cells? Having a strong justification for the cell types considered here is considered important and the evidence that each of these populations are biologically distinct may not be fully established. It is likely beyond the scope of this study to establish such evidence here, but additional consideration of these elements of the classification schema used would strengthen the study.

Response to Major point 6:

We sincerely appreciate your insightful comments. Our methodology for differentiating the previously mentioned cell populations is indeed grounded in the 2015 study by Chan et al. Their research indicates that the PCP, BLSP, and THY subpopulations are derivatives of BCSP, signifying mature functional cells. We acknowledge your observation that the biological evidence supporting the complete independence of these populations may not be entirely conclusive at this time. We agree that additional in vivo evidence is necessary to substantiate the independent characteristics of these subpopulations. To date, the primary method for differentiating these cell populations has been through studies employing flow cytometry marker combinations to distinguish the populations, alongside fluorescence-activated cell sorting (FACS) combined with RNA sequencing (RNA-seq) to evaluate transcriptomic differences between the groups.

Our findings indicate that Zfp260 predominantly regulates the transition from spermatogonial stem cells (SSC) to basal cell spermatogonia (BCSP). However, while Zfp260 does play a role in the subsequent differentiation of BCSP, it is not a decisive factor. Consequently, the objective of our study is not to investigate the specific role of

Zfp260 within any particular subpopulation downstream of BCSP, assuming these subpopulations exist independently. Instead, we aim to categorize them collectively as osteolineage cells in the phased ATAC-seq analysis.

In the manuscript, the PCP, BLSP, and THY subpopulations are depicted in Figures 2, 4, and 7. This inclusion is based on our analysis of the three cell stages—SSC, pre-BCSP, and BCSP—utilizing flow cytometry and multicolor immunohistochemistry. The markers employed for examining these subpopulations were consistent with those required to delineate the SSC to BCSP lineage. Consequently, we presented the proportions of PCP, BLSP, and THY in Figures 2 and 7, along with the percentage of Runx2⁺ cells within these subpopulations in Figure 4. However, these classifications were not the primary focus of our study; consequently, these subpopulations were predominantly categorized as osteolineage cells within the manuscript.

Current biological evidence does not rigorously differentiate the upstream-downstream cascade of skeletal stem cell (SSC) lineage from an in vivo experimental standpoint. Furthermore, this study does not aim to elucidate the distinctions among these cell populations, as this issue lies beyond the scope of the present research. In future research, the application of more advanced methodologies may provide further insights into the differentiation sequence of SSC lineage cells. However, in the present study, and in accordance with your professional recommendations, **we have opted to standardize the classifications of PCP, BLSP, and THY subpopulations as osteolineage cells in Figures 2, 4, and 7.** This decision aims to mitigate potential controversies related to the classification scheme.

7. The baseline phenotype of Zfp260 deletion in CTSK-lineage cells should be established outside of the fracture and MSFL models.

Response to Major point 7:

We sincerely appreciate your professional recommendation. Regarding the developmental phenotype of Zfp260 deletion in the Ctsk-Cre lineage, as you mentioned, we have added images from micro-CT and histological analyses of 6-week-old mice, including body size, skull, and femur. Our observations indicated that *Ctsk^{cre};Zfp260^{fl/fl}* mice exhibit a reduction in body length and diminished skeletal mineralization level, as evidenced by Alizarin Red and Alcian Blue staining of the skeleton (Fig. R1-Ma7-a,b). There was a notable decrease in the calcification rate in both the femur and nasal bone (Fig. R1-Ma7-c). Micro-CT and histological analyses reveal significant cortical bone thinning at the femoral midshaft and metaphysis, accompanied by a reduction in trabecular bone (Fig. R1-Ma7-d,f,g). Additionally, the craniofacial bones demonstrated

decreased bone density, with a particularly pronounced thinning of the nasal bone cortex (Fig. R1-Ma7-e,h). These alterations are observed on the bone surface, aligning with the periosteal distribution pattern of CTSK-lineage cells. This pattern is analogous to the phenotypic changes initially reported by Debnath et al. [PMID: 30250253] upon the knockout of *Osx* in CTSK-Cre lineage mice. The corresponding figure is shown below.

Fig. R1-Ma7 The basal skeletal phenotypes of *Ctsk^{cre};Zfp260^{fl/fl}* mice

Minor points:

1. Line 47-48, typos are present.

Response to Minor point 1:

We express our gratitude for your correction. The necessary revisions have been incorporated into the R1-manuscript.

2. PAGA and trajectory analysis don't really confirm cluster annotation accuracy as claimed on line 127.

Response to Minor point 2:

We express our gratitude for your correction. We have removed the sentence and replaced it with the following: **"To explain the connectivity between cell populations and reveal the pathways of cell differentiation."**

3. Line 131 and 135, how can it be claimed that ZFP260 is upregulated in the SSC to BSCP transition when BCSPs are not annotated in the scRNA dataset?

Response to Minor point 3:

We express our gratitude for your insightful comment. The upregulation of Zfp260 during the transition from SSC to BCSP was elucidated by mapping the expression levels of Zfp260 across various clusters and aligning them in accordance with the differentiation sequence. In the single-cell RNA sequencing (scRNA-seq) dataset pertaining to fractures, we have identified and annotated the BCSP cluster, as illustrated in the accompanying figure, highlighted via scarlet arrow and dotted box.

In the MSFL model, to ensure continuity in our research, we directly adopted the UMAP cluster definitions and presentation from our previous research in *Cell Research* [PMID: 35821090]. In our previous study, we focused on identifying the initial subpopulation responsible for maxillofacial bone regeneration using the MSFL model, and we used Krt14 and Ctsk dual markers to label this group. In the aforementioned study, our primary objective was to investigate the osteogenic initiation properties of the group, without addressing its classification as either stem cells — characterized by trilineage differentiation potential and colony-forming ability — or progenitors. Consequently, we initially adopted a conservative stance, categorizing the group as progenitors. However, as our research advanced, we discovered that Krt14⁺Ctsk⁺ cells demonstrate stem cell properties, including trilineage differentiation and colony-forming unit (CFU) characteristics, as illustrated in the figure below (Fig. R1-Mi3-a-e). Moreover, an in-depth

analysis of the transcriptome data from the previously FACS-sorted ZsGreen⁺tdTomato⁺ cells (identified as Krt14⁺Ctsk⁺ cells) revealed that this cell population shares marker combinations akin to the SSC signatures reported by Chan et al., as shown below (Fig. R1-Mi3-f).

Fig. R1-Mi3 The skeletal stem cell identity of Krt14⁺Ctsk⁺ cells

Consequently, the Op-1 cluster, characterized by Krt14⁺Ctsk⁺, can be said to exhibit prominent SSC characteristics. Therefore, within the context of the MSFL dataset, differentiation may be considered to commence from the SSC stage at Op-1. By preserving the progenitor status of the Op-2 and Op-3 clusters, we can objectively determine that Zfp260 is upregulated during the differentiation process from SSC to progenitor. Nonetheless, to ensure consistency with prior research, we have retained the original cluster annotations as reported in *Cell Research* for this study. Should you deem revisions necessary, we would follow your expert recommendations for future adjustments.

4. Line 142, recommend not claiming that Zfp260 is the sole upregulated transcription factor or at least slightly qualifying this, as the sparse transcriptional sampling in scRNA-seq means that many relevant transcription factors could be missed.

Response to Minor point 4:

We sincerely appreciate your professional recommendation. Accordingly, we have revised the description from the “sole upregulated transcription factor” to “Zfp260 was the TF significantly upregulated from SSC to BCSP among the five candidates” in both Result and Discussion sections and highlighted it in yellow.

5. As more of a comment to consider in the writing than an issue to be addressed experimentally, the initial Chan reports focus much more on establishing SSC identity than the downstream differentiation sequence. While the report here is an accurate representation of the Chan differentiation sequence, it is worth keeping in mind that this topic of whether pre-BSCPs truly turn into BSCPs than in turn turn into THY1⁺ and 6C3⁺

cells is an area that has had only minimal dedicated study and may therefore be subject to future revisions. Some slight additional qualification along these lines is appropriate.

Response to Minor point 5:

We sincerely appreciate your professional recommendation. Chan et al.'s study predominantly aims to elucidate the identity and marker combinations of SSC. While their experiments involving renal subcapsular transplantation and flow sorting have demonstrated that BCSP is downstream of SSC, additional evidence is required to ascertain whether THY1⁺ and 6C3⁺ cells are unequivocally downstream of BCSP. Consequently, we have intentionally minimized the use of upstream-downstream terminology in the manuscript and have included corresponding explanations and revisions in the discussion section of R1 manuscript. The updated text is presented below:

"Although the unidirectional cascade differentiation sequence had been established and demonstrated through in vitro methodologies, there remained insufficient robust evidence to confirm its stable existence *in vivo*. Specifically, the extent to which this differentiation sequence downstream of SSCs adhered to the same strict unidirectional cascade *in vivo* had not been thoroughly investigated."

6. It is worth clarifying (such as line 162) that CTSK alone does not specify PSCs, but rather CTSK in concert with additional cell surface markers and ideally anatomic location.

Response to Minor point 6:

We sincerely appreciate your expert recommendations. In response, we have revised the Results section and highlighted the modifications in the R1-manuscript. The updated text is presented below:

"Previous research has demonstrated that periosteal stromal cells (PSCs), a significant SSC population featured by the expression of Ctsk as a primary biomarker and located on the outer membrane of bone, were crucial for the healing of long bone fractures, facilitating both intramembranous and endochondral ossification processes."

7. Line 176, suggest more clearly spelling out how intramembranous and endochondral regions of the callus are discriminated as this is often not trivial. Keep in mind that CTSK-lineage PSCs gain endochondral bone formation capacity in response to fracture, so endochondral defects might be expected with severe functional defects in this lineage.

Response to Minor point 7:

We sincerely appreciate your insightful suggestion. It is indeed challenging to fully differentiate between the regions of intramembranous and endochondral ossification. To address this, we performed co-staining using the chondrogenic marker (Col2) and the osteogenic marker (Col1) to partially distinguish these two regions, as illustrated in the figure below. Additionally, we have incorporated this data into Figure 2i of the R1 manuscript, as shown below (Fig. R1-Mi7).

Fig. R1-Mi7 The distributions of Col-I and Col-II in the EO or IO region during fracture healing

Corresponding modifications highlighting the phenotypic changes in the endochondral ossification region have also been made to the description of the results for Figure 2b on line 176, with these changes highlighted in yellow in the text. The modified text is shown below:

"In the experimental group, a significantly reduced level of mineralization was observed in the region of endochondral ossification, accompanied by an absence of significant indicators of intramembranous ossification between 14 and 28 dpf, as illustrated in Fig. 2b."

In addition, concerning the PSC of the CTSK lineage referenced, we have briefly addressed in the discussion section of the original manuscript the capacity of the CTSK lineage to undergo endochondral ossification following fracture. In the discussion section of the R1 manuscript, we have further elaborated on this assertion and emphasized it by highlighting it in yellow within the text. The revised text is presented below:

"Notably, the deletion of Zfp260 in Ctsk-expressing cells also exerted a significant reduction on the formation and mineralization of the cartilage matrix within the callus."

According to the findings of Debnath et al., Ctsk lineage cells played a regenerative role via both intramembranous and endochondral ossification. Therefore, it was hypothesized that the phenotypic differences observed in the endochondral ossification region within the fracture model may be attributed to the loss of Zfp260 function in the subset of Ctsk lineage cells responsible for endochondral ossification. To focus on the the influence of Zfp260 on the endochondral ossification process, applications of transgenic mice with Col2a or Sox9 promoter-driven Cre recombinase for the knockout of Zfp260 should be carried out^{29,30}.

8. Fig 2d--what is "dpm"? Fracture BMD can be tricky to interpret. Was the uCT used calibrated with phantoms for this purpose? The fracture/MSFL BMD data may be more appropriate in the extended data.

Response to Minor point 8:

Thank you for your insightful inquiry. The abbreviation "dpm" refers to "days post MSFL". As referenced in other studies [PMID: 34173256; PMID: 35716916], they utilized the term "dpf" to indicate days post fracture. Regarding the analysis of bone mineral density (BMD) at the fracture site, we modified the threshold when defining the region of interest (ROI) to include both mineralized and unmineralized areas. This adjustment was made to enable the precise delineation of callus boundaries. Additionally, we excluded the diaphysis region of femur by implementing a counterclockwise gating technique. In our study, we delineated the complete callus region, encompassing both endochondral and intramembranous ossification regions. The mean bone mineral density (BMD) of this delineated area was utilized as the final measurement outcome. An illustrative example of the reconstructed image is provided below (Fig. R1-Mi8).

Fig. R1-Mi8 The reconstruction image of fracture callus (ROI)

Concerning the micro-CT calibration process referenced, we perform calibration using a standard tube prior to each scan to ensure the measurements remain within the established standard range.

The calibration procedure employed is detailed as follows: Three voltage gradients—50 kV, 70 kV, and 90 kV—are utilized for the density correction of the standard tube sample. This ensures that the density standard value at 70 kV, which is commonly used for skeletal scans, is maintained within the range of 778 to 802 (790±12). Furthermore, geometric correction is conducted between each pair of scans to ensure the standard value remains within the range of 0.111 to 0.141 (0.126±0.015).

Additionally, the bone mineral density (BMD) data pertinent to fractures/MSFL has been incorporated into the Source data table.

9. Why was phospho-regulation of Zfp260 expected? The logical flow of the manuscript at this point is a bit weaker, and "transcription factors are often phosphorylated" is not the strongest rationale to justify this line of investigation. It should be better explained in the text that Prkca was identified by an IP mass spec approach in Fig 6a, b.

Response to Minor point 9:

We sincerely appreciate your expert recommendations. In response, we have revised the Results section and highlighted the modifications in the R1-manuscript. The updated text is presented below:

"Modifications of transcription factors frequently served as a regulatory mechanism for their translocation and function. Through GST pull-down and mass spectrometry analyses (Fig. 6a), we identified the enzymes interacting with Zfp260. Among these, Prkca was found to be the most enriched, as detailed in Fig. 6b."

10. It should be clarified how sufficient numbers of PSCs were obtained for the pulldown study in Fig 6, typically obtainable cell numbers are far too small for biochemical studies. Was in vitro expansion used and what were the conditions and duration of such expansion? Recommend putting this in the Fig legend.

Response to Minor point 10:

We greatly appreciate your meticulous attention to the details of our study. Given the substantial number of PSCs necessary for the IP, we utilized an immortalization-based amplification approach. Specifically, we selected *Ctsk^{creER};H11^{RSR-tdTomato-LSL-ZsGreen}* mice, for preparing PSCs, and *Ctsk^{creER};H11^{RSR-tdTomato-LSL-ZsGreen};Zfp260^{fl/fl}* mice, for preparing PSC *Zfp260^{-/-}*. We conducted a three-day tamoxifen labeling protocol one week prior to the FACS.

Utilizing the marker combination Lin⁻Itgav⁺ZsGreen⁺THY1⁻6C3⁻CD105⁻CD200⁺, we isolated PSCs from the long bones and craniofacial bones of mice. These cells were

subsequently co-cultured in vitro. After a two-week culture period, all clones were selected and passaged to P1. This was followed by the overexpression of hTERT via lentiviral infection to achieve semi-immortalization. The cells were then further expanded to establish a cell line suitable for protein pull-down and ChIP experiments. Due to the necessary quantity of cells, only these two experiments utilized immortalized cell lines.

We have updated the treatment conditions in the Materials and Methods section and revised the specific descriptions in the corresponding legend accordingly and marked them in yellow.

11. Line 352--it is questionable whether this phenotype is best described as an osteogenesis imperfecta phenotype. Perhaps just "impaired fracture healing" is more appropriate. Recommend rephrasing the section title in more positive terms: "Zfp260 phosphorylation sites are required for..."

Response to Minor point 11:

We express our sincere gratitude for your expert and comprehensive revision guidance. In response, we have amended the title of this section within the Results part to "Zfp260 phosphorylation sites are essential for both intramembranous and endochondral ossification processes in vivo," and have highlighted it in yellow in the R1 manuscript.

Furthermore, we have revised the concluding sentence, which initially referenced the "osteogenesis imperfecta phenotype," to "In conclusion, the introduction of Zfp260 mutants did not ameliorate the impaired bone healing phenotype associated with Zfp260 deficiency in PSCs" and highlighted it in yellow in the R1 manuscript.

12. Line 430 should be Chan et al., not Charles et al. This discussion is where some slight qualification that the differentiation sequence downstream of SSCs is not well studied overall is suggested.

Response to Minor point 12:

We express our sincere gratitude for your meticulous identification of our typographical error and for offering professional guidance on its revision. In accordance with your suggestions, we have amended this portion of the discussion and incorporated your comments into the paragraph, which is highlighted in yellow. The revised section is presented as follows:

"Chan et al.¹ performed a comprehensive analysis of surface marker combinations and, utilizing a series of FACS-RNA sequencing and renal subcapsular transplantation experiments, delineated the differentiation stages of the skeletal stem cell (SSC) lineage,

endeavoring to map their differentiation sequence. Chan et al. illustrated a progression from skeletal stem cells (SSCs) to bone, cartilage, and stromal progenitors (BCSPs), ultimately leading to the differentiation into osteolineage/osteochondro-lineage or stromal lineage, as evidenced by extensive post-sorting *in vitro* data. **Although the unidirectional cascade differentiation sequence had been established and demonstrated through *in vitro* methodologies, there remained insufficient robust evidence to confirm its stable existence *in vivo*. Specifically, the extent to which this differentiation sequence downstream of SSCs adhered to the same strict unidirectional cascade *in vivo* had not been thoroughly investigated.”**

13. *Is Zfp260 specific for CTSK-lineage stem cells or is it more widely expressed during the differentiation of non-CTSK lineage stem cells?*

Response to Minor point 13:

Thank you for your inquiry. Our observations indicate that Zfp260 is extensively expressed in cells outside the CTSK lineage. Initially, scRNA-seq data suggest that Zfp260 expression may overlap with *lepr*-lineage and *grem1*-lineage cells in the fracture dataset, **as demonstrated below (Fig. R1-Mi13-1).**

Fig. R1-Mi13-1 Zfp260 shows wider expressing pattern in non-CTSK lineage stem cells

Secondly, we conducted co-staining of Collagen Type I and Zfp260 in mice at embryonic day E16.5, postnatal day 3 (P3), and postnatal day 7 (P7) to investigate the expression dynamics of Zfp260 during early skeletal development under physiological conditions. Our findings indicate that in long bones, Zfp260 is expressed not only in the periosteal tissue (magnified region 4), but also in the growth plate chondrocytes (magnified region 2), trabecular surfaces of the metaphysis, along with surrounding bone marrow stromal cells (Col-1⁺), are highlighted in the magnified region 3. As development advances to postnatal day 7 (P7), chondrocytes within the secondary ossification center progressively commence the expression of Zfp260, as depicted in magnified region 1. In the nasal bone tissue, both nasal chondrocytes and cells of the periosteal tissue exhibit expression of Zfp260, **as shown below (Fig. R1-Mi13-2).**

**Fig. R1-Mi13-2 Zfp260's expression during early skeletal development
(Scale Bar = 50µm)**

Based on its distribution, it is reasonable to infer that the expression of Zfp260 is associated not only with the development of the periosteal stem cell lineage but also with the regulation of mesenchymal stromal cell development and regeneration.

Reviewer #2

In the manuscript, “Zfp260 choreographs the early stage osteo-lineage commitment of skeletal stem cells”, Weng et al identified Zinc Finger Protein 260 (Zfp260) as a key driving factor of skeletal stem cells (SSCs) to bone, cartilage and stromal progenitors (BCSPs) maturation and osteogenic fate determination during skeletal injury. The deletion of Zfp260 in periosteal stem cells using CTSKCRE line arrests the transition of pre-BCSP to PCSP stage, resulting in impaired bone fracture healing and regeneration. The authors demonstrated that during the early stages of skeletal injury repair, Zfp260 expression is upregulated and that it regulates osteogenic gene expression via formation of the super-enhancer complexes. They also identified PKCa as an upstream kinase of Zfp260 and demonstrated that PKC-mediated phosphorylation is important for biological function of Zfp260. Overall, novel roles of Zfp260 in osteogenic differentiation of SSCs, the soundness of the experimental approach, and potential in bone fracture therapy are appealing for publication in the Nature Communications. However, experiments are limited to understand Zfp260’s roles in periosteal stem cell development during the early stages of bone fracture repair and regeneration. Additionally, the roles of Zfp260 in osteoclast development due to CTSKCRE-mediated deletion in osteoclasts and in general skeletal stem cell populations under physiological conditions are largely missing. Finally, there is no evidence showing Zfp260 as a transcription factor.

We express our sincere gratitude for your thorough review of our manuscript and for offering professional and detailed guidance for our revisions. We acknowledge that, although you have provided a generally positive evaluation of our study, you have also identified several issues that necessitate further clarification.

Summary point 1

“Experiments are limited to understand Zfp260’s roles in periosteal stem cell development during the early stages of bone fracture repair and regeneration”

Response to Summary point 1:

We appreciate your objective comments. As researchers with a background in clinical medicine, we are keen to investigate potential target molecules that may contribute to suboptimal clinical bone healing/regeneration and to develop targeted therapeutic strategies based on these insights, which exert significant implications for clinical treatment approaches. Utilizing widely recognized clinical post-traumatic healing models, including fracture and MSFL, we identified the pivotal role of Zfp260 and elucidated its

mechanism of action. Furthermore, we examined the developmental impact of Zfp260 on CTSK-lineage skeletal stem cells, noting its influence on this cell population and skeletal development under physiological conditions. Although these findings may have implications for hereditary bone diseases in children and adolescents, this was not the primary focus of our study. Furthermore, to preserve the coherence and flow of the manuscript, data regarding the impact of Zfp260 on CTSK-lineage cells under physiological conditions has been excluded from the figures. Nonetheless, we intend to present these findings comprehensively in our point-by-point response.

Summary point 2

“Additionally, the roles of Zfp260 in osteoclast development due to CTSKCRE-mediated deletion in osteoclasts and in general skeletal stem cell populations under physiological conditions are largely missing.”

Response to Summary point 2:

We greatly appreciate your meticulous and thoughtful review. It is noteworthy that CTSK-lineage cells not only identify periosteum-derived skeletal stem cells but also act as crucial markers for osteoclasts. The concern you have raised has been anticipated, and through a range of experimental methodologies, we have demonstrated that the influence of Zfp260 on osteoclasts is minimal. We will furnish a comprehensive report on these findings in our forthcoming detailed response.

Summary point 3

“Finally, there is no evidence showing Zfp260 as a transcription factor.”

Response to Summary point 3:

We appreciate your insightful comments. Notably, Zfp260 was initially identified within the realm of cardiovascular research, where numerous studies have established its role as a pivotal transcription factor in the regulation of cardiac hypertrophy. Utilizing luciferase reporter assays, researchers have demonstrated that it is responsive to alpha-adrenergic signal stimulation and modulates the transcription of various downstream target genes, including Nppa and Mmp9 [PMID: 21051538; PMID: 16166646; PMID: 27826738].

Our study is the first to elucidate the regulatory role of Zfp260 in skeletal stem cells, revealing that Zfp260 can influence the downstream transcription of established transcription factors such as Runx2 and Zfhx4 through the formation of super-enhancers. Nonetheless, this finding does not preclude the possibility that Zfp260 functions as a

transcription factor. The data obtained from H3K4me1, H3K27ac, and ATAC-seq analyses reveal an extensive peak platform exceeding 5 kb upstream of the transcription start sites (TSS) of Runx2 and Zfhx4. Concurrently, ChIP-seq results demonstrate a substantial overlap between Zfp260 binding peaks and this platform. These findings not only corroborate the pivotal regulatory roles of Runx2 and Zfhx4 in skeletal stem cells, as documented in prior studies, but also furnish evidence supporting the role of Zfp260 as a chromatin remodeling factor.

Furthermore, our observations indicate that Zfp260 demonstrates narrow binding peaks within the promoter regions of several pro-osteogenic genes, implying that it shares characteristics with general transcription factors. This finding was corroborated through additional experiments. A comprehensive report detailing these findings will be included in our subsequent point-by-point response.

Major points:

1. The authors need to clarify Zfp260 as a chromatin remodeling factor, not a transcription factor. There is no evidence supporting the biological functions of Zfp260 as a transcription factor.

Response to major point 1:

We express our gratitude for your expert guidance. As outlined in the summary section, Zfp260 was initially identified within the domain of cardiovascular research, where numerous studies have elucidated its function as a pivotal transcription factor in the regulation of cardiac hypertrophy. Utilizing a luciferase reporter assay, researchers have demonstrated that Zfp260 is responsive to alpha-adrenergic signal stimulation and modulates the transcription of several downstream target genes, including the previously identified Nppa and Mmp9 [PMID: 21051538; PMID: 16166646; PMID: 27826738]. Collectively, these findings indicate that Zfp260 performs characteristic functions of a transcription factor.

In the domain of bone injury repair, our investigation centered on the regulatory role of Zfp260 within skeletal stem cells, revealing its capacity to modulate downstream transcription through the formation of super-enhancer complex, which transcribes prominent transcription factors such as Runx2 and Zfhx4. This study specifically elucidates Zfp260's regulatory influence on Runx2, highlighting its involvement in super-enhancer formation, as evidenced by (1) the distinct sequential expression patterns exhibited by Zfp260 and Runx2; (2) the integration of H3K4me1 and H3K27ac markers with ATAC-seq data reveals a broad peak platform extending over 5 kb upstream of the

transcription start sites (TSS) of Runx2 and Zfhx4. Concurrently, ChIP-seq analysis demonstrates that the binding peaks of Zfp260 significantly overlap with this platform. This observation not only corroborates the pivotal regulatory roles of Runx2 and Zfhx4 in skeletal stem cells, as documented in prior studies, but also substantiates the role of Zfp260 as a chromatin remodeling factor.

Nevertheless, this finding does not suggest that Zfp260 is devoid of the typical biological functions associated with transcription factors. Our observations indicate that Zfp260 demonstrates narrow binding peaks within the promoter regions of several pro-osteogenic genes (see Supplementary Figure 5a-c; Figure R2-Ma1-a, b, c), including *Ctnnb1*, *Smad5*, and *Gli1*, which implies that it shares common features with general transcription factors. Furthermore, we developed reporter vectors incorporating specific promoter regions of these genes and performed dual-luciferase reporter assays. Our results corroborate that Zfp260 possesses the capacity to directly initiate the transcription of these genes (Fig. R2-Ma1-d).

Fig. R2-Ma1 The characteristics of Zfp260 as a transcription factor

Building upon prior research, our findings, and your insightful feedback, we will refine our discussion to **underscore the role of Zfp260 as a transcription factor with chromatin regulatory functions.**

The following is the sequence used to construct the reporter vector:

Ctnnb1-reporter: sequence of TSS+300bp

agtttccaatggagcggcgtctggacgccccgccccggctcggcccctcccggcggcgcacgcgcagagcgactcgga

ctggcggggactactttccccggcccggcccgcgccccgcccctcgtcctcgcgcggcggaacgctccgcgcggagcggc
agcggcaggatacacgggtccgcgcccgttataaatcgtccttgtgcgggcgcacatctaagccctcgcctcgggtggcggcc
cgtcagctcgtgtcctgtgaagcccgcggcccggggaggcggagacggagcacggtgggcg

Smad5-reporter: sequence of TSS+300bp to TSS+800bp

gtttcgtataaggcgggtgagcttggttaaccaagtgccacttcaggctcgtcgcggacagcagaagccaggcctcgg
ccactcccagcgcgatgcgccggccgcccggccctccgccgccccccccctcccctcatcatccgggtccccggc
gagcggcgccgagcgttgtcccggggccgagctgtaataaagtgcggcgcgtgcacagcgcggcgcagcggcgtga
ggagagcgcgcctgggcccggggaggtagtgagggggcccaggcggcgcctcggggcccggcggaggggacaa
gcgccggcggcagcggcccgcgtgaggctgaggcctagaggctcccacgcgggacctgacggcacgggacgggg
ctccgcgcagcgcgggaggccccggtgtaaggaggccccgcgcccggcagcagggccggcggcagcagggccgctg
ccacctcggcgcgccaccgacgcccgggcccgcg

Gli1-reporter: sequence of TSS+1000bp

ggcctcgaactcagaaatccgcctgcctcctcccagtgctgggattaaaggtgatgccaccactgcttgcccatctt
aatttaaaaaacaaaaacaaacaaaaacacttactttgcagagccaagaaggtggttaaaaaacagcgcctatgg
cctgagagatcgtttaagggaagactctctccccaccgccatgctttgagctttctcaggtgggaaactgaaataga
ggccctctctgtgtccagtcaccatggtaacacagattcaagagacagggtactcccctatcacgaactaccaaggg
gctcgaatctccctggcgtgcccttctgtttcttgagctcacgcagctccctgggcatccgattgggctttgggacaggaa
cctgtcaaatttcctaagttgccagtgcaatcattccactccgaggggtgccatgtgctaggacaccagcagagggcgc
gaaaagcctaaagagagaggtgtggagggtggcaactgcgtccagaatttgagatggagaacggagtttctccagaga
ggtactctacagagcaccactgtcctcagtgatctgtaatgcacattctgcacttagaactctagcccgtttcttctctat
gttccataggctgcaccccctcccctctctacatcagtcaacccaacttcttctccaaacccaactctccaaatgctccac
gttccatccaaagagtgaggctggcctacagaccacgacatggggaaagctcagttacgaccaatgaagccgaagacc
gaggcttctggttaacttctgggtgagctttccccctcccctcccgcgcaatatttctcaatgtttctcgtcctggttc
agagcctctcgtggacagccgatccagggatccgcgcccgggattactgtccaccaagagcagcgtgtcgggcccggg
ggggggt

2. The data clearly demonstrated Zfp260's roles in the transition between pre-BCSP and BCSP. This should be clarified throughout the manuscript.

Response to major point 2:

We express our sincere gratitude for your expert advice and guidance. The pre-BCSP stage was initially defined and reported by Chan et al [PMID: 25594184]. In their seminal study, they performed RNA sequencing and functional assessments across each stage of the SSC lineage. Their findings demonstrated that SSC and pre-BCSP stages did not show significant differences at the functional or transcriptional levels. The primary

distinction between these stages was the presence or absence of CD200⁺ in the FACS surface marker. Consequently, they proposed that SSC and pre-BCSP constitute a transitional phase within the SSC category, collectively termed phenotypic-SSC (p-SSC).

Given that our research indicates a potential role of Zfp260 in maintaining ATAC-seq signal accessibility within SSC lineage cells, we performed ATAC-seq analysis across various stages of the SSC lineage. Our findings revealed that pre-BCSP demonstrated a greater degree of openness in specific ATAC-seq signals relative to CD200⁺ SSC. Nonetheless, this increased openness has not yet been extensively manifested at the transcriptional level. In light of your professional feedback and the comments from other reviewers, it appears that relying exclusively on ATAC-seq evidence to definitively differentiate between SSC and pre-BCSP as distinct populations or states necessitates further investigation. Consequently, to maintain rigor in our presentation of the results, we adopt a broader definition of SSC in this study, which includes the pre-BCSP state, referred to here as phenotypic SSC. And we made a notification in the Discussion part and highlighted it in yellow.

“Through screening and identifying TFs involved in the early stages of long bone fracture repair and craniofacial bone regeneration in mouse MSFL models, we have identified Zfp260 as a critical regulator of SSC differentiation into BCSP during bone repair and regeneration. Deletion of Zfp260 in Ctsk-labeled SSCs resulted in compromised fracture healing and MSFL bone regeneration, characterized by an accumulation of pre-BCSP (defined as phenotypic SSC).”

3. Since CTSKCRE line is also used to delete osteoclast genes and osteoclasts are important for bone fracture healing/regeneration processes, Zfp260's roles in osteoclast differentiation and bone resorption should be examined.

Response to major point 3:

We sincerely appreciate your comprehensive and insightful evaluation. It is important to note that CTSK-lineage cells not only identify periosteum-derived skeletal stem cells but also function as a crucial marker for osteoclasts. The issue you mentioned has been previously addressed in our earlier research. Through various experimental methodologies, we have established that the influence of Zfp260 on osteoclasts is minimal.

Osteoclasts originate from monocytes/macrophages. Our initial investigation aimed to determine whether osteoclasts express Zfp260. Analysis of single-cell sequencing data revealed that Zfp260 expression within the monocyte/macrophage lineage is exceedingly limited (Fig. R2-Ma3-a, b). In comparison to PSCs, the expression levels of Zfp260 in the

Raw264.7 monocyte-macrophage cell line, as well as in osteoclasts derived from Raw264.7 cells (induced with RANKL at 30 ng/ml for 7 days), were markedly low (Fig. R2-Ma3-c). Additionally, we conducted co-staining for Zfp260 and Tartrate-Resistant Acid Phosphatase (TRAP) on the long bones of postnatal day 14 (P14) wild-type mice, with a specific focus on the trabecular region. We did not observe significant colocalization of Zfp260 with TRAP signals (Fig. R2-Ma3-d).

Given that *Ctsk* is a marker for both osteoclasts and periosteal stem cells, we employed *Lyz2^{cre}; Zfp260^{fl/fl}* mice, previously bred for this purpose, to specifically delete Zfp260 in myeloid cells. Samples were collected from 3-month-old experimental and control mice for further analysis. Micro-CT and TRAP staining results indicated no significant phenotypic changes in the long bones of *Lyz2^{cre}; Zfp260^{fl/fl}* mice, including cortical bone thickness and trabecular bone density in the diaphysis region (Fig. R2-Ma3-e). Furthermore, TRAP staining demonstrated that the deletion of Zfp260 in myeloid cells did not affect the number of TRAP-positive cells (refer to Fig. R2-Ma3-f, marked by scarlet arrows).

The periodontitis model is widely utilized for investigating osteoclast maturation and differentiation [PMID: 33486152]. Upon establishing the periodontitis model in both control and experimental mice and conducting histological analyses, we observed no significant differences in bone resorption levels or osteoclast numbers in the root furcation area between the two groups (see Fig. R2-Ma3-g, indicated by cyan arrows).

Furthermore, the Raw264.7 cell line was employed for in vitro induction of osteoclasts, with transfections conducted using si-NC and si-Zfp260. Our observations revealed no significant differences in osteoclast size between the si-Zfp260 group and the control group (Fig. R2-Ma3-h). These findings indicate that Zfp260 exerts minimal influence on osteoclast development and maturation. Consequently, the conditional knockout of Zfp260 in the *Ctsk*-lineage is unlikely to produce phenotypic differences attributable to effects on osteoclastogenesis.

Fig. R2-Ma3 The role of Zfp260 in osteoclastogenesis (Scale Bar = 50µm)

4. What is the kinetics of Zfp260 expression during skeletal stem cell development under physiological conditions (early skeletal development)? Its expression and functions are specific to periosteal stem cell development?

Response to major point 4:

Thank you for your insightful inquiry. As reported by Debnath et al. [PMID: 30250253], Ctsk cells are detectable in the periosteum of long bones as early as embryonic day E14.5. In our study, we conducted co-staining of Collagen Type I and Zfp260 in mice at embryonic day E16.5, postnatal day 3 (P3), and postnatal day 7 (P7) to investigate the expression dynamics of Zfp260 during early skeletal development under physiological conditions. Our findings indicate that in long bones, Zfp260 is expressed not only in the periosteal tissue (magnified region 4), but also in the growth plate chondrocytes (magnified region 2), trabecular surfaces of the metaphysis, along with surrounding bone marrow stromal cells (Col-I⁺), are highlighted in the magnified region 3. As development advances to postnatal day 7 (P7), chondrocytes within the secondary ossification center progressively commence the expression of Zfp260, as depicted in magnified region 1. In the nasal bone tissue, both nasal chondrocytes and cells of the periosteal tissue exhibit expression of Zfp260, as shown below (Fig. R2-Ma4).

Fig. R2-Ma4 The kinetics of Zfp260 expression during early skeletal development (Scale Bar = 50 μ m)

Based on its distribution, it is reasonable to infer that the expression of Zfp260 is associated not only with the development of the periosteal stem cell lineage but also with the regulation of mesenchymal stromal cell development and regeneration. This study primarily investigates the effects of Zfp260 on the differentiation and function of skeletal stem cells during post-injury repair. Future study may further explore the regulatory roles and impacts of Zfp260 on skeletal stem cells, including those represented by the Ctsk lineage, as well as on other stem cell lineages.

5. It is important to recognize that callus size alone does not necessarily correlate with improved healing outcomes. The authors need to examine unionization rates of fracture sites and fibrosis formation in nonunionization sites.

Response to major point 5:

We express our gratitude for your professional insights. Following your recommendations, we investigated the unionization status and discovered that the unhealed fracture ends were enveloped by fibrotic tissue, encasing cartilage or calcified cartilage. This data has been compiled into a statistical graph (as shown below, Fig. R2-Ma5) and is presented in Fig. 2f of R1 manuscript.

Fig. R2-Ma5 The fracture healing status of control and Zfp260^{iΔPSC} mice at 28 dpf

6. *The rationale using CTSKCRE-ERT2 line, instead of CTSKCRE line, is not clear since tamoxifen treatment also affects osteoclast development during fracture healing. What are basal skeletal phenotypes of Zfp260^{fl/fl};CTSKCRE mice?*

Response to major point 6:

Thank you very much for your inquiry. The rationale for using CTSK-CreERT2 instead of CTSK-CRE is as follows:

During development, the Ctsk-Cre labeled periosteal stem cell lineage gradually contributes to terminally differentiated cells due to cumulative effects, resulting in the labeling of many terminally differentiated mature stromal cells, osteoblasts, and osteocytes. Consequently, the proportion of true stem/progenitor cell components being labeled is relatively reduced [PMID: 35319300]. According to the scRNA-seq results of both fracture and MSFL derived SSC lineage subclusters, Zfp260 is expressed to varying degrees in osteolineage, osteochondro-lineage, and stem & progenitors (see below, Fig. R2-Ma6-1). Using the Ctsk-CRE line for labeling may lead to Zfp260 knockout in a broader range of cell types beyond skeletal stem/progenitor cells, making the results difficult to interpret.

Fig. R2-Ma6-1 The feature plot and expression level of Zfp260 in each subclusters

Debnath et al. have demonstrated that the immediately labeled cells by Ctsk contain a higher proportion of stem and progenitor cells, with only a small amount of terminally differentiated mature cells [PMID: 30250253]. This immediate labeling approach is particularly suitable for studying the repair and regeneration processes initiated by stem/progenitor cells after injury. While this process requires tamoxifen injection to preemptively initiate cell labeling, as you mentioned, the use of tamoxifen might have some impact on osteoclastic and osteoblastic effects during fracture healing. However, this approach was applied to both the control and experimental groups, effectively neutralizing any potential positive or negative effects it may cause. Moreover, this CreER-tamoxifen induction strategy has been widely used in studying specific cell types or genes in the regulation of bone fracture. [PMID: 36888549; PMID: 36309308; PMID: 31643106; PMID: 29985470]

Regarding the developmental phenotype of Zfp260 knockout in the Ctsk-Cre lineage, as you mentioned, we have added images from micro-CT and histological analyses of 6-week-old mice, including body size, skull, and femur. Our observations indicated that *Ctsk^{cre};Zfp260^{fl/fl}* mice exhibit a reduction in body length and diminished skeletal mineralization level, as evidenced by Alizarin Red and Alcian Blue staining of the skeleton (Fig. R2-Ma6-2-a,b). There was a notable decrease in the calcification rate in both the femur and nasal bone (Fig. R2-Ma6-2-c). Micro-CT and histological analyses reveal significant cortical bone thinning at the femoral midshaft and metaphysis, accompanied by a reduction in trabecular bone (Fig. R2-Ma6-2-d,f,g). Additionally, the craniofacial bones demonstrated decreased bone density, with a particularly pronounced thinning of the nasal bone cortex (Fig. R2-Ma6-2-e,h). These alterations are observed on the bone surface, aligning with the periosteal distribution pattern of CTSK-lineage cells. This pattern is analogous to the phenotypic changes initially reported by Debnath et al. [PMID:

30250253] upon the knockout of Osx in CTSK-Cre lineage mice. The corresponding figure is shown below (Fig. R2-Ma6-2).

Fig. R2-Ma6-2 The basal skeletal phenotypes of *Ctsk^{cre};Zfp260^{fl/fl}* mice

7. Figure 4j and 6m: ARS staining represents mineralization activity only. To examine osteogenic development of SSCs, the authors need to perform CFU, alkaline phosphatase activity, and osteogenic gene expression. Kidney capsule transplantation of SSCs would be beneficial to assess bone forming activity of SSCs.

Response to major point 7:

We express our sincere gratitude for your professional recommendations. In response to your suggestions, we have incorporated additional experiments into the revised manuscript (R1), including colony-forming unit (CFU) assays, alkaline phosphatase (ALP) staining, and analysis of osteogenic gene expression in both wild-type and *Zfp260* knockout skeletal stem cells (SSCs). Our findings indicate that the knockout of *Zfp260* does not significantly influence the colony-forming capacity of SSCs; however, it markedly affects ALP activity and the expression of genes associated with osteogenesis.

Additionally, adhering to your expert guidance, we have conducted the renal subcapsular transplantation experiments using control and Zfp260 knockout SSCs. Utilizing micro-CT and flow cytometry analyses, we identified that the loss-of-function of Zfp260 resulted in compromised formation of mineralized nodules in skeletal stem cells (SSCs), an accumulation of pre-BCSP in the newly formed tissue, and a decreased proportion of osteolineage cells (combination of PCP, THY, BLSP), which is in line with our in-vivo findings. These findings have been integrated into Figure 4k-m of the revised manuscript (R1), **as elaborated below (Fig. R2-Ma7).**

Fig. R2-Ma7 The CFU-Fs, ALP assay, qPCR and renal subcapsular transplantation of SSCs from control and Zfp260^{ΔPSC} mice

8. Figure 6: Phospho-mass spectrometry data identifying phosphorylation sites Y173, S182, and S197 should be included in the figure. At least the phosphorylated peptide sequences should be provided.

Response to major point 8:

Thank you very much for your professional suggestions. Based on your advice, we have updated the phosphorylated peptide sequence in Fig. 6h of the R1-manuscript, **as shown below (Fig. R2-Ma8).**

Fig. R2-Ma8 The revised Fig. 6h displaying the phosphorylated peptide sequences

9. Figure 6f and 6l are not convincing to show nuclear and cytosolic localization of Zfp260. Nuclear staining is needed. Additionally, it is difficult to see the increase of Zfp260 in cytosol when it decreases in nucleus. Nuclear/cytosolic fractionation assay would be beneficial.

Response to major point 9:

Thank you for your valuable suggestions. We have indeed conducted DAPI nuclear staining, and the regions delineated by white dashed circles in the manuscript correspond to the DAPI-stained areas. However, the overlap between the nuclear staining and Zfp260 staining resulted in a less distinct Zfp260 signal in the merged channel. Consequently, we opted not to include the DAPI layer in the manuscript. Below, we provide a comparative analysis of the images before and after merging the DAPI channel (Fig. R2-Ma9-1).

Fig. R2-Ma9-1 A comparative analysis of the images before and after merging the DAPI channel

In response to your suggestions for supplementary data on Zfp260 expression levels following nuclear and cytoplasmic separation, we have conducted the necessary experiments and completed the Western blot (WB) analysis. The results from the nuclear-cytoplasmic separation Western blot analysis corroborate the results obtained from immunofluorescence staining, suggesting that the nuclear translocation of Zfp260 is modulated by PKC α -mediated phosphorylation. Additionally, mutations at the phosphorylation sites Y173, S182, and S197 were found to diminish the efficiency of Zfp260's nuclear translocation, thereby altering its nuclear-cytoplasmic distribution ratio. The pertinent results have been incorporated into Fig. 6g and Fig. 6n in the R1-manuscript, as shown below (Fig. R2-Ma9-2). The relative data has been incorporated into the Source Data table.

Fig. R2-Ma9-2 The Nuclear/cytosolic fractionation assay of Fig. 6f (Left) and Fig. 6l (Right)

10. Figure 7: Validation experiments showing the expression levels of *Zfp260* WT and mutants in AAV-treated *Zfp260* cKO mice are missing. AAV-mediated expression of a transgene under a short version of the *CTSK* promoter might be minimal.

Response to major point 10:

Thank you for your professional suggestions and reminders. Indeed, we had considered your concerns prior to conducting this portion of the experiment. We would like to address a labeling error: the promoter region we selected is the TSS+2kb region, and we have made the necessary corrections in the R1-manuscript.

The empty backbone length of the pAAV expression vector is 5.7kb, which includes 0.5kb of the CMV promoter and enhancer region. This vector can accommodate a maximum exogenous gene length of approximately 4kb to maintain its packaging capacity (PMID: 16014954; PMID: 16824801). This implies that the combined length of the substituted promoter and the exogenous gene must be constrained to within 4.5kb. Although the conventional promoter length is TSS+3kb, incorporating the CDS region of *Zfp260* (1224bp) would approach the upper limit of the pAAV expression vector's capacity, potentially resulting in vector instability and diminished virus packaging and in vivo infection efficiency. Upon examining H3K4me1 Cut&Tag data, we observed that in SSC lineage cells, the 2kb sequence upstream of the *Ctsk* promoter fully encompasses the K4me1 modification peak, as illustrated below (Fig. R2-Ma10-1). Consequently, we selected the TSS+2kb scheme for packaging the *Ctsk*pro-rAAV9.

Fig. R2-Ma10-1 The H3K4me1 peaks are located in the region of TSS+2kb of *Ctsk*

Additionally, employing small animal in vivo imaging, micro-CT imaging, and histological analysis, we observed that *Ctsk*pro-rAAV9 carrying *Zfp260*-wild type significantly ameliorated the phenotype of *Zfp260* conditional knockout (cKO) mice. Conversely, other mutants exhibited limited efficacy in rescuing the phenotype, suggesting that the TSS+2kb strategy for *Ctsk*pro-rAAV9 achieves both targeted and functional outcomes.

To enhance the rigor of our experimental approach, and in accordance with your recommendation, we administered an additional injection of rAAV9 to a new cohort of *Zfp260*-cKO mice. Subsequently, Lin⁻ZsGreen⁺ cells were isolated using flow cytometry, and quantitative PCR (qPCR) analysis was conducted. Our findings indicate an about

70%-230% increase in the transcriptional levels of the exogenous Zfp260 gene introduced via AAV9. The detailed results are presented below (Fig. R2-Ma10-2).

Fig. R2-Ma10 Zfp260 expression of FACS sorted Lin⁺ZsGreen⁺ cells from AAV infected control and Zfp260-cKO mice

Minor points:

1. Key words: *zfp260* and *Ctsk* need full names.

Response to minor point 1:

Thank you for your professional reminder. We have revised the Key Words section of the R1-manuscript accordingly and highlighted the changes in yellow.

2. Figure 3m: Which tissues were used for mIHC?

Response to minor point 2:

Thank you for your question. We performed mIHC on *in vitro* cultured PSCs to observe the distribution of SuperEnhancer structure elements within the cell nucleus.

3. No description of Figure 8 is included in text.

Response to minor point 3:

Thank you for your valuable suggestion. We have incorporated a summary of the study's findings in the first paragraph of the discussion section, ensuring alignment with the content depicted in the schematic diagram of Figure 8. And we have cited Figure 8 within this section in the R1 manuscript, highlighted in yellow.

“These results indicated that Zfp260 played a crucial role in regulating the transition of SSC to BCSP, thereby connecting this conversion process with the determination of the

osteogenic differentiation fate of BCSPs during the stem cell response to bone injury repair (Fig. 8)."

4. sFig. 8s should be changed to sFig7s.

Response to minor point 4:

Thank you very much for your suggestion. We have made the necessary revisions in the corresponding section of the R1-manuscript.

5. sFig.5a-g: Remove a-g.

Response to minor point 5:

Thank you very much for your suggestion. We have made the necessary revisions in the corresponding section of the R1-manuscript.

6. The subtitle, Deficiency of Zfp260 in PSCs resulted in decreased co-staining rates of Runx2 within the SSC hierarchy, should be rephrased. What is biological meaning of "co-staining rates"?

Response to minor point 6:

Thank you for your valuable suggestion. We have revised the subtitle in the manuscript to "Deficiency of Zfp260 in PSCs results in decreased Runx2 expression within the SSC hierarchy" and have highlighted this change in yellow. Given that Runx2⁺ is regarded as a marker of progenitor cells committed to osteogenic differentiation, we utilized this co-staining rate as an indicator to label the SSC hierarchy that has undergone fate determination in situ on tissue sections.

7. sFig.6a-g: Remove a-g.

Response to minor point 7:

Thank you very much for your suggestion. We have made the necessary revisions in the corresponding section of the R1-manuscript.

8. The diagram showing the generation of Zfp260-floxed mice should be added to Supplementary Figure since they are a novel mouse line.

Response to minor point 8:

Thank you very much for your professional advice. We have added a schematic diagram to depict the construction strategy for Zfp260-flox mouse in sFig. 8 of the R1-manuscript, **as shown below**.

SFig. 8 The construction strategy for Zfp260-flox mouse (C57BL/6J background)

The Zfp260 gene (NCBI Reference Sequence: NM_011981; Ensembl: ENSMUSG00000049421) is located on mouse chromosome 7. Three exons are identified, with the ATG start codon in exon 3 and the TAA stop codon in exon 3 (Transcript Zfp260-201: ENSMUST00000050735). Exon 3 will be selected as conditional knockout region (cKO region). The region contains 1224 bp coding sequence. Deletion of this region should result in the loss of function of the mouse Zfp260 gene. To engineer the targeting vector, homologous arms and cKO region will be generated by PCR using BAC clone RP23-137H12 as template. Ribonucleoprotein (RNP) and targeting vector will be co-injected into fertilized eggs for cKO mouse production.

9. Sequences of qPCR primers, Chip-seq primers, the CTSK promoter of AAV (TSS + 1kb) should be moved to Supplementary Table.

Response to minor point 9:

Thank you very much for your suggestions. We have moved the qPCR primer sequences, ChIP-seq primer sequences, and the CTSK promoter sequence of AAV in the Supplementary Table of the R1-manuscript.

Reviewer #3

In this study, Yuteng Weng and coauthors investigated the molecular mechanisms underlying bone healing processes. They performed comprehensive transcriptome analysis on specific cell types, including SSCs and BCSPs, and identified Zfp260 as a key transcription factor in the transition from SSCs to BCSPs. Through genetic studies using a conditional gene knockout approach, they found that Zfp260 KO resulted in a significant delay in bone healing and impaired osteogenesis. Further epigenetic analysis showed that Zfp260 is necessary for active enhancer markers and open chromatin associated with osteogenesis. Additionally, they identified crucial phosphorylation sites in the ZFP260 protein. Overall, the study provided significant findings in bone biology with convincing data, although several points need to be addressed.

We sincerely thank you for your thorough review of our manuscript and for providing professional and insightful guidance and suggestions for our revisions. In response to the comments you raised, we will address each one individually in the following sections.

Comments:

1. In Figure 1c, the process used to identify candidate transcription factors was not clear. The authors need to describe the details of this process and provide a comprehensive list of all transcription factor candidates they identified.

Response to comment 1:

Thank you for your professional advice. In response to your suggestions, we have refined the screening criteria for candidate transcription factors in the Results section and elaborated on the process of identifying candidate transcription factors within each module.

“To identify candidate transcription factors (TFs) that satisfy the dual criteria of high expression in SSC lineage cells and relevance to early SSC fate determination, we employed the following strategy: Utilizing scRNA-seq datasets from fracture and MSFL, we filtered for TFs that are specifically highly expressed in SSC lineage cells (encompassing stem and progenitor cells, osteolineage cells, and osteochondro lineage cells). The filtering parameters applied were avg Log2FC > 0.25, pct.1 > 0.25, pct.1 - pct.2 > 0.25, and p.adj < 0.05. Furthermore, a time course analysis of the bulk RNA-seq data from fracture and MSFL was conducted, with a focus on TFs significantly enriched in clusters associated with extracellular matrix organization. By intersecting the four identified TF clusters, we identified five potential candidate TFs: Zfp260, Plagl1, Snai2,

Creb3l1, and Prrx1 (Fig. 1c). All clusters and their corresponding features from the time course analysis were presented in sFig. 1a and b. The TF list filtered in each module has been supplemented in the Supplementary Table 2.”

Additionally, we revised the Fig. 1c (as shown below, Fig. R3-C1) and supplemented a comprehensive list of all TF candidates in the appendix.

Fig. R3-C1 The revised Figure 1c displaying the detailed screening strategy

2. Although the bulk RNA-seq data clearly showed upregulation of Zfp260 during bone healing, the expression of Zfp260 was not clearly presented in the scRNA-seq analysis. The authors need to show the expression as a feature map in Figure 1i and Figure 2h, and as a violin plot in Supplementary Figures S2b and S3b.

Response to comment 2:

Thank you for your professional advice. In response to your suggestion, we have incorporated the feature map of Zfp260 into Figures 1e and 1i of the revised R1 manuscript, as illustrated below. However, Figure 2h represents a flow cytometry clustering map in tSNE reduced form, rather than scRNA-seq data; therefore, no modifications have been made to this figure. Additionally, following your recommendation, we have depicted the expression of Zfp260 using violin plots in Supplementary Figures S2b and S3b, as shown below (Fig. R3-C2).

Fig. R2-C2 The feature plot and expression level of Zfp260 in each subclusters

3. In Figure 2f, toluidine blue stains cartilage matrix. The results clearly indicate that chondrogenesis was severely delayed in Zfp260 knockout. Immunostaining for chondrogenic markers, such as Type II collagen and SOX9, and osteogenic markers, such as Type I collagen and Osterix, are required.

Response to comment 3:

Thank you for your professional suggestions. We have also observed the delayed chondrogenesis phenotype upon Zfp260 knockout in the Ctsk lineage. Based on your valuable advice, we conducted co-staining for the chondrogenic marker (Col2) and the osteogenic marker (Col1), as shown below (Fig. R3-C3), and have updated the images accordingly in Fig. 2h of the revised manuscript (R1-manuscript).

Fig. R3-C3 The distributions of Col-I and Col-II in the EO or IO region during fracture healing

4. Related to comment 3, if chondrogenic differentiation is affected by Zfp260, how do the authors interpret this data? They may need to discuss whether Zfp260 is crucial for both osteogenesis and chondrogenesis.

Response to comment 4:

Thank you for your valuable reminder. Indeed, we observed a markedly reduced rate of cartilage matrix formation and mineralization in the *Zfp260*^{iAPSC} mouse fracture model. According to the findings of Debnath et al., Ctsk lineage cells play a crucial role in bone regeneration post-fracture via both intramembranous and endochondral ossification. Furthermore, the scRNA-seq data revealed that Zfp260 is also expressed in osteochondro-lineage cells. Therefore, we hypothesize that the phenotypic differences observed in the endochondral ossification region within the fracture model may be attributed to the loss of Zfp260 function in the subset of Ctsk lineage cells responsible for endochondral ossification. In vitro chondrogenic induction of PSCs revealed a loose cartilage structure and reduced expression of several cartilage-related markers in the chondrospheres with Zfp260 knockout, as shown below (Fig. R3-C4). To further investigate the role of Zfp260 in endochondral ossification and cartilage development, future studies should employ transgenic mice with Col2a or Sox9 promoter-driven Cre recombinase to achieve targeted knockout of Zfp260.

This discussion above has been incorporated into the revised manuscript (R1-manuscript) and highlighted in yellow.

Fig. R3-C4 In vitro chondrogenesis assay of control and Zfp260-KO PSCs
(scale bar = 100 μ m)

5. In Figure 3j, what are the actual expression differences in Runx2 genes between wild-type and Zfp260 gene knockout? The authors need to show Runx2 expression between WT and KO in both BF and MSFL studies.

Response to comment 5:

Thank you for your professional suggestions. We have incorporated the expression data of Runx2 in both fracture and MSFL-derived Lin⁺ZsGreen⁺ cells between the control and *Zfp260*^{iAPSC} groups in Figure 3k of the revised manuscript (R1-manuscript). Our observations indicate that Runx2 expression is significantly downregulated in the *Zfp260* conditional knockout group, as shown below (Fig. R3-C5).

Fig. R3-C5 The TPM of Runx2 in both models displayed in Fig. 3k of R1 manuscript

6. In Figures 4f and 4g, the authors selectively showed enrichment of ATAC signals associated with specific terms. However, it is important to present a more comprehensive view. What gene ontology terms were most enriched in the analysis of differentially accessible regions? Were there any upregulated ATAC signals? What do these regions represent?

Response to comment 6:

Thank you very much for your valuable suggestions. These specific GO ontologies represent the essential pathways through which stem cells gradually lose stemness, secrete extracellular matrix, and differentiate into mature functional cells. These three representative Gene Ontologies illustrate the potential stages at which *Zfp260* may be involved in regulating the opening of ATAC signals, which is why we emphasized these three Gene Ontologies in the initial manuscript.

In accordance with your expert advice, we have now provided the top 10 ranked differential up-regulated gene ontologies for each stage (SSC, pre-BCSP, BCSP, and osteolineage cells) in the WT vs. *Zfp-260* KO SSC lineage, as shown below (R3-C6).

At the SSC stage, our analysis revealed that the knockout of *Zfp260* predominantly influences the openness of ATAC signals in genes associated with actin cytoskeleton organization and cell morphology, which play crucial roles in the regulation of cellular structure. This observation aligns with prior studies indicating that the early fate determination of stem cells is initiated by their response to mechanical forces, leading to subsequent adaptive modifications in their cytoskeleton [PMID: 15068789; PMID: 15068789].

23839578. Regarding cellular behavior, Zfp260 modulates the ATAC signal accessibility of genes associated with cell migration and locomotory behavior, indicating a diminished migratory capacity of SSCs in response to injury.

Notably, Zfp260 knockout in MSFL-derived SSCs also alters the ATAC signal accessibility of genes involved in extracellular matrix organization, collagen fibril organization, and bone mineralization. The data suggest that the SSC population derived from MSFL demonstrates a superior regenerative capacity compared to SSCs derived from fractures. This observation may partially corroborate our earlier findings that bone regeneration mediated by MSFL is independent of patient age and gender [PMID: 35821090].

At the pre-BCSP stage, the most significantly enriched gene ontology category among the increased open ATAC signals was "multicellular organism development," indicating that Zfp260 plays a role in regulating the transition and cell fate determination at this stage.

At the BCSP stage, the upregulated ATAC signals were predominantly enriched in fundamental intracellular biochemical processes, including RNA transcription, protein translation, phosphorylation, chromatin organization, and cell cycle regulation. Analysis of the profile curves in Fig. 4d and 4e revealed that the loss of Zfp260 function had the most pronounced effect on the opening of ATAC signals at this stage. This finding suggests that the activation of ATAC signals associated with transcription and translation genes plays a crucial role in determining the subsequent fate of BCSPs.

These findings indicate that Zfp260 is pivotal at each stage of the transition from SSCs to osteolineage cells, ensuring the maintenance of open ATAC signals within key stage-specific gene ontologies, thereby systematically guiding the osteogenic differentiation trajectory of SSCs. Additionally, we have appended the table from Fig. R3-C6 to the supplementary file in the R1-manuscript.

Fracture-SSC (Con vs KO up-regulated)		p-value	MSFL-SSC (Con vs KO up-regulated)		p-value
GO:0072659-protein localization to plasma membrane		9.86E-04	GO:0030036-actin cytoskeleton organization		1.29E-15
GO:0008360-regulation of cell shape		0.003148801	GO:0016477-cell migration		2.14E-10
GO:0007626-locomotory behavior		0.005314959	GO:0030198-extracellular matrix organization		2.95E-09
GO:0006366-transcription from RNA polymerase II promoter		0.007344334	GO:0030199-collagen fibril organization		4.04E-08
GO:0032922-circadian regulation of gene expression		0.009299198	GO:0030282-bone mineralization		1.83E-07
GO:0006468-protein phosphorylation		0.011239567	GO:0030335-positive regulation of cell migration		4.30E-07
GO:0007191-adenylate cyclase-activating dopamine receptor signaling pathway		0.012022351	GO:0001525-angiogenesis		4.58E-07
GO:0032755-positive regulation of interleukin-6 production		0.013228523	GO:0045944-positive regulation of transcription from RNA polymerase II promoter		4.66E-07
GO:0001764-neuron migration		0.01350183	GO:0007155-cell adhesion		4.78E-07
GO:0048762-mesenchymal cell differentiation		0.017239372	GO:0001501-skeletal system development		4.86E-07
Fracture-proBCSP (Con vs KO up-regulated)		p-value	MSFL-proBCSP (Con vs KO up-regulated)		p-value
GO:0007275-multicellular organism development		1.43E-07	GO:0007275-multicellular organism development		4.46E-06
GO:0007411-axon guidance		2.04E-06	GO:0007399-nervous system development		6.51E-05
GO:0043547-positive regulation of GTPase activity		3.61E-06	GO:0043547-positive regulation of GTPase activity		1.49E-04
GO:0016055-Wnt signaling pathway		2.02E-05	GO:0045944-positive regulation of transcription from RNA polymerase II promoter		2.31E-04
GO:0090630-activation of GTPase activity		6.00E-05	GO:0090630-activation of GTPase activity		4.11E-04
GO:0007399-nervous system development		6.27E-05	GO:0031175-neuron projection development		9.95E-04
GO:0030154-cell differentiation		1.14E-04	GO:0007416-synapse assembly		0.00113625
GO:0030182-neuron differentiation		1.56E-04	GO:0006816-calcium ion transport		0.003419378
GO:0001649-osteoblast differentiation		3.52E-04	GO:0030154-cell differentiation		0.003510462
GO:0016477-cell migration		6.01E-04	GO:0007264-small GTPase mediated signal transduction		0.004177957
Fracture-BCSP (Con vs KO up-regulated)		p-value	MSFL-BCSP (Con vs KO up-regulated)		p-value
GO:0045944-positive regulation of transcription from RNA polymerase II promoter		1.09E-20	GO:0015031-protein transport		8.80E-10
GO:0045893-positive regulation of transcription, DNA-templated		5.50E-19	GO:0007049-cell cycle		1.03E-09
GO:0006468-protein phosphorylation		2.57E-13	GO:0045893-positive regulation of transcription, DNA-templated		8.67E-09
GO:0016310-phosphorylation		4.32E-13	GO:0016310-phosphorylation		5.72E-08
GO:0007049-cell cycle		5.71E-12	GO:0007420-brain development		3.30E-07
GO:0015031-protein transport		1.23E-10	GO:0051301-cell division		3.63E-07
GO:0016477-cell migration		4.83E-10	GO:0030968-endoplasmic reticulum unfolded protein response		3.96E-07
GO:0006325-chromatin organization		1.90E-08	GO:0032922-circadian regulation of gene expression		3.34E-06
GO:0060271-cilium assembly		4.49E-08	GO:0006338-chromatin remodeling		4.76E-06
GO:0006338-chromatin remodeling		9.05E-08	GO:0030036-actin cytoskeleton organization		5.39E-06
Fracture-Osteo (Con vs KO up-regulated)		p-value	MSFL-Osteo (Con vs KO up-regulated)		p-value
GO:0043547-positive regulation of GTPase activity		9.83E-05	GO:0030199-collagen fibril organization		2.58E-08
GO:0030198-extracellular matrix organization		0.004137581	GO:0045944-positive regulation of transcription from RNA polymerase II promoter		2.64E-08
GO:0006325-face morphogenesis		0.005459438	GO:0008284-positive regulation of cell proliferation		4.34E-06
GO:0007275-multicellular organism development		0.006078819	GO:0043410-positive regulation of MAPK cascade		5.60E-06
GO:0007528-neuromuscular junction development		0.006220743	GO:0030335-positive regulation of cell migration		6.34E-06
GO:0071277-cellular response to calcium ion		0.01130625	GO:0007275-multicellular organism development		9.53E-06
GO:0043149-stress fiber assembly		0.016219321	GO:0030198-extracellular matrix organization		1.49E-05
GO:1904754-positive regulation of vascular associated smooth muscle cell migration		0.017595931	GO:0030036-actin cytoskeleton organization		2.05E-05
GO:0022409-positive regulation of cell-cell adhesion		0.022008649	GO:0007155-cell adhesion		2.27E-05
GO:0001501-skeletal system development		0.025206374	GO:00060349-bone morphogenesis		2.59E-05

Fig. R3-C6 The comprehensive view of ATAC signals enriched Gene ontology terms

7. In the abstract, the authors wrote “Zfp260 drove chromatin opening”; however, there is no evidence that Zfp260 plays a role in opening the Runx2 regulatory regions. The authors only showed that Zfp260 was required for the maintenance of open chromatin states in these regions. The authors need to revise this statement.

Response to comment 7:

We appreciate your meticulous and constructive feedback. Based on your recommendations, we have revised the abstract in the R1 manuscript to state, "**Zfp260 was essential for the maintenance of open chromatin states.**"

Response to the referee comments on NCOMMS-24-37853-R2

We would like to thank all three referees for their kind reviews of our manuscript. We are gratified that the reviewers have expressed satisfaction with our response. We have carefully studied all comments from the reviewers, and made corresponding modifications. The following are our point-to-point responses:

Reviewer #1

The authors have thoroughly responded to the comments on the original submission, and the revised manuscript is felt to be technically solid and of interest for the field.

As a very minor point, it is suggested that this text change highlighted by the authors in the rebuttal letter be slightly modified as indicated:

“Previous research has demonstrated that periosteal skeletal stem cells (PSCs), a periosteal SSC population defined by the expression of Ctsk together as a primary biomarker together with additional cell surface markers located on the outer membrane of bone, were crucial for the healing of long bone fractures, facilitating both intramembranous and endochondral ossification processes.” It is felt to be useful to highlight that lineage reporters alone (e.g, CTSK-cre) cannot define these cells, but really the combination of the lineage reporter, surface markers and anatomic location. This was felt to be helpful to clarify as it is a frequent point of confusion in the field.

Response

Thank you for your positive feedback on our revised manuscript. We are pleased to learn that our responses and the additional data have been recognized. We greatly appreciate your professional editing suggestions, and we have replaced the original wording in the manuscript with your recommended description to more clearly define this key stem cell population of periosteal stem cells. Once again, thank you for your guidance and for your contributions to research in the field of bone biology.

Reviewer #2

All of the reviewer's comments were well responded along with additional new data. No more comments by the reviewer.

Response

Thank you to for your positive feedback on our revised manuscript. We are pleased to learn that our response and additional data have been acknowledged.

Reviewer #3

The authors addressed the reviewer's comments and revised the manuscript properly. The reviewer accepts the revised manuscript.

Response

Thank you to for your positive feedback on our revised manuscript. We are pleased to learn that our response and additional data have been acknowledged.